# ERIS: Enhancing Privacy and Communication Efficiency in Serverless Federated Learning

## Abstract

Scaling federated learning (FL) to billion-parameter models introduces critical trade-offs between communication efficiency, network load distribution, model accuracy, and privacy guarantees. Existing solutions often tackle these challenges in isolation, sacrificing accuracy or relying on costly cryptographic tools. We propose ERIS, a serverless FL framework that balances privacy and accuracy while eliminating the server bottleneck and significantly reducing communication overhead. ERIS combines a model partitioning strategy, distributing aggregation across multiple client-side aggregators, with a distributed shifted gradient compression mechanism. We theoretically prove that ERIS (i) converges at the same rate as FedAvg under standard assumptions, and (ii) bounds mutual information leakage inversely with the number of aggregators, enabling strong privacy guarantees with no accuracy degradation. Extensive experiments on image and text datasets—ranging from small networks to modern large language models—confirm our theory: compared to six baselines, ERIS consistently outperforms all privacy-enhancing methods and matches the accuracy of non-private FedAvg, while reducing model distribution time by up to $1000\times$ and communication cost by over 94%, lowering membership inference attack success rate from $\sim$83% to $\sim$65%—close to the unattainable $\sim$64% limit—and reducing data reconstruction to random-level quality. ERIS establishes a new Pareto frontier for scalable, privacy-preserving FL for next-generation foundation models without relying on heavy cryptography or noise injection.

## 1 Introduction

The widespread digitalization has led to an unprecedented volume of data being continuously recorded. However, most of these data are sensitive, introducing privacy risks and regulatory constraints that limit its usability (EU, 2024). Federated Learning (FL) has emerged as a distributed and privacy-preserving paradigm that enables multiple devices (clients) to collaboratively train machine learning (ML) models without sharing their private local data (McMahan et al., 2017). By decentralizing training, FL can incorporate a much broader range of potential data sources—moving beyond publicly available web data or isolated institutional datasets—to include sensitive distributed data from corporations, hospitals, vehicles, and personal devices that would otherwise remain inaccessible.

Despite its potential to democratize access to richer and more diverse training data, FL faces critical challenges that hinder its large-scale development. First, large-scale data availability necessitates high-capacity models capable of accurately capturing diverse data distributions, such as foundation models and large language models (LLMs) (Khan et al., 2025). However, as model sizes grow, FL training becomes increasingly impractical due to prohibitive communication costs. In traditional FL, the server synchronously transmits updated models to all clients in each round. With modern large models easily exceeding billions of parameters (Devlin et al., 2019; OpenAI, 2023), this process overloads the server's network connection, creating a major bottleneck that limits scalability. Reducing the number of transmitted parameters can mitigate communication costs but typically degrades performance (Jiang et al., 2023; Haddadpour et al., 2021). Similar limitations hold for parameter-efficient fine-tuning (PEFT) for large pre-trained models, which remain consistently outperformed by full model fine-tuning (Raje et al., 2025; Sun et al., 2024). Second, although FL prevents direct data sharing, exchanged gradients still encode sensitive information about the underlying training data, posing privacy risks. Adversaries may exploit these gradients to reconstruct input data or infer whether specific samples were used for training (Yue et al., 2023; Hu et al., 2021; Bai et al., 2024). Existing

privacy-preserving solutions attempt to mitigate these risks, but often introduce trade-offs, sacrificing either model accuracy or training efficiency (Geyer et al., 2018; Shen et al., 2024; Zhou et al., 2023).

To address these challenges and unlock the full potential of FL, we introduce ERIS, a novel, scalable, serverless FL framework that significantly reduces communication costs and enhances privacy. Unlike existing decentralized learning approaches that fragment collaboration, ERIS fully preserves the model utility of standard FL. To the best of our knowledge, ERIS is the first FL framework to simultaneously achieve decentralized aggregation, strong communication efficiency, and provable information-theoretic privacy guarantees without sacrificing model utility. ERIS is also the first to extend privacy-enhancing federated training to modern LLMs, demonstrating feasibility at scale where prior methods fail to preserve utility and efficiency. **Our key contributions are:**

- We introduce a novel gradient partitioning scheme that decentralizes the aggregation process across multiple aggregators (clients) without introducing approximation errors—ensuring that the final model remains mathematically equivalent to FedAvg, while removing bottlenecks and balancing network load to maximize efficiency.
- We combine decentralized aggregation with a distributed shifted compression mechanism that reduces transmitted parameters to less than 3.3% of the model size and cuts distribution time by up to three orders of magnitude in the worst case—pushing communication efficiency to its limits while preserving model convergence and utility. We provide convergence guarantees and empirical results across three image and two text datasets, from small networks to LLMs.
- We prove theoretically and empirically that ERIS's gradient partitioning mitigates privacy leakage without noise injection or cryptographic overhead. Since no single entity observes a full client update—only a small, randomized subset—the privacy risk is reduced and scales with the number of aggregators. Experiments on four model architectures and six SOTA baselines under two common threat models confirm ERIS's superior privacy–utility trade-off.

## 2 BACKGROUND

**Traditional Federated Learning.**  Traditional FL systems (McMahan et al., 2017) consist of $K \in \mathbb{N}$ clients, denoted by $\mathcal{K} = \{1, 2, \ldots, K\}$, coordinated by a central server to collaboratively train an ML model over a distributed dataset $D$. Each client $k \in \mathcal{K}$ holds a private dataset $D_k = \{d_{k,s}\}_{s=1}^{S_k}$ with $S_k$ samples. During each training round, clients independently update their model parameters $\mathbf{x^t} \in \mathbb{R}^n$ by minimizing a nonconvex local loss $f(D_k; \mathbf{x^t})$. Local updates are expressed in terms of stochastic gradients $\tilde{\mathbf{g}}_k^t$. After local training, each client transmits its gradients to the server, which aggregates them using a permutation-invariant operation, and updates the global model as $\mathbf{x}^{t+1} = \mathbf{x}^t - \lambda_t \tilde{\mathbf{g}}^t$, where $\lambda_t$ is the learning rate. The updated global model $\mathbf{x}^{t+1}$ is broadcast back to the clients, serving as initialization for the next training round. In general, FL aims to minimize:

$$\arg\min_{\mathbf{x}} \frac{1}{K} \sum_{k=1}^{K} f(D_k; \mathbf{x}), \quad \text{where} \quad f(D_k; \mathbf{x}) := \frac{1}{S_k} \sum_{s=1}^{S_k} f(d_{k,s}; \mathbf{x}). \tag{1}$$

For clarity, in the rest of the paper, we denote the loss function of the current model as $f(\mathbf{x}^t)$ for the entire dataset $D$, $f_k(\mathbf{x}^t)$ for the local dataset $D_k$, and $f_{k,s}(\mathbf{x}^t)$ for a single sample $d_{k,s}$, respectively.

**Communication efficiency.**  Communication between the server and clients is widely recognized as the primary limitation in optimizing the efficiency of traditional FL systems. The two key communication challenges that hinder scalability are a *network utilization imbalance* and *high communication costs per round*. First, as *the number of clients grows*, FL introduces a severe imbalance in network utilization, leading to server-side congestion, which makes large-scale FL impractical. Decentralized architectures can alleviate this issue by distributing communication across multiple nodes, balancing network utilization and reducing overload on any single node (Kalra et al., 2023; Chen et al., 2023). However, existing architectures often restrict collaboration to local neighbor exchanges, which reduces the collaborative power of traditional FL and results in client-specific models. Second, as *model sizes increase*—reaching billions of parameters—the volume of transmitted data per round escalates, amplifying communication costs and making it impractical to train large models efficiently. Compression techniques, such as quantization (Michelusi et al., 2022; Zhao et al., 2022a) and sparsification (Richtarik et al., 2022; Li et al., 2022d), mitigate communication overhead by reducing the amount of transmitted data. However, naive compressions often degrade model utility or require additional reconstruction steps, increasing the total number of rounds (Li et al., 2020).

**Privacy protection.** Although FL prevents direct data sharing, the training process still exposes transmitted information—such as gradients—that can reveal sensitive information (Zhang et al., 2023a;b; He et al., 2024). The server represents the primary vulnerability in FL, as it collects full client gradients, directly derived from private data during optimization. To ensure user-level privacy, two main approaches have been proposed. The first relies on *cryptographic techniques*, such as secure aggregation (Chen et al., 2019; Reagen et al., 2021), and trusted execution environments (Zhao et al., 2022b; Yazdinejad et al., 2024), to mask client gradients from the server. However, these methods introduce significant computational overhead or require specialized hardware. The second approach perturbs gradients using privacy-preserving mechanisms such as *local differential privacy (LDP)* (Xie et al., 2021; Ziegler et al., 2022) or *model pruning* (Zhang et al., 2023c; Bibikar et al., 2022), which reduce privacy leakage but often degrade model utility. In this work, we focus on perturbation-based methods, which provide software-level privacy without requiring cryptographic infrastructure.

## 3 ERIS

In this work, we propose ERIS, a novel serverless FL framework designed to address key limitations of traditional systems. This section formalizes the problem setting (Section 3.1), describes the pipeline (Section 3.2), provides theoretical foundations for the convergence of the learning process (Section 3.3), and establishes an information-theoretic upper bound on the privacy leakage (Section 3.4).

### 3.1 PROBLEM DEFINITION

We consider a traditional distributed environment with $K$ clients, each holding a private dataset $D_k = \{d_{k,s}\}_{s=1}^{S_k}$ with $S_k$ samples. Our objective is to collaboratively train a global model while addressing the following challenges: (i) optimizing network bandwidth usage by equally distributing the computational and communication load across the network without introducing approximation errors during aggregation; (ii) reducing the number of transmitted parameters while ensuring model convergence and maintaining utility; and (iii) minimizing the information available to an honest-but-curious adversary, thereby reducing privacy leakage without degrading the learning process.

**Assumption 3.1** (Smoothness). Each local function $f_{i,j}$ is $L$-smooth: there exists $L \geq 0$ such that

$$f_{i,j}(x_1) \leq f_{i,j}(x_2) + \langle \nabla f_{i,j}(x_2), x_1 - x_2 \rangle + \frac{L}{2}\|x_1 - x_2\|^2, \quad \forall x_1, x_2 \in \mathbb{R}^d. \tag{2}$$

**Assumption 3.2** (Unbiased local estimator). The gradient estimator $\tilde{\mathbf{g}}_k^t$ is unbiased $\mathbb{E}_t[\tilde{\mathbf{g}}_k^t] = \nabla f_k(\mathbf{x^t})$, where $\mathbb{E}_t$ is the expectation conditioned on all history before round $t$, and there exist $C_1, C_2, C_3, C_4$ $\theta$ such that

$$\mathbb{E}_t\Big[\frac{1}{K}\sum_{k=1}^{K}\|\tilde{\mathbf{g}}_k^t - \nabla f_k(\mathbf{x^t})\|^2\Big] \leq C_1\Delta^t + C_2 \tag{3a}$$

$$\mathbb{E}_t[\Delta^{t+1}] \leq (1-\theta)\Delta^t + C_3\|\nabla f(\mathbf{x}^t)\|^2 + C_4\mathbb{E}_t[\|\mathbf{x}^{t+1} - \mathbf{x}^t\|^2] \tag{3b}$$

*Remark* 3.3. The parameters $C_1$ and $C_2$ capture the variance of the gradient estimators, e.g., $C_1 = C_2 = 0$ if the client computes local full gradient $\tilde{\mathbf{g}}_i^t = \nabla f_i(\mathbf{x^t})$, and $C_1 \neq 0$ and $C_2 = 0$ if the client uses variance-reduced gradient estimators such as SVRG/SAGA.

### 3.2 THE ERIS PIPELINE

The ERIS pipeline is detailed in Algorithm 1 and shown in Figure 1, which outlines the client-side computation and distributed learning process at round $t$. At each round, each client computes one or more local updates using a (stochastic) gradient estimator $\tilde{\mathbf{g}}$, such as SGD, SAGA (Defazio et al., 2014), or stochastic variance-reduced gradient (SVRG) (Johnson & Zhang, 2013), on its dataset $D_k$. Before transmission, clients perform two key operations: *shifted compression* and *model partitioning*.

**Shifted Compression.** We begin by introducing the standard definition of an unbiased compressor, widely adopted in FL algorithms (Li et al., 2020; Li & Richtarik, 2021; Gorbunov et al., 2021).

**Definition 3.4** (Compression operator). A randomized map $\mathcal{C} : \mathbb{R}^n \to \mathbb{R}^n$ is an $\omega$-compression operator if for all $\mathbf{x} \in \mathbb{R}^n$, it satisfies $\omega \geq 0$ and:

$$\mathbb{E}(\mathcal{C}(\mathbf{x})) = \mathbf{x}, \qquad \mathbb{E}\big[\|\mathcal{C}(\mathbf{x}) - \mathbf{x}\|^2\big] \leq \omega\|\mathbf{x}\|^2 \tag{4}$$

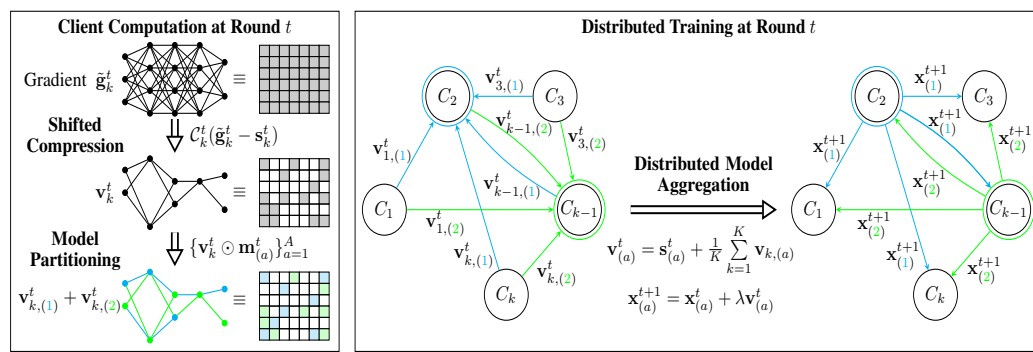

Figure 1: Illustration of ERIS at training round $t$ for two aggregators ($A = 2$). **Left:** each client performs shifted compression and model partitioning, generating shards $\mathbf{v}^t_{k,(a)}$ sent to aggregators $C_2$ and $C_{k-1}$. **Right:** each aggregator collects and aggregates the corresponding shards across clients to produce partial updated models $\mathbf{x}^{t+1}_{(a)}$, which are then sent back to the clients.

*Remark* 3.5. Definition 3.4 encompasses a wide range of common compressors like random quantization and sparsification (Devlin et al., 2019; Alistarh et al., 2017; Li et al., 2020; Li & Richtarik, 2021; Li et al., 2022d). For instance, random sparsification in FL can be represented as $\mathcal{C}^t_k(\mathbf{x}) = \mathbf{x} \odot \mathbf{m}_{\mathcal{C}^t_k}$, where $\mathbf{m}_{\mathcal{C}^t_k}$ is a scaled binary mask with entries equal to $1/p_k$ with probability $p_k$ and 0 otherwise, ensuring $\mathbb{E}[\mathbf{m}_{\mathcal{C}^t_k}] = \mathbf{1}_d$ and $\omega = \frac{1-p_k}{p_k}$. The mask can vary between clients and time, enabling dynamic adjustment of the compression throughout the learning process.

To improve convergence behavior (i.e., reduce communication rounds), we extend the shifted compression of Li et al. (2022d) to a distributed setting. Each client maintains a local reference vector $\mathbf{s}^t_k$, and compresses the shifted gradients $\tilde{\mathbf{g}}^t_k - \mathbf{s}^t_k$ (Line 4 in Algorithm 1). This vector is iteratively updated to track the compressed gradient as $\mathbf{s}^{t+1}_k = \mathbf{s}^t_k + \gamma^t \mathcal{C}^t_k(\tilde{\mathbf{g}}^t_k - \mathbf{s}^t_k)$, where $\gamma^t$ is the step-size.

**Model Partitioning.** After shifted compression, each client partitions the compressed gradient vector $\mathbf{v}^t_k$ into $A$ disjoint shards using a structured masking scheme to avoid information loss. Let $\{\mathbf{m}^t_{(a)}\}^A_{a=1} \subset \{0,1\}^d$ denote a set of categorical masks at round $t$, where each mask $\mathbf{m}^t_{(a)}$ satisfies:

$$\text{Disjointness:} \quad \mathbf{m}^t_{(a)} \odot \mathbf{m}^t_{(a')} = 0 \quad \forall a \neq a', \qquad \text{Completeness:} \sum_{a=1}^{A} \mathbf{m}^t_{(a)} = \mathbf{1}_d,$$

where $\mathbf{1}_d$ is the all-ones vector, and $\odot$ denotes element-wise multiplication. These masks partition the gradient $\mathbf{v}^t_k$ into $A$ non-overlapping shards as defined in Line 5. Each shard $\mathbf{v}^t_{k,(a)}$ is transmitted to its corresponding aggregator $a$, ensuring that parameter updates are distributed across the network. The masks $\{\mathbf{m}^t_{(a)}\}$ can be either (i) predefined via a deterministic or random partition shared across clients (e.g., interleaved indices) or (ii) dynamically sampled by each client at each round.

**Distributed Model Aggregation.** Each aggregator $a$ receives $\{\mathbf{v}^t_{k,(a)}\}^K_{k=1}$ from all participating clients and computes a permutation-invariant aggregation over its assigned subset of parameters. To account for the client-level shifted compression, the aggregator adds the global reference vector $\mathbf{s}^t_{(a)}$ to the aggregated shard. The resulting aggregated shards $\mathbf{v}^t_{(a)}$ are then used to update the corresponding segments of the global model $\mathbf{x}^t_{(a)}$ (Lines 9–10 in Algorithm 1). These updated segments are broadcast back to all clients to synchronize the next training round. Concurrently, the aggregator updates the global reference vector as $\mathbf{s}^{t+1}_{(a)} = \mathbf{s}^t_{(a)} + \gamma_t \frac{1}{K} \sum_{k=1}^{K} \mathbf{v}^t_{k,(a)}$ for the next round.

### 3.3 THEORETICAL ANALYSIS OF CONVERGENCE AND UTILITY

As established in Appendix B.1, the distributed aggregation process across $A$ aggregators in ERIS introduces no loss of information or deviation in algorithm convergence. Building on this result, we present the following theorem, which characterizes the utility and communication efficiency of ERIS.

---

**Algorithm 1:** ERIS

---

**Input:** Initial global model $\mathbf{x}^0$, number of aggregators $A$, learning rate $\lambda_t$, number of clients $K$, number of communication rounds $T$, initial reference vector $\mathbf{s}_k^0 = 0$
**Output:** Final global model $\mathbf{x}^T$

1 **for** $t = 0, 1, \ldots, T-1$ **do**
    // Client-side operations
2    **for** *each client* $k \in \{1, \ldots, K\}$ *in parallel* **do**
3        Compute local stochastic gradient $\tilde{\mathbf{g}}_k^t$ ;
4        *Compression:* $\mathbf{v}_k^t = \mathcal{C}_k^t\big(\tilde{\mathbf{g}}_k^t - \mathbf{s}_k^t\big)$ ;
5        *Privacy:* Partition compressed gradient into $A$ shards: $\{\mathbf{v}_{k,(a)}^t\}_{a=1}^A = \{\mathbf{v}_k^t \odot \mathbf{m}_{(a)}^t\}_{a=1}^A$
6        Send each shard $\mathbf{v}_{k,(a)}^t$ to aggregator $a$ for $a = 1, \ldots, A$ ;
7        Update reference vector $\mathbf{s}_k^{t+1} = \mathbf{s}_k^t + \gamma^t \mathbf{v}_k^t$ ;
    // Aggregator-side operations
8    **for** *each aggregator* $a \in \{1, \ldots, A\}$ *in parallel* **do**
9        Aggregates compressed information and compensates shift $\mathbf{v}_{(a)}^t = \mathbf{s}_{(a)}^t + \frac{1}{K}\sum_{k=1}^K \mathbf{v}_{k,(a)}^t$ ;
10      Updates shard of the global model $\mathbf{x}_{(a)}^{t+1} = \mathbf{x}_{(a)}^t + \lambda \mathbf{v}_{(a)}^t$ ;
11      Updates reference $\mathbf{s}_{(a)}^{t+1} = \mathbf{s}_{(a)}^t + \gamma_t \frac{1}{K}\sum_{k=1}^K \mathbf{v}_{k,(a)}^t$ ;
12      Broadcast updated shard $\mathbf{x}_{(a)}^{t+1}$ to all clients ;
13    Each client $k$ reassembles the global model $\mathbf{x}_k^{t+1} = \sum_{a=1}^A \mathbf{m}_{(a)}^t \odot \mathbf{x}_{(a)}^{t+1}$

---

**Theorem 3.6** (Utility and communication for ERIS). *Consider* ERIS *under Assumptions 3.1 and 3.2, where the compression operators $\mathcal{C}_k^t$ satisfy Definition 3.4. Let the learning rate be defined as:*

$$\lambda_t \equiv \lambda \leq \min\left\{ \frac{\sqrt{\beta K}}{\sqrt{1 + 2\alpha C_4 + 4\beta(1+\omega)}(1+\omega)L}, \frac{1}{(1 + 2\alpha C_4 + 4\beta(1+\omega) + 2\alpha C_3/\lambda^2)L} \right\}, \quad (5)$$

*where $\alpha = \frac{3\beta C_1}{2(1+\omega)L^2\theta}$, for any $\beta > 0$, and let the shift stepsize be $\gamma_t = \sqrt{\frac{1+2\omega}{2(1+\omega)^3}}$. Then, ERIS satisfies the following utility bound:*

$$\frac{1}{T}\sum_{t=0}^{T-1} \|\nabla f(\mathbf{x}^t)\|^2 \leq \frac{2\Phi_0}{\lambda T} + \frac{3\beta C_2}{(1+\omega)L\lambda}, \quad (6)$$

*where $\Phi_0 := f(\mathbf{x}^0) - f^* + \alpha L \Delta^0 + \frac{\beta}{KL}\sum_{k=1}^K \|\nabla f_k(\mathbf{x}^0) - \mathbf{s}_k^0\|^2$. Equation 6 implies that the asymptotic utility of* ERIS *is governed by by the gradient estimator variance $C_2$, which vanishes for lower-variance estimators such as SVRG/SAGA. Similarly, a larger local batch size reduces gradient variance, leading to improved convergence, with $C_2 = 0$ when full local gradients are used.*

Theorem 3.6 establishes a convergence guarantee for ERIS, providing a utility bound that holds for common gradient estimators such as SGD, SAGA, and SVRG, which satisfy Assumption A.2. In contrast to prior communication-efficient privacy-preserving FL methods (Ding et al., 2021a; Li et al., 2022d; Lowy et al., 2023), the bound in equation 6 depends primarily on the gradient-estimator variance $C_2$ and is independent of the specific privacy-preserving mechanism applied; notably, it contains no term that grows with $T$. The proof and additional details are provided in Appendix C.

## 3.4 THEORETICAL ANALYSIS OF PRIVACY GUARANTEES

We analyze ERIS under the standard *honest-but-curious* threat model (Huang et al., 2021; Gupta et al., 2022; Arevalo et al., 2024), where an adversary observes and stores transmitted model updates (e.g., via eavesdropping, or compromised aggregator/server) and attempts to infer sensitive information about clients' private data $D_k$. ERIS inherently reduces information leakage through two mechanisms: (i) compression via an operator $\mathcal{C}_k^t$ with mask $\mathbf{m}_{\mathcal{C}_k^t}$, and (ii) partitioning via disjoint masks $\{\mathbf{m}_{(a)}^t\}_{a=1}^A$, ensuring that a fixed adversary observes at most $n/A$ (random) parameters per round. To quantify privacy, we bound the mutual information between $D_k$ and the adversary's view $\mathbf{v}_{k,(a)}^t$ over $T$ rounds.

**Theorem 3.7** (Privacy guarantee of ERIS). *Let* $\mathbf{v}_{k,(a)}^t = (\tilde{\mathbf{g}}_k^t - \mathbf{s}_k^t) \odot \mathbf{m}_{\mathcal{C}_k^t} \odot \mathbf{m}_{(a)}^t$ *denote the $a$-th compressed shard of client $k$ at round $t$, where $\mathbf{m}_{\mathcal{C}_k^t}$ is a compression mask satisfying Definition 3.4 with probability $p$, and $\mathbf{m}_{(a)}^t$ selects one of $A$ disjoint shards. Assume that $\max_{i,t,\mathcal{H}_t} I(D_k; \mathbf{x}_{k,i}^t \mid \mathcal{H}_t) < \infty$, where $\mathcal{H}_t$ denotes the full history up to round $t$ of the revealed masked updates and weights. Then, under the honest-but-curious model, the mutual information over $T$ rounds satisfies:*

$$I_k = I\big(D_k; \{\mathbf{v}_{k,(a)}^{t+1}\}_{t=0}^{T-1}\big) \leq n \, T \, \frac{p}{A} \, C_{\max}, \tag{7}$$

*where $n$ is the model size and $C_{\max}$ bounds the per-coordinate mutual information at any round.*

*Remark* 3.8. If each weight satisfies $\mathbf{x}_{k,i}^{t+1} \mid D_k, \mathcal{H}_t \sim \mathcal{N}(\mu(D_k), \sigma_{\text{cond}}^2)$ and $\mathbf{x}_{k,i}^{t+1} \mid \mathcal{H}_t \sim \mathcal{N}(\mu, \sigma^2)$ independently of $(i, t)$ and $\mathcal{H}_t$, then by the entropy of Gaussians:

$$C_{\max} = \sup_{i,t,\mathcal{H}_t} I\big(D_k; \mathbf{x}_{k,i}^{t+1} \mid \mathcal{H}_t\big) = \sup_{i,t,\mathcal{H}_t} \big[H(\mathbf{x}_{k,i}^{t+1} \mid \mathcal{H}_t) - H(\mathbf{x}_{k,i}^{t+1} \mid D_k, \mathcal{H}_t)\big] = \tfrac{1}{2} \ln \frac{\sigma^2}{\sigma_{\text{cond}}^2}.$$

Theorem 3.7 shows that the leakage bound scales as $n \, T \, \frac{p}{A}$, and hence decreases with stronger compression (lower retention probability $p$) and a larger number of shards $A$, which together reduce the number of observable parameters per round. Full proofs and the extension to colluding adversaries are deferred to Appendix D, where we also empirically verify that model weights follow the Gaussian assumption of Remark 3.8. These theoretical findings are also corroborated by our experiments.

## 4 RESULTS

In this section, we present the experimental setup and numerical results evaluating the privacy-utility tradeoff of ERIS. We compare its performance to SOTA methods, showing its effectiveness in balancing communication efficiency, accuracy, and privacy across diverse real-world FL scenarios.

### 4.1 EXPERIMENTAL SETUP

**Datasets.** We evaluate ERIS on five publicly available datasets spanning image classification and text generation. For image classification, we use MNIST (LeCun et al., 2005) and CIFAR-10 (Krizhevsky et al., 2009); for text classification, IMDB Reviews (Maas et al., 2011); and for text generation, CNN/DailyMail (See et al., 2017). To evaluate data reconstruction attacks, we additionally use LFW (Huang et al., 2008). Datasets are randomly partitioned among $K$ clients ($K=10$ for CNN/DailyMail, $K=25$ for IMDB, and $K=50$ for the others); while non-IID scenarios are generated using a Dirichlet distribution with $\alpha \in \{0.2, 0.5\}$. We adopt GPT-Neo (Black et al., 2021) (1.3B) and DistilBERT (Sanh et al., 2019) (67M) as pre-trained models for CNN/DailyMail and IMDB, respectively, and train ResNet-9 (He et al., 2016) (1.65M) and LeNet-5 (Lecun et al., 1998) (62K) from scratch for CIFAR-10, MNIST, and LFW. All experiments use 5-fold cross-validation, and reported results are averaged across folds. Training hyperparameters are detailed in Appendix E.1.

**Baselines.** We compare ERIS against several state-of-the-art methods for communication efficiency and client-side privacy in FL: *Ako* (Watcharapichat et al., 2016) and *Shatter* (Biswas et al., 2025), decentralized approaches with partial gradient exchange; *SoteriaFL* (Li et al., 2022d), which combines centralized shifted compression with differential privacy; *PriPrune* (Chu et al., 2024), a pruning strategy that withholds the most informative gradient components from communication; and *LDP* (Sun et al., 2021b). We also include *FedAvg* (McMahan et al., 2017) as the standard baseline with no defenses or compression, and report results for an idealized upper bound (*Min. Leakage*), where clients transmit no gradients and the attack is applied only to the last-round global model.

**Privacy Attacks.** Under the standard honest-but-curious model, we assume the attacker is a compromised aggregator or server with access to client-transmitted gradients. We evaluate five representative attacks across two widely studied categories: *Membership Inference Attacks (MIA)* and *Data Reconstruction Attacks (DRA)*. For MIA, we adopt the privacy auditing framework of Steinke et al. (2023), repeating the evaluation at each round for every client; for text generation, we adapt the SPV-MIA of Fu et al. (2024) to our auditing setting. Reported results correspond to the maximum, over all $T$ rounds, of the average MIA accuracy across $K$ clients. For DRA, we consider the strongest white-box threat model, which assumes access to the gradient of a single training sample, and implement DLG (Zhu et al., 2019), iDLG (Zhao et al., 2020), and ROG (Yue et al., 2023), with

the latter specifically designed for reconstruction from obfuscated gradients. Reconstruction quality is measured with LPIPS, SSIM, and SNR. Implementation details are provided in Appendix E.2.

## 4.2 NUMERICAL EXPERIMENTS

**Effect of Model Partitioning and Shifted Compression.** We first analyze how the two key mechanisms of ERIS affect privacy leakage. Figure 2 (left) reports the impact of *model partitioning* on MIA accuracy, as a function of the number of aggregators $A$, evaluated on MNIST. Consistent with Theorem 3.7, increasing $A$ significantly reduces privacy leakage by limiting the number of observable parameters per round without affecting model accuracy. Notably, the experimental trend closely mirrors the linear dependency predicted by the theoretical bound on mutual information. Figure 2

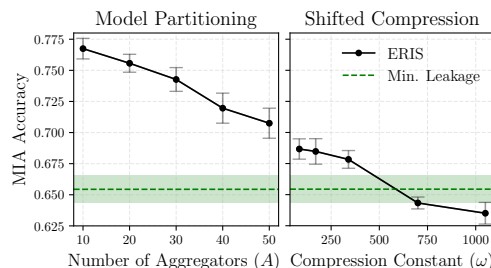

Figure 2: Effect of model partitioning (**left**) and shifted compression (**right**) on privacy.

(right) shows the impact of the *compression* constant $\omega$ with $A = 50$ fixed: stronger compression (higher $\omega$, i.e., lower retention probability $p$) steadily drives MIA accuracy toward the idealized minimum-leakage baseline. These results empirically validate Theorem 3.7, underscoring the role of shifted compression in reducing MIA risk, while Appendix F.3 quantifies its effect on model utility. For DRA, we find that compression alone is insufficient, especially against the ROG attack (Table 5), whereas partitioning is highly effective: even with $A = 2$ (i.e., half of the gradient exposed), reconstructions are highly distorted and no longer preserve meaningful features of the original.

**Balancing Utility and Privacy.** To evaluate the utility–privacy trade-off, we benchmark ERIS against SOTA baselines across settings that influence memorization and overfitting. First, we vary model capacity, a key factor in memorization, spanning from large-scale architectures with 1.3B parameters on CNN/DailyMail to lightweight models with 62K parameters on MNIST. Second, we control overfitting by varying the number of training samples per client—from 4 to 128. Figure 3 shows that ERIS (blue) consistently maintains high utility, on par with non-private FedAvg (orange), while significantly reducing privacy leakage—approaching the idealized upper-bound of the *Min. Leakage* scenario. In contrast, privacy-preserving methods like FedAvg-LDP, PriPrune, and SoteriaFL reduce leakage only at the cost of severely degraded performance. This confirms prior findings (Li et al., 2022c) that DP can significantly impair utility, particularly for large models, resulting in low privacy leakage largely due to the model's inability to effectively learn the task. Notably, while Shatter's partial gradient exchange offers privacy protection comparable to or weaker than ERIS, its fragmented collaboration substantially slows convergence, particularly when models are trained from scratch. Table 1 summarizes mean and MIA accuracy, averaged over varying client training samples. These results confirm that ERIS achieves the best overall utility–privacy trade-off among all baselines. Appendix F.7 reports the same experiments under non-IID setting, confirming equivalent conclusions.

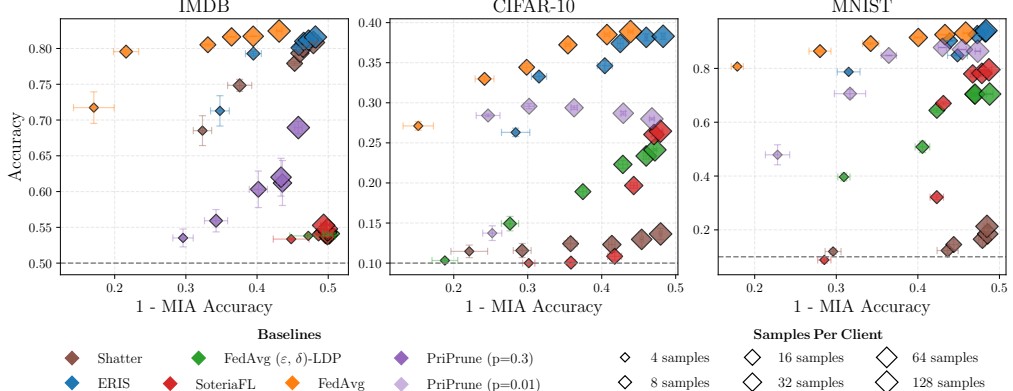

Figure 3: Comparison of test accuracy and MIA accuracy across varying model capacities (one per dataset) and client-side overfitting levels, controlled via the number of training samples per client.

| Method | CNN/DailyMail – GPT-Neo | | IMDB – DistilBERT | | CIFAR-10 – ResNet9 | | MNIST – LeNet5 | |
|---|---|---|---|---|---|---|---|---|
| | R-1 (↑) | MIA Acc. (↓) | Acc. (↑) | MIA Acc. (↓) | Acc. (↑) | MIA Acc. (↓) | Acc. (↑) | MIA Acc. (↓) |
| FedAvg | $33.22 \pm 0.99$ | $97.94 \pm 0.63$ | $79.60 \pm 0.83$ | $68.21 \pm 1.36$ | $34.86 \pm 0.31$ | $68.46 \pm 0.96$ | $88.91 \pm 0.35$ | $65.11 \pm 0.78$ |
| FedAvg $(\epsilon, \delta)$-LDP | $26.00 \pm 0.28$ | $51.98 \pm 3.13$ | $53.97 \pm 0.04$ | $50.55 \pm 1.18$ | $19.00 \pm 0.47$ | $63.35 \pm 0.85$ | $61.03 \pm 1.03$ | $57.24 \pm 0.59$ |
| SoteriaFL $(\epsilon, \delta)$ | $25.40 \pm 0.70$ | $52.14 \pm 2.97$ | $54.24 \pm 0.15$ | $51.25 \pm 1.19$ | $17.18 \pm 0.24$ | $58.83 \pm 0.56$ | $57.27 \pm 0.88$ | $57.13 \pm 0.56$ |
| PriPrune $(p_1)$ | $24.67 \pm 4.64$ | $71.35 \pm 2.83$ | $74.15 \pm 1.00$ | $66.36 \pm 1.13$ | $26.30 \pm 0.39$ | $65.67 \pm 0.84$ | $77.41 \pm 1.52$ | $62.21 \pm 1.01$ |
| PriPrune $(p_2)$ | $24.67 \pm 4.64$ | $71.35 \pm 2.83$ | $66.30 \pm 2.14$ | $63.61 \pm 1.11$ | $11.24 \pm 0.71$ | $56.55 \pm 0.78$ | $27.36 \pm 1.04$ | $52.69 \pm 0.83$ |
| PriPrune $(p_3)$ | $24.67 \pm 4.64$ | $71.35 \pm 2.83$ | $60.32 \pm 1.98$ | $60.54 \pm 1.03$ | $10.01 \pm 0.01$ | $54.86 \pm 0.77$ | $17.83 \pm 0.60$ | $52.01 \pm 0.80$ |
| Shatter | $31.95 \pm 0.71$ | $70.49 \pm 4.03$ | $76.94 \pm 1.40$ | $57.41 \pm 2.01$ | $12.40 \pm 1.85$ | $63.02 \pm 2.01$ | $15.86 \pm 4.82$ | $56.23 \pm 1.50$ |
| ERIS | $32.83 \pm 0.78$ | $69.55 \pm 3.94$ | $79.07 \pm 0.80$ | $56.31 \pm 0.81$ | $34.68 \pm 0.48$ | $60.48 \pm 0.91$ | $89.00 \pm 0.23$ | $55.97 \pm 0.77$ |
| Min. Leakage | $33.23 \pm 0.99$ | $60.53 \pm 4.83$ | $79.68 \pm 0.36$ | $55.58 \pm 0.76$ | $34.92 \pm 0.29$ | $58.85 \pm 0.93$ | $88.90 \pm 0.40$ | $55.22 \pm 0.64$ |

Table 1: Mean test performance (ROUGE-1 for CNN/DailyMail, accuracy for others) and MIA accuracy, averaged over varying local sample sizes. For DP-based methods, $\epsilon=10$; for PriPrune, pruning rates are $p \in \{0.1, 0.2, 0.3\}$ on IMDB/CNN-DailyMail and $p \in \{0.01, 0.05, 0.1\}$ on others.

**Pareto Analysis under Varying Privacy Constraints.** To further investigate the utility–privacy trade-off, we evaluate each method under varying strengths of its respective privacy-preserving mechanism: for DP-based methods (e.g., SoteriaFL, FedAvg-LDP), we vary the privacy budget $\epsilon$ and clipping thresholds; for PriPrune, the pruning rate; and for ERIS and Shatter, we add LDP on top of their native masking. Full configurations are provided in Appendix F.9. Figure 4 plots accuracy against MIA accuracy on CIFAR-10 under 16 training samples per client. The Pareto front represents the set of trade-off solutions for which no method achieves better utility without incurring higher privacy leakage, or vice versa. ERIS consistently contributes a majority of the points on the Pareto front, confirming its ability to balance privacy and utility more effectively than the baselines.

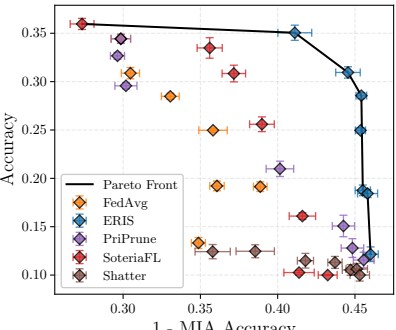

Figure 4: Utility–privacy trade-off on CIFAR-10 under varying strengths of the privacy-preserving mechanisms.

**Communication Efficiency.** Table 2 compares communication efficiency across methods, measured by per-client upload size and minimum distribution time per round (assuming 20MB/s bandwidth), using the same experimental setting that produced the results in Table 1. Results show that ERIS achieves dramatic improvements over all baselines. On CNN/DailyMail, where a 1.3B-parameter pre-trained model is used, ERIS reduces the upload size from 5.2GB in FedAvg to only 52MB (1%), and cuts distribution time from 5200s to less than 4.7s. On CIFAR-10, comparable gains

| Method | CNN/DailyMail | | CIFAR-10 | |
|---|---|---|---|---|
| | Exchanged | Dist. Time | Exchanged | Dist. Time |
| FedAvg (-LDP) | 5.2GB (100%) | 5200s | 6.6MB (100%) | 33s |
| Shatter | 5.2GB (100%) | 780s | 6.6MB (100%) | 1.32s |
| PriPrune (0.01) | 4.68GB (90%) | 4680s | 6.53MB (99%) | 32.65s |
| PriPrune (0.05) | 4.16GB (80%) | 4160s | 6.27MB (95%) | 31.35s |
| PriPrune (0.1) | 3.64GB (70%) | 3640s | 5.9MB (90%) | 29.5s |
| SoteriaFL | 0.26GB (5%) | 260s | 0.33MB (5%) | 1.65s |
| ERIS | 46.8MB (1%) | 4.68s | 0.04MB (0.6%) | 0.0039s |

Table 2: Communication efficiency: per-client upload and minimum distribution time per round.

are observed: communication drops to 0.6% of the full gradient, while distribution time decreases from 33s to 0.004s. These gains stem from two complementary mechanisms: (i) *shifted compression*, which reduces transmitted parameters by orders of magnitude without harming convergence; and (ii) *decentralized aggregation*, which balances network load and removes the server bottleneck. However, unlike prior decentralized learning methods, ERIS preserves full collaborative power of traditional FL: the final aggregated model is equivalent to FedAvg, with no loss of client contributions. Together, these properties enable ERIS to scale seamlessly to billion-parameter models. A full scalability analysis, detailing the effect of increasing clients and model size, is provided in Appendix F.2.

## 5 DISCUSSION

### 5.1 RELATED WORKS

**Decentralized and Communication-Efficient FL.** To alleviate the server bottleneck and improve network scalability, numerous decentralized approaches have emerged (Kalra et al., 2023; Liu et al., 2022; Bornstein et al., 2023). These methods can be grouped into two categories: (i) *peer-to-peer synchronization* schemes, where clients directly exchange updates with selected neigh-

bours (Watcharapichat et al., 2016; Roy et al., 2019; Shi et al., 2023; Zehtabi et al., 2024); and (ii) *gossip-based protocols*, which rely on randomized message passing to propagate updates across the network (Hu et al., 2019; Pappas et al., 2021; Kempe et al., 2003; Bornstein et al., 2023; Zehtabi et al., 2024). In parallel, compression techniques have been proposed to reduce communication overhead per round. These include quantization (Karimireddy et al., 2019; Li et al., 2020; Reisizadeh et al., 2020; Li & Richtarik, 2021; Gorbunov et al., 2021; Mishchenko et al., 2023) and sparsification (Li & Richtarik, 2021; Richtarik et al., 2022; Li et al., 2022d; Ivkin et al., 2019; Gorbunov et al., 2021; Khirirat et al., 2018), which limit the size of transmitted updates. Though effective in balancing network load and reducing bandwidth, these methods can hinder convergence and, similarly, offer no provable privacy guarantees. A few methods, such as Ako (Watcharapichat et al., 2016) and C-DFL (Liu et al., 2022), integrate decentralized architectures with partitioning or compression to improve communication efficiency, but do not consider privacy leakage in their design.

**Privacy-Preserving FL.** Among perturbation-based privacy-preserving mechanisms, two prominent approaches have been widely explored to mitigate client-side leakage from gradient sharing: *LDP* (Bai et al., 2024; Kairouz et al., 2021; Girgis et al., 2021; Ziegler et al., 2022; Lowy et al., 2023; Miao et al., 2022; Adnan et al., 2022; Yang et al., 2024) and *gradient pruning* (Jiang et al., 2023; Chu et al., 2024; Shen et al., 2024; Zhang et al., 2023c; Sun et al., 2021a; Bibikar et al., 2022; Li et al., 2021). LDP methods typically apply gradient clipping followed by random noise injection to each client's updates, providing formal privacy guarantees. Pruning-based techniques, instead, reduce leakage by systematically removing the most informative gradient components. While effective in limiting information exposure, both approaches often incur substantial utility degradation—especially when applied to large models (Li et al., 2022c). To attenuate this, recent works such as LotteryFL (Li et al., 2021) and PriPrune (Chu et al., 2024) propose personalized pruning schemes tailored to each client's data and model state, aiming to preserve performance while reducing leakage. Other methods combine LDP with compression to balance communication efficiency and privacy protection (Agarwal et al., 2018; Zong et al., 2021; Ding et al., 2021a; Li et al., 2022d; Jin et al., 2023), though often at the cost of increased algorithmic complexity or reduced convergence speed.

## 5.2 LIMITATIONS AND FUTURE WORKS

While ERIS demonstrates strong empirical and theoretical performance, it also introduces trade-offs. First, decentralizing the aggregation process shifts coordination to clients, which may vary in computational resources and connection stability—particularly in cross-device settings. However, the aggregation workload per node is significantly reduced compared to centralized FL, as each aggregator processes only a fraction of the total parameters (at most $n/A$), making the requirement substantially lighter. For cross-silo deployments, this is typically not an issue; in cross-device scenarios, minimal resource requirements may be needed to ensure reliable participation as an aggregator. Second, ERIS provides its strongest privacy guarantees when aggregators operate independently. In the presence of collusion among multiple honest-but-curious aggregators, the privacy benefits gradually diminish. Nonetheless, as shown in Corollary D.2, the mutual information leakage still scales linearly with the number of colluding nodes, and remains significantly lower than in traditional FL, where full gradients are exposed to a single entity. In future work, we plan to analyze the impact of poisoning attacks and exploit ERIS's decentralized design to integrate secure aggregation schemes.

## 6 CONCLUSION

We introduced ERIS, a novel FL framework that achieves high utility, strong privacy protection, and communication efficiency by decentralizing aggregation, employing shifted compression, and introducing gradient partitioning. Unlike existing methods, ERIS avoids central bottlenecks, balances network utilization, and formalizes privacy guarantees through an information-theoretic lens—ensuring no single entity observes full client updates. We provide theoretical convergence bounds and privacy guarantees, and validate them through extensive experiments across diverse datasets and model scales. ERIS consistently outperforms state-of-the-art privacy-preserving baselines, achieving a better utility–privacy trade-off without compromising scalability. Our results demonstrate that effective privacy preservation in FL does not require sacrificing performance with perturbation-based mechanisms—nor relying on heavy cryptographic assumptions. ERIS lays the foundation for practical, large-scale distributed training of large models that are both efficient and privacy-aware.

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

## Appendix Contents

APPENDIX

This appendix is organised as follows. Section A restates the main assumptions used in our theoretical analysis. Section B proves that ERIS maintains the same convergence behaviour as FedAvg in the absence of compression. Section C presents the convergence and communication analysis for the full ERIS framework. Section D provides the privacy guarantees and includes an extension of Theorem 3.7 to colluding aggregators. Section E details our experimental setup, including models, hyperparameters, privacy attacks, datasets, licenses, and hardware for full reproducibility. Finally, Section F.1–F.9 reports additional experimental results that support our claims, including evaluations of scalability, compression, data reconstruction attacks, and privacy–utility trade-offs under varying heterogeneity conditions (IID and non-IID) and both biased and unbiased gradient estimators.

## A  ASSUMPTIONS

For clarity and completeness of the Appendix, we restate the core assumptions used in the main theorems—Theorem 3.6 and Theorem 3.7. These include smoothness and unbiased local estimator conditions commonly adopted in the FL literature.

**Assumption A.1** (Smoothness). There exists some $L \geq 0$, such that for all local functions $f_{i,j}$ (indexed by $i \in [n]$ and $j \in [m]$), we have

$$\|\nabla f_{i,j}(x_1) - \nabla f_{i,j}(x_2)\| \leq L \|x_1 - x_2\|, \quad \forall x_1, x_2 \in \mathbb{R}^d, \tag{8}$$

or equivalently expressed with the following general bound:

$$f_{i,j}(x_1) \leq f_{i,j}(x_2) + \langle \nabla f_{i,j}(x_2), x_1 - x_2 \rangle + \frac{L}{2}\|x_1 - x_2\|^2. \tag{9}$$

**Assumption A.2** (Unbiased local estimator). The gradient estimator $\tilde{\mathbf{g}}_k^t$ is unbiased $\mathbb{E}_t[\tilde{\mathbf{g}}_k^t] = \nabla f_k(\mathbf{x^t})$ for $k \in \mathcal{N}$, where $\mathbb{E}_t$ takes the expectation conditioned on all history before round $t$. Moreover, there exist constants $C_1$, and $C_2$ such that:

$$\mathbb{E}_t\big[\frac{1}{K}\sum_{k=1}^{K}||\tilde{\mathbf{g}}_k^t - \nabla f_k(\mathbf{x^t})||^2\big] \leq C_1\Delta^t + C_2 \tag{11a}$$

$$\mathbb{E}_t[\Delta^{t+1}] \leq (1-\theta)\Delta^t + C_3||\nabla f(\mathbf{x}^t)||^2 + C_4\mathbb{E}_t[||\mathbf{x}^{t+1} - \mathbf{x}^t||^2] \tag{11b}$$

*Remark* A.3. The parameters $C_1$ and $C_2$ capture the variance of the gradient estimators, e.g., $C_1 = C_2 = 0$ if the client computes local full gradient $\tilde{\mathbf{g}}_i^t = \nabla f_i(\mathbf{x^t})$, and $C_1 \neq 0$ and $C_2 = 0$ if the client uses variance-reduced gradient estimators such as SVRG/SAGA.

## B  CONVERGENCE OF ERIS–BASE (NO COMPRESSION)

This section shows that ERIS–Base—Algorithm 1 instantiated with the identity compressor $\mathcal{C}_k^t = \mathrm{Id}$ and with the reference vectors fixed to zero so that only model partitioning is active—produces exactly the same global iterate sequence as the standard single–server algorithm (e.g., *FedAvg*). Consequently, every convergence guarantee proved for FedAvg carries over verbatim. The proof is algebraic and does not rely on any additional smoothness or convexity assumptions beyond those already stated in Section A (or Section 3.1).

**Notation.** Recall that client $k$ holds $S_k$ data points and that $S := \sum_{k=1}^{K} S_k$. Let $\tilde{\mathbf{g}}_k^t$ denote the (possibly stochastic) gradient that client $k$ transmits at communication round $t$ and write $\tilde{\mathbf{g}}^t = \frac{1}{S}\sum_{k=1}^{K} S_k \tilde{\mathbf{g}}_k^t$ for the sample–weighted mean gradient.

**Theorem B.1** (Convergence equivalence of ERIS–Base). *Run Algorithm 1 with $A \geq 1$ aggregators, $\mathcal{C}_k^t = \mathrm{Id}$, and $\mathbf{s}_k^t = \mathbf{0}$ for all $k, t$. Let $\mathbf{x}^t$ with $t \geq 0$ be the resulting iterates and let $\tilde{\mathbf{x}}^t$ be the iterates obtained by FedAvg ($A = 1$) using the same initialization, learning rates $\lambda_t$, and client gradients $\tilde{\mathbf{g}}_k^t$. Then for every round $t \geq 0$*

$$\mathbf{x}^t = \tilde{\mathbf{x}}^t. \tag{11}$$

*Hence all convergence bounds that hold for FedAvg under Assumptions A.1 and A.2 (with $\omega = 0$) apply unchanged to ERIS–Base.*

*Sketch.* Partition each client gradient into $A$ disjoint coordinate shards using the categorical masks $\{\mathbf{m}^t_{(a)}\}^A_{a=1}$ introduced in Section 3.2: $\tilde{\mathbf{g}}^t_{k,(a)} = \tilde{\mathbf{g}}^t_k \odot \mathbf{m}^t_{(a)}$. Because the masks are disjoint and sum to the all–ones vector, the original gradient decomposes exactly as $\tilde{\mathbf{g}}^t_k = \sum^A_{a=1} \tilde{\mathbf{g}}^t_{k,(a)}$. Aggregator $a$ forms the weighted average of its shard

$$\bar{\mathbf{g}}^t_{(a)} \; := \; \frac{1}{S} \sum^K_{k=1} S_k \, \tilde{\mathbf{g}}^t_{k,(a)}. \tag{12}$$

Summing over all aggregators and swapping summation order yields

$$\sum^A_{a=1} \bar{\mathbf{g}}^t_{(a)} = \frac{1}{S} \sum^K_{k=1} S_k \sum^A_{a=1} \tilde{\mathbf{g}}^t_{k,(a)} = \frac{1}{S} \sum^K_{k=1} S_k \, \tilde{\mathbf{g}}^t_k = \tilde{\mathbf{g}}^t. \tag{13}$$

ERIS–Base therefore updates the global model via $\mathbf{x}^{t+1} = \mathbf{x}^t - \lambda_t \sum^A_{a=1} \bar{\mathbf{g}}^t_{(a)} = \mathbf{x}^t - \lambda_t \tilde{\mathbf{g}}^t$, which is exactly the FedAvg rule. By induction on $t$ the iterates coincide. $\qquad\square$

*Remark B.2.* The identity above is purely algebraic, hence it remains valid when clients perform multiple local SGD steps, when the data are non IID, or when the global objective is nonconvex (e.g., (McMahan et al., 2017; Li et al., 2019; 2022b)). The key insight is that splitting the gradient vector dimension-wise introduces no additional approximation error; the final aggregated gradient is mathematically identical to that obtained by a single server aggregating all client gradients in one place. This ensures that the convergence behavior of ERIS-base matches that of traditional federated learning approaches, while its sole effect is to distribute network load.

## C  UTILITY AND COMMUNICATION FOR ERIS

In this section, we present the proof of Theorem 3.6 (Utility and communication for ERIS), modifying the general proof strategy of (Li et al., 2022d) to accommodate our decentralized setting, model partitioning, and the absence of differential privacy.

### C.1  PROOF OF THEOREM 3.6

*Proof.* Let $\mathbb{E}_t$ denote the expectation conditioned on the full history up to round $t$. By invoking Theorem B.1, we simplify the analysis by omitting model partitioning and treating $\mathbf{v}^t$ as the aggregated update. Thus, the update rule becomes $\mathbf{x}^{t+1} = \mathbf{x}^t - \lambda_t \mathbf{v}^t$. We now apply this rule within the smoothness inequality equation 9:

$$\mathbb{E}_t[f(\mathbf{x}^{t+1})] \le \mathbb{E}_t\left[ f(\mathbf{x}^t) - \lambda_t \langle \nabla f(\mathbf{x}^t), \mathbf{v}^t \rangle + \frac{\lambda^2 L}{2} \|\mathbf{v}^t\|^2 \right] \tag{14}$$

First, to verify the unbiased nature of $\mathbf{v}^t$, we consider:

$$\mathbb{E}_t[\mathbf{v}^t] = \mathbb{E}_t\left[ \mathbf{s}^t + \frac{1}{K} \sum^K_{k=1} \mathbf{v}^t_k \right]$$

$$= \mathbb{E}_t\left[ \frac{1}{K} \sum^K_{k=1} \mathbf{s}^t_k + \frac{1}{K} \sum^K_{k=1} \mathcal{C}^t_k(\tilde{\mathbf{g}}^t_k - \mathbf{s}^t_k) \right]$$

$$\overset{(4)}{=} \mathbb{E}_t\left[ \frac{1}{K} \sum^K_{k=1} \tilde{\mathbf{g}}^t_k \right] = \frac{1}{K} \sum^K_{k=1} \mathbb{E}_t[\tilde{\mathbf{g}}^t_k] \overset{(a)}{=} \frac{1}{K} \sum^K_{k=1} \nabla f_k(\mathbf{x}^t) = \nabla f(\mathbf{x}^t) \tag{15}$$

where (a) due to Assumption A.2, which states that each $\tilde{\mathbf{g}}^t_k$ is an unbiased estimator of $\nabla f_k(\mathbf{x}^t)$ (i.e., $\mathbb{E}_t[\tilde{\mathbf{g}}^t_k] = \nabla f_k(\mathbf{x}^t)$).

Substituting Equation equation 15 into equation 14, we obtain:

$$\mathbb{E}_t[f(\mathbf{x}^{t+1})] \le \mathbb{E}_t\left[ f(\mathbf{x}^t) - \lambda_t \|\nabla f(\mathbf{x}^t)\|^2 + \frac{\lambda^2 L}{2} \|\mathbf{v}^t\|^2 \right] \tag{16}$$

We further bound the term $\mathbb{E}_t[||\mathbf{v}^t||^2]$ in Lemma C.1, whose proof is available in the Appendix C.2.

**Lemma C.1.** *Consider that $\mathbf{v}^t$ is constructed according to Algorithm 1, it holds that*

$$\mathbb{E}_t[\|\mathbf{v}^t\|^2] \le \mathbb{E}_t\left[\frac{(1+\omega)}{K^2}\sum_{k=1}^{K}\|\tilde{\mathbf{g}}_k^t - \nabla f_k(\mathbf{x}^t)\|^2\right] + \frac{\omega}{K^2}\sum_{k=1}^{K}\|\nabla f_k(\mathbf{x}^t) - \mathbf{s}_k^t\|^2 + \|\nabla f(\mathbf{x}^t)\|^2. \quad (17)$$

To further our analysis, we now derive upper bounds for the first two terms on the right-hand side of Equation equation 17. The first term can be controlled using Equation equation 11a from Assumption A.2, yielding the bound $C_1\Delta^t + C_2$. Next, we establish that the second term decreases over time, as formalized in the following lemma (proof available in Appendix C.3).

**Lemma C.2.** *Let Assumption A.1 hold, and let the shift $\mathbf{s}_k^{t+1}$ be updated according to Algorithm 1. Then, for $\gamma_t = \sqrt{\frac{1+2\omega}{2(1+\omega)^3}}$, we have:*

$$\mathbb{E}_t\left[\frac{1}{K}\sum_{k=1}^{K}\|\nabla f_k(\mathbf{x}^{t+1}) - \mathbf{s}_k^{t+1}\|^2\right] \le \mathbb{E}_t\left[\left(1 - \frac{1}{2(1+\omega)}\right)\frac{1}{K}\sum_{k=1}^{K}\|\nabla f_k(\mathbf{x}^t) - \mathbf{s}_k^t\|^2\right.$$

$$+ \frac{1}{(1+\omega)K}\sum_{k=1}^{K}\|\tilde{\mathbf{g}}_k^t - \nabla f_k(\mathbf{x}^t)\|^2$$

$$\left. + 2(1+\omega)L^2\|\mathbf{x}^{t+1} - \mathbf{x}^t\|^2\right]. \quad (18)$$

For clarity, we introduce the notation: $\mathcal{S}^t := \frac{1}{K}\sum_{k=1}^{K}\|\nabla f_k(\mathbf{x}^t) - \mathbf{s}_k^t\|^2$. For some $\alpha \ge 0, \beta \ge 0$, we now define a potential function to analyze the convergence behavior:

$$\Phi_t := f(\mathbf{x}^t) - f^* + \alpha L\Delta^t + \frac{\beta}{L}\mathcal{S}^t, \quad (19)$$

Using Lemmas C.1 and C.2, we demonstrate in Lemma C.3 that this potential function decreases in expectation at each iteration (proof provided in Appendix C.4).

**Lemma C.3.** *Under Assumptions A.1 and A.2, if the learning rate is chosen as*

$$\lambda_t \triangleq \lambda \le \min\left(\frac{1}{(1 + 2\alpha C_4 + 4\beta(1+\omega) + 2\alpha C_3/\lambda^2)L}, \frac{\sqrt{\beta K}}{\sqrt{1 + 2\alpha C_4 + 4\beta(1+\omega)}(1+\omega)L}\right), \quad (20)$$

*where $\alpha = \frac{3\beta C_1}{2(1+\omega)\theta L^2}$ for any $\beta > 0$, and the shift step size $\gamma_t$ is defined as in Lemma C.2, it follows that for every round $t \ge 0$, the expected potential function satisfies the following bound:*

$$\mathbb{E}_t[\Phi_{t+1}] \le \Phi_t - \frac{\lambda_t}{2}\|\nabla f(\mathbf{x}^t)\|^2 + \frac{3\beta C_2}{2(1+\omega)L}. \quad (21)$$

*Remark C.4.* Since the last term is generally a small constant during time (see Assumption A.2) and $\frac{\lambda_t}{2}\|\nabla f(\mathbf{x}^t)\|^2$ is positive, Equation equation 21 indicates that the potential decrease over the time.

With Lemma C.3 established, we now proceed to the proof of Theorem 3.6, which characterizes the utility and the number of communication rounds required for ERIS to reach a given accuracy level. We begin by summing Equation equation 21 from rounds $t = 0$ to $T - 1$:

$$\sum_{t=0}^{T-1}\mathbb{E}[\Phi_{t+1}] \le \sum_{t=0}^{T-1}\mathbb{E}[\Phi_t] - \sum_{t=0}^{T-1}\left(\frac{\lambda_t}{2}\|\nabla f(\mathbf{x}^t)\|^2 + \frac{3\beta C_2}{2(1+\omega)L}\right)$$

$$\mathbb{E}[\Phi_T] - \mathbb{E}[\Phi_0] \le -\sum_{t=0}^{T-1}\frac{\lambda_t}{2}\|\nabla f(\mathbf{x}^t)\|^2 + \frac{3\beta C_2 T}{2(1+\omega)L}$$

Since by construction, we typically have $\mathbb{E}[\Phi_t] \geq 0$, by choosing the learning rate $\lambda_t$ as in Lemma C.3, we finally obtain

$$\frac{1}{T}\sum_{t=0}^{T-1}\|\nabla f(\mathbf{x}^t)\|^2 \leq \frac{2\Phi_0}{\lambda T} + \frac{3\beta C_2}{(1+\omega)L\lambda}, \tag{22}$$

which proves that per $T \to \infty$

$$\lim_{T\to\infty}\frac{1}{T}\sum_{t=0}^{T-1}\|\nabla f(\mathbf{x}^t)\|^2 \leq \frac{3\beta C_2}{(1+\omega)L\lambda}. \tag{23}$$

While to achieve a predefined utility level $\epsilon \geq \frac{1}{T}\sum_{t=0}^{T-1}\|\nabla f(\mathbf{x}^t)\|^2$, the total rounds $T$ must satisfy:

$$T \geq \frac{2\,\Phi_0}{\lambda\left(\epsilon - \frac{3\beta\,C_2}{(1+\omega)\,L\,\lambda}\right)}. \tag{24}$$

If $\epsilon$ is strictly less than the residual $\frac{3\beta\,C_2}{(1+\omega)\,L\,\lambda}$, no finite $T$ can achieve the utility $\epsilon$ in an average sense, therefore conditions on the adopted estimator need to be changed.

*Remark* C.5. Eris utility is asyntotically governed by the variance in g̃, which directly depends on the used estimator (e.g., $C_2 = 0$ with SVRG/SAGA) or on the dimension of the batch size (e.g., $C_2 = 0$ with local full gradients). Compared to SoteriaFL (Li et al., 2022d), the upper bound of ERIS utility does not have a component growing with $T$, limiting the convergence.

$\qquad\square$

**Corollary C.6** (Utility of ERIS–SGD). *Consider the FL setting in equation 1 with $K$ clients, where each client $k \in \mathcal{K}$ holds a local dataset $D_k = \{d_{k,s}\}_{s=1}^{S_k}$, and let Assumptions 3.1 and 3.2 hold. Assume that, at each round $t$, client $k$ uses a mini-batch SGD estimator*

$$\tilde{\mathbf{g}}_k^t = \frac{1}{b_k}\sum_{s\in\mathcal{B}_k^t}\nabla f_{k,s}(\mathbf{x}^t),$$

*where $\mathcal{B}_k^t \subseteq \{1,\ldots,S_k\}$ is a uniformly sampled mini-batch of size $b_k$, and that stochastic gradients are uniformly bounded as $\|\nabla f_{k,s}(\mathbf{x})\| \leq G$ for all $k, s, \mathbf{x}$.*

*For notational simplicity, suppose that all clients share the same dataset size and batch size, i.e., $S_k \equiv m$ and $b_k \equiv b$ for all $k$. Then the constants in Assumption 3.2 are*

$$C_1 = C_3 = C_4 = 0, \qquad C_2 = \frac{(m-b)G^2}{mb}, \qquad \theta = 1.$$

*Let the compression operators $\mathcal{C}_k^t$ satisfy Definition 3.4 with parameter $\omega \geq 0$, and run ERIS with constant learning rate $\lambda_t \equiv \lambda$ and shift stepsize $\gamma_t \equiv \gamma$ as in Theorem 3.6, with*

$$\lambda \leq \frac{1}{(1+4\beta(1+\omega))L}, \qquad \gamma = \sqrt{\frac{1+2\omega}{2(1+\omega)^3}}, \tag{25}$$

*for some fixed $\beta > 0$. Then ERIS–SGD satisfies*

$$\frac{1}{T}\sum_{t=0}^{T-1}\mathbb{E}\|\nabla f(\mathbf{x}^t)\|^2 \leq \frac{2\Phi_0}{\lambda T} + \frac{3\beta(m-b)G^2}{(1+\omega)L\,\lambda\,mb}, \tag{26}$$

*where*

$$\Phi_0 := f(\mathbf{x}^0) - f^* + \alpha L\Delta^0 + \frac{\beta}{KL}\sum_{k=1}^K\|\nabla f_k(\mathbf{x}^0) - \mathbf{s}_k^0\|^2,$$

*with $\alpha = \frac{3\beta C_1}{2(1+\omega)L^2\theta}$, $\Delta^t := \frac{1}{K}\sum_{k=1}^K\|\mathbf{s}_k^t - \nabla f_k(\mathbf{x}^t)\|^2$, and the stepsize $\lambda$ is constrained by Theorem 3.6 as*

$$\lambda \leq \lambda_{\max} := \min\{\lambda_1,\lambda_2\}, \quad \lambda_1 = \frac{\sqrt{\beta K}}{\sqrt{1+4\beta(1+\omega)}(1+\omega)L}, \quad \lambda_2 = \frac{1}{(1+4\beta(1+\omega))L}.$$

*For any fixed $K, \omega$ we can choose $\beta > 0$ sufficiently small so that $\lambda_1 \leq \lambda_2$, and hence $\lambda_{\max} = \lambda_1$. We then set $\lambda = \lambda_1$ and let $T \to \infty$, so that the term $\frac{2\Phi_0}{\lambda T}$ vanishes. Substituting $\lambda_1$ into equation 26 yields*

$$\frac{1}{T}\sum_{t=0}^{T-1}\mathbb{E}\big\|\nabla f(\mathbf{x}^t)\big\|^2 \leq \frac{3\beta(m-b)G^2}{(1+\omega)L\,mb}\cdot\frac{\sqrt{1+4\beta(1+\omega)}(1+\omega)L}{\sqrt{\beta K}}, \tag{27}$$

$$\leq \frac{3\sqrt{\beta}(m-b)G^2}{mb}\frac{\sqrt{1+4\beta(1+\omega)}}{\sqrt{K}}. \tag{28}$$

*Since $\beta$ is a fixed constant and $\sqrt{1+4\beta(1+\omega)} = \Theta(\sqrt{1+\omega})$, we obtain the asymptotic bound*

$$\frac{1}{T}\sum_{t=0}^{T-1}\mathbb{E}\big\|\nabla f(\mathbf{x}^t)\big\|^2 = \mathcal{O}\bigg(\frac{(m-b)G^2}{mb}\frac{\sqrt{1+\omega}}{\sqrt{K}}\bigg). \tag{29}$$

*Finally, for $b = \Theta(m)$, we can remove the explicit dependence on $b$ and write*

$$\frac{1}{T}\sum_{t=0}^{T-1}\mathbb{E}\big\|\nabla f(\mathbf{x}^t)\big\|^2 = \mathcal{O}\bigg(\frac{G^2\sqrt{1+\omega}}{\sqrt{K}m}\bigg). \tag{30}$$

*Consequently, in this regime the stationarity error of* ERIS–SGD *is controlled solely by the variance of the local mini-batch gradients and grows with the compression variance $(1+\omega)$ while decreasing with the square root of the number of clients $K$ and the amount of local data $m$.*

## C.2  PROOF OF LEMMA C.1

*Proof.* By the definition of $\mathbf{v}^t$, we derive the following expression:

$$\mathbb{E}_t[\|\mathbf{v}^t\|^2] = \mathbb{E}_t\Bigg[\bigg\|\frac{1}{K}\sum_{k=1}^{K}\mathbf{s}_k^t + \frac{1}{K}\sum_{k=1}^{K}\mathcal{C}_k^t(\tilde{\mathbf{g}}_k^t - \mathbf{s}_k^t)\bigg\|^2\Bigg]$$

$$= \mathbb{E}_t\Bigg[\bigg\|\frac{1}{K}\sum_{k=1}^{K}\mathbf{s}_k^t + \frac{1}{K}\sum_{k=1}^{K}\mathcal{C}_k^t(\tilde{\mathbf{g}}_k^t - \mathbf{s}_k^t) + \frac{1}{K}\sum_{k=1}^{K}\tilde{\mathbf{g}}_k^t - \frac{1}{K}\sum_{k=1}^{K}\tilde{\mathbf{g}}_k^t\bigg\|^2\Bigg]$$

$$= \mathbb{E}_t\Bigg[\bigg\|\frac{1}{K}\sum_{k=1}^{K}\mathcal{C}_k^t(\tilde{\mathbf{g}}_k^t - \mathbf{s}_k^t) - \frac{1}{K}\sum_{k=1}^{K}(\tilde{\mathbf{g}}_k^t - \mathbf{s}_k^t) + \frac{1}{K}\sum_{k=1}^{K}\tilde{\mathbf{g}}_k^t\bigg\|^2\Bigg]$$

$$= \mathbb{E}_t\Bigg[\bigg\|\frac{1}{K}\sum_{k=1}^{K}\mathcal{C}_k^t(\tilde{\mathbf{g}}_k^t - \mathbf{s}_k^t) - \frac{1}{K}\sum_{k=1}^{K}(\tilde{\mathbf{g}}_k^t - \mathbf{s}_k^t)\bigg\|^2\Bigg] + \mathbb{E}_t\Bigg[\bigg\|\frac{1}{K}\sum_{k=1}^{K}\tilde{\mathbf{g}}_k^t\bigg\|^2\Bigg]$$

$$+ 2\bigg\langle\mathbb{E}_t\bigg[\frac{1}{K}\sum_{k=1}^{K}\mathcal{C}_k^t(\tilde{\mathbf{g}}_k^t - \mathbf{s}_k^t) - \frac{1}{K}\sum_{k=1}^{K}(\tilde{\mathbf{g}}_k^t - \mathbf{s}_k^t)\bigg], \mathbb{E}_t\bigg[\frac{1}{K}\sum_{k=1}^{K}\tilde{\mathbf{g}}_k^t\bigg]\bigg\rangle$$

$$\overset{(4)}{\leq} \mathbb{E}_t\Bigg[\frac{\omega}{K^2}\sum_{k=1}^{K}\|\tilde{\mathbf{g}}_k^t - \mathbf{s}_k^t\|^2\Bigg] + \mathbb{E}_t\Bigg[\bigg\|\frac{1}{K}\sum_{k=1}^{K}\tilde{\mathbf{g}}_k^t\bigg\|^2\Bigg], \tag{31}$$

where equation 31 follows because the cross term vanishes: the compression error has zero mean $(\mathbb{E}_t[\mathcal{C}_k^t(\cdot)] = \cdot)$, making their inner product zero in expectation.

Next, we establish upper bounds for each term in Equation equation 31.

• For the first term, we expand and decompose it as follows:

$$\mathbb{E}_t\left[\frac{\omega}{K^2}\sum_{k=1}^{K}\|\tilde{\mathbf{g}}_k^t - \mathbf{s}_k^t\|^2\right] = \mathbb{E}_t\left[\frac{\omega}{K^2}\sum_{k=1}^{K}\|(\tilde{\mathbf{g}}_k^t - \nabla f_k(\mathbf{x}^t)) + (\nabla f_k(\mathbf{x}^t) - \mathbf{s}_k^t)\|^2\right]$$

$$= \mathbb{E}_t\left[\frac{\omega}{K^2}\sum_{k=1}^{K}\left(\|\tilde{\mathbf{g}}_k^t - \nabla f_k(\mathbf{x}^t)\|^2 + \|\nabla f_k(\mathbf{x}^t) - \mathbf{s}_k^t\|^2\right.\right.$$

$$\left.\left. +2(\tilde{\mathbf{g}}_k^t - \nabla f_k(\mathbf{x}^t))^\top(\nabla f_k(\mathbf{x}^t) - \mathbf{s}_k^t))\right]\right. \tag{32}$$

$$= \mathbb{E}_t\left[\frac{\omega}{K^2}\sum_{k=1}^{K}\|\tilde{\mathbf{g}}_k^t - \nabla f_k(\mathbf{x}^t)\|^2\right]$$

$$+ \frac{\omega}{K^2}\sum_{k=1}^{K}\|\nabla f_k(\mathbf{x}^t) - \mathbf{s}_k^t\|^2, \tag{33}$$

where the last equality holds because the expectation of the cross-term vanishes due to the unbiased estimator assumption, i.e., $\mathbb{E}_t[\tilde{\mathbf{g}}_k^t] = \nabla f_k(\mathbf{x}^t)$, as specified in Assumption A.2.

• Similarly, for the second term, we proceed as follows:

$$\mathbb{E}^t\left[\left\|\frac{1}{K}\sum_{k=1}^{K}\tilde{\mathbf{g}}_k^t\right\|^2\right] = \mathbb{E}^t\left[\frac{1}{K^2}\sum_{k=1}^{K}\left\|(\tilde{\mathbf{g}}_k^t - \nabla f_k(\mathbf{x}^t)) + \nabla f_k(\mathbf{x}^t)\right\|^2\right]$$

$$= \mathbb{E}^t\left[\frac{1}{K^2}\sum_{k=1}^{K}\left(\|\tilde{\mathbf{g}}_k^t - \nabla f_k(\mathbf{x}^t)\|^2 + \|\nabla f_k(\mathbf{x}^t)\|^2\right.\right.$$

$$\left.\left. +2(\tilde{\mathbf{g}}_k^t - \nabla f_k(\mathbf{x}^t))^\top\nabla f_k(\mathbf{x}^t))\right]\right.$$

$$= \mathbb{E}^t\left[\frac{1}{K^2}\sum_{k=1}^{K}\|\tilde{\mathbf{g}}_k^t - \nabla f_k(\mathbf{x}^t)\|^2\right] + \|\nabla f(\mathbf{x}^t)\|^2, \tag{34}$$

The proof concludes by substituting equation 33 and equation 34 into equation 31. □

### C.3  PROOF OF LEMMA C.2

*Proof.* By the definition of the shift update $\mathbf{s}_k^{t+1} = \mathbf{s}_k^t + \gamma^t \mathcal{C}_k^t(\tilde{\mathbf{g}}_k^t - \mathbf{s}_k^t)$, we have:

$$\mathbb{E}_t\left[\frac{1}{K}\sum_{k=1}^{K}\|\nabla f_k(\mathbf{x}^{t+1}) - \mathbf{s}_k^{t+1}\|^2\right] = \mathbb{E}_t\left[\frac{1}{K}\sum_{k=1}^{K}\left\|\nabla f_k(\mathbf{x}^{t+1}) - \mathbf{s}_k^t - \gamma_t \mathcal{C}_k^t(\tilde{\mathbf{g}}_k^t - \mathbf{s}_k^t)\right\|^2\right]$$

$$= \mathbb{E}_t\left[\frac{1}{K}\sum_{k=1}^{K}\left\|\left(\nabla f_k(\mathbf{x}^{t+1}) - \nabla f_k(\mathbf{x}^t)\right) + \left(\nabla f_k(\mathbf{x}^t) - \mathbf{s}_k^t - \gamma_t \mathcal{C}_k^t(\tilde{\mathbf{g}}_k^t - \mathbf{s}_k^t)\right)\right\|^2\right]$$

$$\leq \mathbb{E}_t\left[\frac{1}{K}\sum_{k=1}^{K}\left((1+\frac{1}{\beta_t})\|\nabla f_k(\mathbf{x}^{t+1}) - \nabla f_k(\mathbf{x}^t)\|^2\right.\right.$$

$$\left.\left. +(1+\beta_t)\|\nabla f_k(\mathbf{x}^t) - \mathbf{s}_k^t - \gamma_t \mathcal{C}_k^t(\tilde{\mathbf{g}}_k^t - \mathbf{s}_k^t)\|^2\right)\right] \tag{35}$$

$$\overset{(8)}{\leq} \mathbb{E}_t\left[(1+\frac{1}{\beta_t})L^2\|\mathbf{x}^{t+1} - \mathbf{x}^t\|^2 + (1+\beta_t)\frac{1}{K}\sum_{k=1}^{K}\|\nabla f_k(\mathbf{x}^t) - \mathbf{s}_k^t - \gamma_t \mathcal{C}_k^t(\tilde{\mathbf{g}}_k^t - \mathbf{s}_k^t)\|^2\right], \tag{36}$$

where the Equation equation 35 is obtained from Young's inequality $\|\mathbf{a} + \mathbf{b}\|^2 \leq (1+\frac{1}{\beta})\|\mathbf{a}\|^2 + (1+\beta)\|\mathbf{b}\|^2$ with any $\beta_t > 0$.

To further bound the second term in equation 36, we expand the squared norm:

$$\mathbb{E}_t \left[ \frac{1}{K} \sum_{k=1}^{K} \left\| \nabla f_k(\mathbf{x}^t) - \mathbf{s}_k^t - \gamma_t \mathcal{C}_k^t(\tilde{\mathbf{g}}_k^t - \mathbf{s}_k^t) \right\|^2 \right]$$

$$= \mathbb{E}_t \left[ \frac{1}{K} \sum_{k=1}^{K} \left( \|\nabla f_k(\mathbf{x}^t) - \mathbf{s}_k^t\|^2 + \gamma_t^2 \|\mathcal{C}_k^t(\tilde{\mathbf{g}}_k^t - \mathbf{s}_k^t)\|^2 \right. \right.$$

$$\left. \left. - 2\gamma_t \langle \nabla f_k(\mathbf{x}^t) - \mathbf{s}_k^t, \mathcal{C}_k^t(\tilde{\mathbf{g}}_k^t - \mathbf{s}_k^t) \rangle \right) \right] \tag{37}$$

Since the expectation of the inner product term satisfies:

$$\mathbb{E}_t[\langle \nabla f_k(\mathbf{x}^t) - \mathbf{s}_k^t, \mathcal{C}_k^t(\tilde{\mathbf{g}}_k^t - \mathbf{s}_k^t) \rangle] = \mathbb{E}_t[||\nabla f_k(\mathbf{x}^t) - \mathbf{s}_k^t||^2],$$

Equation equation 37 simplifies to:

$$\mathbb{E}_t \left[ \frac{1}{K} \sum_{k=1}^{K} \left\| \nabla f_k(\mathbf{x}^t) - \mathbf{s}_k^t - \gamma_t \mathcal{C}_k^t(\tilde{\mathbf{g}}_k^t - \mathbf{s}_k^t) \right\|^2 \right]$$

$$= \mathbb{E}_t \left[ \frac{1}{K} \sum_{k=1}^{K} \left( (1 - 2\gamma_t)\|\nabla f_k(\mathbf{x}^t) - \mathbf{s}_k^t\|^2 + \gamma_t^2 \|\mathcal{C}_k^t(\tilde{\mathbf{g}}_k^t - \mathbf{s}_k^t)\|^2 \right) \right] \tag{38}$$

Then, applying Definition 3.4 to the term $(\tilde{\mathbf{g}}_k^t - \mathbf{s}_k^t)$, we derive the following inequality:

$$\mathbb{E}_t \left[ \|\mathcal{C}_k^t(\tilde{\mathbf{g}}_k^t - \mathbf{s}_k^t)\|^2 \right] = \mathbb{E}_t \left[ \left\| (\tilde{\mathbf{g}}_k^t - \mathbf{s}_k^t) + (\mathcal{C}_k^t(\tilde{\mathbf{g}}_k^t - \mathbf{s}_k^t) - (\tilde{\mathbf{g}}_k^t - \mathbf{s}_k^t)) \right\|^2 \right]$$

$$= \mathbb{E}_t \left[ \|\tilde{\mathbf{g}}_k^t - \mathbf{s}_k^t\|^2 \right] + 2\mathbb{E}_t \left[ \langle \tilde{\mathbf{g}}_k^t - \mathbf{s}_k^t, \mathcal{C}_k^t(\tilde{\mathbf{g}}_k^t - \mathbf{s}_k^t) - (\tilde{\mathbf{g}}_k^t - \mathbf{s}_k^t) \rangle \right]$$

$$+ \mathbb{E}_t \left[ \left\| \mathcal{C}_k^t(\tilde{\mathbf{g}}_k^t - \mathbf{s}_k^t) - (\tilde{\mathbf{g}}_k^t - \mathbf{s}_k^t) \right\|^2 \right]$$

$$= \mathbb{E}_t \left[ \|\tilde{\mathbf{g}}_k^t - \mathbf{s}_k^t\|^2 \right] + \mathbb{E}_t \left[ \left\| \mathcal{C}_k^t(\tilde{\mathbf{g}}_k^t - \mathbf{s}_k^t) - (\tilde{\mathbf{g}}_k^t - \mathbf{s}_k^t) \right\|^2 \right]$$

$$\leq (1 + \omega) \mathbb{E}_t \left[ \|\tilde{\mathbf{g}}_k^t - \mathbf{s}_k^t\|^2 \right]. \tag{39}$$

Substituting Equation equation 39 into equation 38, we simplify the second term as follows:

$$\mathbb{E}_t \left[ \frac{1}{K} \sum_{k=1}^{K} \left\| \nabla f_k(\mathbf{x}^t) - \mathbf{s}_k^t - \gamma_t \mathcal{C}_k^t(\tilde{\mathbf{g}}_k^t - \mathbf{s}_k^t) \right\|^2 \right]$$

$$\leq \mathbb{E}_t \left[ \frac{1}{K} \sum_{k=1}^{K} \left( (1 - 2\gamma_t)\|\nabla f_k(\mathbf{x}^t) - \mathbf{s}_k^t\|^2 + \gamma_t^2(1 + \omega)\|\tilde{\mathbf{g}}_k^t - \mathbf{s}_k^t\|^2 \right) \right]$$

$$\stackrel{(33)}{=} \mathbb{E}_t \left[ \frac{1}{K} \sum_{k=1}^{K} \left( (1 - 2\gamma_t + \gamma_t^2(1 + \omega))\|\nabla f_k(\mathbf{x}^t) - \mathbf{s}_k^t\|^2 \right. \right.$$

$$\left. \left. + \gamma_t^2(1 + \omega)\|\tilde{\mathbf{g}}_k^t - \nabla f_k(\mathbf{x}^t)\|^2 \right) \right]. \tag{40}$$

By plugging equation 40 into equation 36, we obtain:

$$\mathbb{E}_t \left[ \frac{1}{K} \sum_{k=1}^{K} \|\nabla f_k(\mathbf{x}^{t+1}) - \mathbf{s}_k^{t+1}\|^2 \right] \leq \mathbb{E}_t \left[ (1 + \frac{1}{\beta_t}) L^2 \|\mathbf{x}^{t+1} - \mathbf{x}^t\|^2 \right.$$

$$+ (1 + \beta_t) \frac{1}{K} \sum_{k=1}^{K} (1 - 2\gamma_t + \gamma_t^2(1+\omega)) \|\nabla f_k(\mathbf{x}^t) - \mathbf{s}_k^t\|^2$$

$$\left. + (1 + \beta_t) \frac{1}{K} \sum_{k=1}^{K} \gamma_t^2(1+\omega) \|\tilde{\mathbf{g}}_k^t - \nabla f_k(\mathbf{x}^t)\|^2 \right] \qquad (41)$$

Finally, setting $\beta_t = \frac{1}{1+2\omega}$, and $\gamma_t = \sqrt{\frac{1+2\omega}{2(1+\omega)^3}}$, we approximate the second term in Equation equation 41 with the following upper bound:

$$(1+\beta_t)(1 - 2\gamma_t + \gamma_t^2(1+\omega)) = \frac{2(1+\omega)}{(1+2\omega)} \left( 1 - 2\sqrt{\frac{1+2\omega}{2(1+\omega)^3}} + \frac{(1+2\omega)}{2(1+\omega)^2} \right)$$

$$\leq 1 - \frac{1}{2(1+\omega)} \qquad \forall \omega \geq 0.$$

Substituting this bound into Equation equation 41, we obtain Equation equation 18, thereby completing the proof of Lemma C.2. $\qquad\square$

## C.4 Proof of Lemma C.3

*Proof.* Given the definition $\mathcal{S}^t := \frac{1}{K} \sum_{k=1}^{K} \|\nabla f_k(\mathbf{x}^t) - \mathbf{s}_k^t\|^2$, we can derive a recursive bound for $\mathcal{S}^{t+1}$ using Lemma C.2:

$$\mathbb{E}_t \left[ \mathcal{S}^{t+1} \right] \leq \mathbb{E}_t \left[ \left( 1 - \frac{1}{2(1+\omega)} \right) \mathcal{S}^t + \frac{1}{(1+\omega)K} \sum_{k=1}^{K} \|\tilde{\mathbf{g}}_k^t - \nabla f_k(\mathbf{x}^t)\|^2 + 2(1+\omega)L^2 \|\mathbf{x}^{t+1} - \mathbf{x}^t\|^2 \right]$$

$$\overset{(11a)}{\leq} \mathbb{E}_t \left[ \left( 1 - \frac{1}{2(1+\omega)} \right) \mathcal{S}^t + \frac{C_1 \Delta^t + C_2}{(1+\omega)} + 2(1+\omega)L^2 \|\mathbf{x}^{t+1} - \mathbf{x}^t\|^2 \right] \qquad (42)$$

We now use Equation equation 42 along with equation 16 to bound the potential function $\Phi_{t+1}$, as defined in Equation equation 19:

$$\mathbb{E}_t[\Phi_{t+1}] := \mathbb{E}_t\left[f(\mathbf{x}^{t+1}) - f^* + \alpha L\Delta^{t+1} + \frac{\beta}{L}\mathcal{S}^{t+1}\right]$$

$$\leq \mathbb{E}_t\left[f(\mathbf{x}^t) - \lambda_t||\nabla f(\mathbf{x}^t)||^2 + \frac{\lambda^2 L}{2}||\mathbf{v}^t||^2 - f^* + \alpha L\Delta^{t+1}\right.$$

$$\left. + \frac{\beta}{L}\left(\left(1 - \frac{1}{2(1+\omega)}\right)\mathcal{S}^t + \frac{C_1\Delta^t + C_2}{(1+\omega)} + 2(1+\omega)L^2||\mathbf{x}^{t+1} - \mathbf{x}^t||^2\right)\right]$$

$$\overset{(11b)}{\leq} \mathbb{E}_t\left[f(\mathbf{x}^t) - f^* - \lambda_t||\nabla f(\mathbf{x}^t)||^2 + \frac{\lambda^2 L}{2}||\mathbf{v}^t||^2\right.$$

$$+ \alpha L\left((1-\theta)\Delta^t + C_3||\nabla f(\mathbf{x}^t)||^2 + C_4||\mathbf{x}^{t+1} - \mathbf{x}^t||^2\right)$$

$$\left. + \frac{\beta}{L}\left(\left(1 - \frac{1}{2(1+\omega)}\right)\mathcal{S}^t + \frac{C_1\Delta^t + C_2}{(1+\omega)} + 2(1+\omega)L^2||\mathbf{x}^{t+1} - \mathbf{x}^t||^2\right)\right]$$

$$= \mathbb{E}_t\left[f(\mathbf{x}^t) - f^* - \lambda_t||\nabla f(\mathbf{x}^t)||^2 + \left(\frac{1}{2} + \alpha C_4 + \beta(1+\omega)\right)L\lambda_t^2||\mathbf{v}^t||^2\right.$$

$$+ \alpha L\left((1-\theta)\Delta^t + C_3||\nabla f(\mathbf{x}^t)||^2\right)$$

$$\left. + \frac{\beta}{L}\left(\left(1 - \frac{1}{2(1+\omega)}\right)\mathcal{S}^t + \frac{C_1\Delta^t + C_2}{(1+\omega)}\right)\right] \tag{43}$$

where the last equality follows the adopted update rule $\mathbf{x}^{t+1} = \mathbf{x}^t - \lambda_t\mathbf{v}^t$. Now adopting $\mathcal{S}^t$ into the the definition of $\mathbb{E}_t[||\mathbf{v}^t||^2]$ provided in Lemma C.1, we obtain:

$$\mathbb{E}_t[||\mathbf{v}^t||^2] \leq \mathbb{E}_t\left[\frac{(1+\omega)}{K^2}\sum_{k=1}^{K}||\tilde{\mathbf{g}}_k^t - \nabla f_k(\mathbf{x}^t)||^2 + \frac{\omega}{K}\mathcal{S}^t + ||\nabla f(\mathbf{x}^t)||^2\right]$$

$$\overset{(11a)}{\leq} \mathbb{E}_t\left[\frac{(1+\omega)}{K}(C_1\Delta^t + C_2) + \frac{\omega}{K}\mathcal{S}^t + ||\nabla f(\mathbf{x}^t)||^2\right] \tag{44}$$

Subsituting Equation equation 44 into equation 43, we obtain as follows:

$$\mathbb{E}_t[\Phi_{t+1}] \leq f(\mathbf{x}^{t+1}) - f^*$$

$$+ \left[\left(\frac{1}{2} + \alpha C_4 + 2\beta(1+\omega)\right)\frac{(1+\omega)C_1\lambda_t^2}{K} + \alpha(1-\omega) + \frac{\beta C_1}{(1+\omega)L^2}\right]L\Delta^t$$

$$+ \left[\left(\frac{1}{2} + \alpha C_4 + 2\beta(1+\omega)\right)\frac{\omega L^2\lambda_t^2}{K} + \beta\left(1 - \frac{1}{2(1+\omega)}\right)\right]\frac{\mathcal{S}^t}{L}$$

$$- \left[\lambda_t - \left(\frac{1}{2} + \alpha C_4 + 2\beta(1+\omega)\right)L\lambda_t^2 - \alpha LC_3\right]||\nabla f(x^t)||^2$$

$$+ \left[\left(\frac{1}{2} + \alpha C_4 + 2\beta(1+\omega)\right)\frac{(1+\omega)L\lambda_t^2}{K} + \frac{\beta}{(1+\omega)L}\right]C_2 \tag{45}$$

To ensure that the right-hand side of Equation equation 45 remains consistent with the potential function $\Phi_t := f(\mathbf{x}^t) - f^* + \alpha L\Delta^t + \frac{\beta}{L}\mathcal{S}_t$, we select the parameters $\alpha$, $\beta$, and $\lambda_t$ to satisfy the following constraints:

$$\left(\frac{1}{2} + \alpha C_4 + 2\beta(1+\omega)\right)\frac{(1+\omega)C_1\lambda_t^2}{K} + \alpha(1-\omega) + \frac{\beta C_1}{(1+\omega)L^2} \leq \alpha \tag{46}$$

| Algorithm | Privacy | Utility / Accuracy |
|---|---|---|
| Distributed DP-SRM (Wang et al., 2023) | $(\varepsilon, \delta)$-DP | $\tilde{\mathcal{O}}\left(\dfrac{\sqrt{n \log(1/\delta)}}{K m \varepsilon}\right)$ |
| SDM-DSGD (Zhang et al., 2020a) | $(\varepsilon, \delta)$-LDP | $\tilde{\mathcal{O}}\left(\dfrac{\sqrt{n \log(1/\delta)}}{\sqrt{K} m \varepsilon}\right)$ |
| Q-DPSGD-1 (Ding et al., 2021b) | $(\varepsilon, \delta)$-LDP | $\tilde{\mathcal{O}}\left(\dfrac{\left(\frac{\tilde{\nu}^2}{K} + \frac{1}{m}\right)^{2/3}\left(n \log(1/\delta)\right)^{1/3}}{m^{2/3} \varepsilon^{2/3}}\right)$ |
| CDP-SGD (Li et al., 2022d) | $(\varepsilon, \delta)$-LDP | $\tilde{\mathcal{O}}\left(\dfrac{\sqrt{(1+\omega)\, n \log(1/\delta)}}{\sqrt{K} m \varepsilon}\right)$ |
| SoteriaFL-SGD (Li et al., 2022d) | $(\varepsilon, \delta)$-LDP | $\tilde{\mathcal{O}}\left(\dfrac{\sqrt{(1+\omega)\, n \log(1/\delta)}}{\sqrt{K} m \varepsilon}\, (1 + \sqrt{\tau})\right)$ |
| ERIS-SGD (no DP) | — | $\tilde{\mathcal{O}}\left(\dfrac{\sqrt{1+\omega}}{\sqrt{K} m}\right)$ |

Table 3: Asymptotic utility / accuracy bounds (average squared gradient norm after $T$ rounds) for different (local) differentially-private FL algorithms for the nonconvex problem in equation 1, compared to the non-DP utility bound of ERIS-SGD. Here $K$ is the number of clients, $m$ the number of samples per client, $n$ the model dimension, $\omega$ the compressor variance parameter, and $(\varepsilon, \delta)$ the privacy parameters. All bounds hide absolute constants and, where standard, additional logarithmic factors. For SoteriaFL, $\tau := (1+\omega)^{3/2}/\sqrt{K}$. Note that smaller values of the bound correspond to better utility / accuracy.

$$\left(\frac{1}{2} + \alpha C_4 + 2\beta(1+\omega)\right) \frac{\omega L^2 \lambda_t^2}{K} + \beta\left(1 - \frac{1}{2(1+\omega)}\right) \le \beta \tag{47}$$

Although these are not the strictest possible bounds for a fair comparison with the utility results of SoteriaFL, we adopt the same choices for $\alpha$, $\beta$, and $\lambda_t$, ensuring they satisfy conditions equation 46 and equation 47:

$$\alpha \ge \frac{3\beta C_1}{2(1+\omega)L^2\theta} \qquad \forall \beta > 0 \tag{48}$$

$$\lambda_t \equiv \lambda \le \frac{\sqrt{\beta K}}{\sqrt{1 + 2\alpha C_4 + 4\beta(1+\omega)}(1+\omega)L} \tag{49}$$

Here, Equation equation 48 follows from the constraint in equation 46, while equation 49 ensures compatibility with the potential function definition in equation 19. Additionally, we impose a further bound on $\lambda_t$ to guarantee that the negative gradient squared term remains sufficiently large (i.e., $\ge \frac{\lambda_t}{2}\|\nabla f(\mathbf{x}^t)\|^2$), obtaining:

$$\lambda_t \equiv \lambda \le \frac{1}{(1 + 2\alpha C_4 + 4\beta(1+\omega) + 2\alpha C_3/\lambda^2)L} \tag{50}$$

Finally, substituting the conditions equation 48–equation 50 into Equation equation 45, we obtain:

$$\mathbb{E}_t[\Phi_{t+1}] \le \Phi_t - \frac{\lambda_t}{2}\|\nabla f(x^t)\|^2 + \frac{3\beta}{2(1+\omega)L}C_2 \tag{51}$$

The last term is obtained directly by applying the bound from Equation equation 50, completing the proof.

$\square$

### C.5 UTILITY COMPARISON

Table 3 summarizes the asymptotic utility/accuracy guarantees of existing differentially-private FL algorithms, most of them with communication compression, and compares them to our non-DP utility bound for ERIS-SGD. Distributed DP-SRM (Wang et al., 2023) provides a global $(\varepsilon, \delta)$-DP baseline without compression: its utility improves linearly in the number of clients $K$ and in the number of samples per client $m$, but it does not consider LDP and therefore is not directly comparable to the LDP-based protocols in the rest of the table.

SDM-DSGD (Zhang et al., 2020a) and Q-DPSGD-1 (Ding et al., 2021b) are early attempts to combine local DP with compressed communication. However, SDM-DSGD assumes random-$k$ sparsification and requires $1 + \omega \ll \log T$ (i.e., communicating at least $k \gtrsim n/\log T$ coordinates per round), and its bound hides logarithmic factors that grow faster than $(1 + \omega)$. Q-DPSGD-1 relies on a different compression assumption ($\mathbb{E}\big[\|\mathcal{C}(\mathbf{x}) - \mathbf{x}\|^2\big] \leq \tilde{\nu}^2$, with parameter $\tilde{\nu}^2$ playing a similar role to our $1 + \omega$) and incurs a strictly worse utility by a factor $T^{1/6}$ compared to later methods, as already observed in (Li et al., 2022d).

CDP-SGD (Li et al., 2022d) can be seen as a direct compressed analogue of DP-SGD: it achieves $(\varepsilon, \delta)$-LDP and a utility that degrades with $\sqrt{(1+\omega)n}/(\sqrt{K}\, m\varepsilon)$, but still requires $\mathcal{O}(m^2)$ communication rounds when the local dataset size $m$ is large. SoteriaFL-SGD/GD improves upon CDP-SGD via shifted compression: it preserves the same dependence on $(1 + \omega), K, m, n$ up to a mild factor $(1 + \sqrt{\tau})$, where $\tau = (1 + \omega)^{3/2}/\sqrt{K}$ becomes negligible as $K \gg (1 + \omega)^3$, while reducing the total communication to $\mathcal{O}(m)$ rounds.

In contrast, ERIS-SGD does not inject any differentially-private noise to ensure formal $(\varepsilon, \delta)$-DP (although standard LDP mechanisms can be applied on top of it as shown in Figure 4), and thus its bound cannot be directly compared in terms of privacy guarantees. Nevertheless, once privacy noise is removed, our analysis shows that ERIS achieves a dimension-free non-private utility bound that scales as $\tilde{\mathcal{O}}\big(\sqrt{1 + \omega}/(m\sqrt{K})\big)$, yielding faster convergence under the same optimization assumptions. Moreover, ERIS exhibits the same favorable dependence on the number of clients $K$ and on the compression variance $(1 + \omega)$ as SoteriaFL-style methods, while operating in a fully serverless, sharded architecture. Empirically (see Figure 4), when we add the *same* LDP mechanism to both methods, ERIS and SoteriaFL achieve comparable utility for a given $(\varepsilon, \delta)$, but ERIS requires less additional noise thanks to the inherent privacy amplification provided by its decentralised aggregation scheme. Empirically, when we add the same LDP mechanism to both methods, ERIS and SoteriaFL converge to essentially the same $(\varepsilon, \delta)$-DP utility bound (i.e., the same dependence on $(1 + \omega)$, $K$, $m$, and $n$). However, ERIS requires less injected noise to reach this regime, thanks to the privacy amplification inherent in its decentralized, sharded aggregation architecture.

## D PRIVACY GUARANTEES FOR ERIS

In this section, we present the detailed proof of Theorem 3.7, which establishes an upper bound on the information leakage incurred by ERIS under the honest-but-curious threat model. The analysis follows an information-theoretic approach by bounding the mutual information between a client's local dataset $D_k$ and the adversary's partial view of the transmitted model updates $\mathbf{v}_{k,(a)}^t = (\tilde{\mathbf{g}}_k^t - \mathbf{s}_k^t) \odot \mathbf{m}_{\mathcal{C}_k^t} \odot \mathbf{m}_{(a)}^t$ over $T$ communication rounds. We then extend the result to colluding adversaries, who may share observations to amplify their attack.

### D.1 PROOF OF THEOREM 3.7

*Proof.* For lighter notation, we first define a single combined mask $\mathbf{m}_k^t := \mathbf{m}_{\mathcal{C}_k^t} \odot \mathbf{m}_{(a)}^t$ to streamline notation and directly leverage its properties. Next, rather than working with $\tilde{\mathbf{g}}_k^t - \mathbf{s}_k^t$, we substitute the parameter vector $\mathbf{x}_k^{t+1}$. Because $\mathbf{x}_k^{t+1}$ is fully determined by $\mathbf{x}_k^t$ and $\tilde{\mathbf{g}}_k^t - \mathbf{s}_k^t$, i.e., $\mathbf{x}_k^{t+1} = \mathbf{x}_k^t + \lambda(\tilde{\mathbf{g}}_k^t - \mathbf{s}_k^t)$, it carries the same information in an information-theoretic sense. This allows us to simplify the derivations without affecting the validity of the privacy analysis. We denote, in the end, by $\mathcal{H}_t$ the full public transcript up to round $t$: it contains every masked update and model weight at each round up to $t$. More precisely,

$$\mathcal{H}_t := \sigma\Big( \big\{ \mathbf{x}_k^\ell \odot \mathbf{m}_k^\ell : \ \ell = 0, \ldots, t \big\} \cup \big\{ \mathbf{x}_k^\ell : \ \ell = 0, \ldots, t \big\} \Big).$$

$$I\big(\mathbf{D}_k; \{\mathbf{x}_k^{t+1} \odot \mathbf{m}_k^{t+1}\}_{t=0}^{T-1}\big) \overset{(a)}{=} \sum_{t=0}^{T-1} I\big(\mathbf{D}_k; \mathbf{x}_k^{t+1} \odot \mathbf{m}_k^{t+1} \mid \mathcal{H}_t\big)$$

$$\overset{(b)}{\leq} \sum_{t=0}^{T-1} I\big(\mathbf{D}_k; \mathbf{x}_k^{t+1} \odot \mathbf{m}_k^{t+1} \mid \mathcal{H}_t, \mathbf{m}_k^{t+1}\big)$$

$$= \sum_{t=0}^{T-1} \mathbb{E}\big[ I\big(\mathbf{D}_k; \mathbf{x}_k^{t+1} \odot \mathbf{m}_k^{t+1} \mid \mathcal{H}_t, \mathbf{m}_k^{t+1} = \mathbf{m}\big)\big]$$

Step (a) follows from the chain rule for mutual information and definition of $\mathcal{H}_t$, while step (b) follows from the identities:

$$I(U; V \mid H) = I(U; V, M \mid H) - I(U; M \mid H, V)$$

$$= \big[ I(U; M \mid H) + I(U; V \mid H, M) \big] - I(U; M \mid H, V)$$

$$= I(U; V \mid H, M) - I(U; M \mid H, V)$$

$$\leq I(U; V \mid H, M)$$

$$= I\big(\mathbf{D}_k; \mathbf{x}_k^{t+1} \odot \mathbf{m}_k^{t+1} \mid \mathcal{H}_t, \mathbf{m}_k^{t+1}\big).$$

where $U = \mathbf{D}_k$, $V = \mathbf{x}_k^{t+1} \odot \mathbf{m}_k^{t+1}$, $M = \mathbf{m}_k^{t+1}$, $H = \mathcal{H}_t$. Here, we used the independence of the mask ($I(U; M \mid H) = 0$), and the inequality follows from the nonnegativity of mutual information.

Finally, fix any mask realization $\mathbf{m}$, and let

$$S(\mathbf{m}) = \{\, i : m_i = 1 \}$$

denote the set of revealed coordinates. Then

$$I\big(\mathbf{D}_k; \mathbf{x}_k^{t+1} \odot \mathbf{m}_k^{t+1} \mid \mathcal{H}_t, \mathbf{m}_k^{t+1} = \mathbf{m}\big) = I\big(\mathbf{D}_k; \{\mathbf{x}_{k,i}^{t+1}\}_{i \in S(\mathbf{m})} \mid \mathcal{H}_t\big)$$

$$\leq \sum_{i \in S(\mathbf{m})} I\big(\mathbf{D}_k; \mathbf{x}_{k,i}^{t+1} \mid \mathcal{H}_t\big) \leq |S(\mathbf{m})| C_{\max},$$

where

$$C_{\max} := \max_{i, t, \mathcal{H}_t} I\big(\mathbf{D}_k; \mathbf{x}_{k,i}^{t+1} \mid \mathcal{H}_t\big).$$

Since each of the $n/A$ coordinates is retained with probability $p$, we have $\mathbb{E}[\|S(\mathbf{m})\|] = np/A$. Taking expectations gives

$$\mathbb{E}\big[ I\big(\mathbf{D}_k; \mathbf{x}_k^{t+1} \odot \mathbf{m}_k^{t+1} \mid \mathbf{m}_k^{t+1}, \mathcal{H}_t\big)\big] \leq \frac{n}{A} p\, C_{\max},$$

and summing over $t = 0, \ldots, T-1$ yields

$$I\big(\mathbf{D}_k; \{\mathbf{x}_k^{t+1} \odot \mathbf{m}_k^{t+1}\}_{t=0}^{T-1}\big) \leq T\, \frac{n}{A} p\, C_{\max}.$$

$$\square$$

*Remark* D.1. Assuming the individual model weights are distributed conditionally on $D_k$ and $\mathcal{H}_t$ as $\mathbf{x}_{k,i}^{t+1} \mid D_k, \mathcal{H}_t \sim \mathcal{N}(\mu(D_k), \sigma_{\text{cond}}^2)$, while $\mathbf{x}_{k,i}^{t+1} \mid \mathcal{H}_t \sim \mathcal{N}(\mu, \sigma^2)$. This allows us to use properties of differential entropy for Gaussian distributions. Thus, we have:

$$I\big(\mathbf{D}_k; \mathbf{x}_{k,i}^{t+1} \mid \mathcal{H}_t\big) = H(\mathbf{x}_{k,i}^{t+1} \mid \mathcal{H}_t) - H(\mathbf{x}_{k,i}^{t+1} \mid D_k, \mathcal{H}_t) \leq \frac{1}{2} \log\left(\frac{\sigma^2}{\sigma_{\text{cond}}^2}\right) = \frac{1}{2} \log\left(1 + \text{SNR}\right),$$

where the signal-to-noise ratio (SNR) is defined as $\text{SNR} = \frac{\sigma^2 - \sigma_{\text{cond}}^2}{\sigma_{\text{cond}}^2}$. Thus, from the above, it follows that $C_{\max} \leq \frac{1}{2} \log(1 + \text{SNR})$.

## D.2 PRIVACY UNDER COLLUDING AGGREGATORS

We now extend our analysis to a coalition of aggregators that share their shards before attempting the attack. Let $\mathcal{C} \subseteq \{1, \ldots, A\}$ denote the set of colluding aggregators with cardinality $A_c := |\mathcal{C}|$.

**Corollary D.2** (Colluding–aggregator privacy bound). *Assume the setting of Theorem 3.7. For every communication round $t \in \{1, \ldots, T\}$ let the union mask*

$$\mathbf{m}_{\mathrm{col}}^t := \bigvee_{a \in \mathcal{C}} \mathbf{m}_{(a)}^t \quad \left(\vee \text{ denotes the element-wise logical OR}\right)$$

*select the coordinates revealed to the colluding coalition. Define the coalition's view of client $k$ at round $t$ as*

$$\mathbf{v}_{k,\mathrm{col}}^t := (\tilde{\mathbf{g}}_k^t - \mathbf{s}_k^t) \odot \mathbf{m}_{\mathcal{C}_k^t} \odot \mathbf{m}_{\mathrm{col}}^t.$$

*Assuming that $\max_{i,t,\mathcal{H}_t} I\left(D_k; \mathbf{x}_{k,i}^{t+1} \mid \mathcal{H}_t\right) < \infty$ then, under the honest-but-curious threat model, the mutual information between the client's private dataset $D_k$ and the coalition's transcript over $T$ rounds satisfies*

$$I\left(D_k; \{\mathbf{v}_{k,\mathrm{col}}^t\}_{t=1}^T\right) \leq n\,T\,\frac{p A_c}{A}\,C_{\max},$$

*where $C_{\max}$ is exactly the per-coordinate mutual information bound given in Theorem 3.7.*

*Proof.* The extension to colluding parties follows exactly the same steps as in Appendix D, with a single modification: replace the per-shard mask $\mathbf{m}_{(a)}^t$ by the union mask

$$\mathbf{m}_{\mathrm{col}}^t = \bigvee_{a \in \mathcal{C}} \mathbf{m}_{(a)}^t,$$

where $\mathcal{C}$ is the set of colluding shards of size $A_c$. Since the original shards are pairwise disjoint, $\mathbf{m}_{\mathrm{col}}^t$ exposes exactly

$$\left|\mathbf{m}_{\mathrm{col}}^t\right| = A_c \frac{n}{A}$$

coordinates per round, and remains statistically independent of the corresponding values of $\mathbf{x}_k^{t+1}$.

Under collusion, the set $S$ of revelead coordinates simply enlarges is mean to $A_c\,n/A$ coordinates, while the retention probability $p$ remain unchanged. Hence the entire inner sum is multiplied by $A_c$.

Substituting this modification into the rest of the derivation yields

$$I\left(D_k; \{\mathbf{v}_{k,\mathrm{col}}^t\}_{t=0}^{T-1}\right) \leq T\,n\,\frac{p\,A_c}{A}\,C_{\max},$$

as claimed. In particular:

- If $A_c = 1$, this reduces to Theorem 3.7.

- If $A_c = A$, the sharding protection vanishes and the bound becomes $I \leq n\,T\,p\,C_{\max}$, governed solely by the compression mechanism.

$\square$

*Remark D.3.* Corollary D.2 shows that the privacy loss grows *linearly* with the coalition size $A_c$. Consequently, anticipating up to $A_c^{\max}$ colluding aggregators, one can retain the original leakage level of Theorem 3.7 by increasing the shard count to $A \mapsto A \cdot A_c^{\max}$ or, equivalently, by decreasing the retention probability to $p \mapsto p/A_c^{\max}$, thereby preserving the product $\frac{p A_c}{A}$.

**Empirical Validation** To assess the robustness of ERIS against coordinated leakage attempts, we evaluate how MIA accuracy evolves as multiple honest-but-curious clients collude by sharing their received shards. Figure 5 reports the resulting leakage curve. As the collusion group grows, MIA accuracy increases smoothly but remains consistently below the FEDAVG baseline and close to the minimum achievable leakage, even when 50% of clients collude. These results confirm that the shard-based decomposition in ERIS meaningfully amplifies privacy, limiting the adversary's advantage even under strong collusion scenarios.

Figure 5: Impact of honest-but-curious client collusion in ERIS.

## E  EXPERIMENTAL SETUP

This section provides additional details on the experimental configuration used throughout the paper, including model architectures, training protocols, and hardware resources. We also describe the software libraries, dataset licenses, and implementation details to ensure full reproducibility.

### E.1  MODELS AND HYPERPARAMETER SETTINGS

We use 5-fold cross-validation across all experiments, varying the random seed for both data generation and model initialization to ensure reproducibility. Each dataset is paired with an appropriate architecture: GPT-Neo (Black et al., 2021) (`EleutherAI/gpt-neo-1.3B`, 1.3B parameters) from HuggingFace for CNN/DailyMail, DistilBERT (Sanh et al., 2019) (`distilbert-base-uncased`, 67M parameters) for IMDB, ResNet-9 (He et al., 2016) (1.65M parameters) for CIFAR-10, and LeNet-5 (Lecun et al., 1998) (62K parameters) for MNIST. For both IID and non-IID settings, we use one local update per client per round (i.e., unbiased gradient estimator), except for GPT-Neo, where memory constraints require two local epochs with a batch size of 8. In the biased setting (multiple local updates per round), we use a batch size of 16 for IMDB and 64 for CIFAR-10 and MNIST under IID conditions. In all settings, each client reserves 30% of its local data for evaluation. To ensure fair comparison of communication costs—which directly depend on the number of rounds—we cap the total rounds for all baselines at the point where FedAvg converges, determined by the minimum validation loss (generally the first to converge). This results in 2-4 rounds for CNN/DailyMail, 14–22 for IMDB, 80–140 for CIFAR-10, and 120–250 for MNIST in the unbiased setting. In the biased setting (two local epochs per round), the ranges are 4–16 for IMDB, 60–140 for CIFAR-10, and 80–200 for MNIST. We use a learning rate of $5e-5$ for CNN/DailyMail and IMDB, and 0.01 for CIFAR-10 and MNIST. For optimization, we adopt Adam (Kingma & Ba, 2017) (with `weight_decay` $= 0.0$, $\beta_1 = 0.9$, $\beta_2 = 0.999$, and $\epsilon = 1e-8$) on CNN/DailyMail and IMDB, and SGD (Robbins & Monro, 1951) with momentum 0.9 for CIFAR-10 and MNIST. For experiments involving differential privacy, we use the Opacus library (Yousefpour et al., 2021).

### E.2  IMPLEMENTATION DETAILS OF PRIVACY ATTACKS

We evaluate privacy leakage under the standard *honest-but-curious* threat model, where an adversary (e.g., a compromised aggregator or server) can observe all transmitted model updates derived from each client's private dataset. We implement two widely studied categories of attacks: *Membership Inference Attacks (MIA)* (Shokri et al., 2017; Zari et al., 2021; Li et al., 2022a; Zhang et al., 2023b; He et al., 2024) and *Data Reconstruction Attacks (DRA)* (Hitaj et al., 2017; Zhang et al., 2020b; Ren et al., 2022; Zhao et al., 2020; Yin et al., 2021; Dimitrov et al., 2022; Zhang et al., 2023a).

*Membership Inference Attacks.* We adopt a distributed variant of the privacy auditing framework of Steinke et al. (2023). For each client, 50% of the local training samples are designated as *canary* samples, equally split between those included and excluded from training. After training, canaries are ranked by model confidence or gradient alignment; the top third are labeled as "in," the bottom third as "out," while the middle third are discarded to mitigate uncertainty bias. Evaluation is repeated on the same canary sets across all methods and folds of the cross-validation. To capture privacy leakage

throughout training, MIA accuracy is computed at each round and for each client; the reported score corresponds to the maximum, over all $T$ rounds, of the average accuracy across $K$ clients. This ensures comparability across methods with different convergence speeds.

*Data Reconstruction Attacks.* For DRA, we adopt the strongest white-box threat model, where the adversary is assumed to access the gradient of a single training sample. We implement three representative gradient inversion methods: DLG (Zhu et al., 2019), iDLG (Zhao et al., 2020), and ROG (Yue et al., 2023), the latter specifically tailored to reconstruct images from obfuscated gradients. All methods are evaluated on the same subset of 200 randomly sampled data points within each cross-validation fold to ensure fairness. Reconstruction quality is assessed with LPIPS, SSIM, and SNR, capturing perceptual similarity, structural fidelity, and signal-to-noise characteristics. Further algorithmic descriptions on each attack are provided in Appendix F.5.

### E.3 LICENSES AND HARDWARE

All experiments were implemented in Python 3.13 using open-source libraries: PyTorch 2.6 (Paszke et al., 2019) (BSD license), Flower 1.12 (Beutel et al., 2022) (Apache 2.0), Matplotlib 3.10 (Hunter, 2007) (BSD), Opacus 1.5 (Yousefpour et al., 2021) (Apache 2.0) and Pandas 2.2 (Wes McKinney, 2010) (BSD). We used publicly available datasets: MNIST (GNU license), CIFAR-10, IMDB (subject to IMDb's Terms of Use), and CNN/DailyMail (Apache license 2.0). The complete codebase and instructions for reproducing all experiments are available on GitHub[1] under the MIT license. Publicly available implementations were used to reimplement all baselines, except for FedAvg and Min. Leakage, which we implemented directly using Flower Library (Beutel et al., 2022). We follow the recommended hyperparameters for baselines, setting the compression ratio of SoteriaFL to 5% and the graph degree of Shatter to 4.

Experiments were run on a workstation with four NVIDIA RTX A6000 GPUs (48 GB each), dual AMD EPYC 7513 32-core CPUs, and 512 GB RAM.

## F ADDITIONAL EXPERIMENTS AND ANALYSIS

This section presents complementary experiments and empirical validations that reinforce the theoretical claims made in the main paper. We analyze the distributional properties of model weights to support the Gaussian condition in our privacy analysis, evaluate the scalability of ERIS through distribution time comparisons, and further assess its robustness against data reconstruction attacks. Additionally, we provide detailed utility–privacy trade-off results under both IID and non-IID settings, and with unbiased and biased gradient estimators across multiple datasets and training configurations.

### F.1 EMPIRICAL VALIDATION OF THE GAUSSIAN ASSUMPTION FOR MODEL WEIGHTS

Remark 3.8 gives a closed-form bound for $C_{\max}$ when each conditional weight $\mathbf{x}_{k,i}^{t+1} \mid D_k, \mathcal{H}_t$ and $\mathbf{x}_{k,i}^{t+1} \mid \mathcal{H}_t$ are (approximately) Gaussian. Verifying Gaussianity of the first case is the stricter—and therefore more informative—requirement. We thus track, for every client, the weights it uploads each round and examine these conditional distributions empirically. Figure 6 plots these conditional weight histograms for three representative models—DistilBERT on IMDB, ResNet-9 on CIFAR-10, and LeNet-5 on MNIST. Each 3-D panel shows weight value (x-axis), training round (depth), and frequency (z-axis). Across all datasets, the distributions consistently approximate a zero-mean Gaussian shape ($\sim \mathcal{N}(0, \sigma_{\text{cond}})$. Although the standard deviation slightly varies during training, it remains well below 0.2 throughout. This evidence supports the sub-Gaussian premise in Remark 3.8 and validates the constant $C_{\max}$ used in Theorem 3.7.

### F.2 SCALABILITY AND EFFICIENCY OF ERIS

To evaluate the scalability and communication efficiency of ERIS, we provide both a theoretical analysis of model distribution time and empirical comparisons with existing FL frameworks.

---

[1] https://github.com/

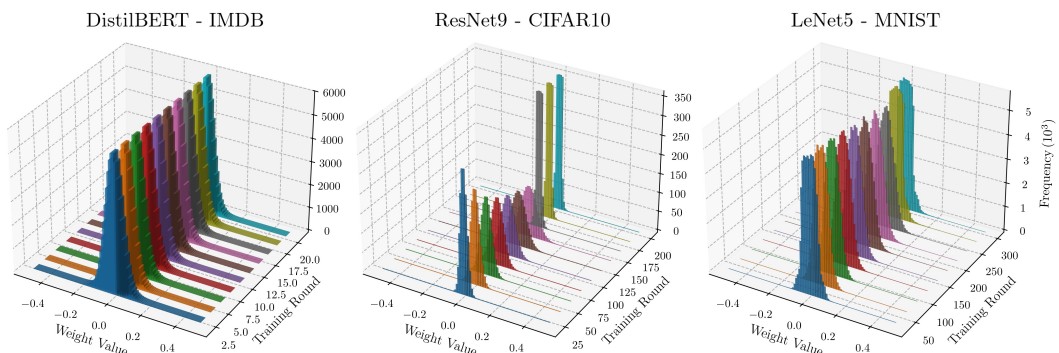

Figure 6: Conditional weight distributions $(\mathbf{x}_{k,i}^{t+1} \mid D_k, \mathcal{H}_t)$ over training rounds for DistilBERT, ResNet-9, and LeNet-5. Each 3D plot shows the distribution of weight values (horizontal axis) over time (depth axis), with frequency represented on the vertical axis. In all cases, the weight distributions remain $\sim \mathcal{N}(0, \sigma_{\text{cond}})$ with a $\sigma_{\text{cond}} < 0.2$, validating the sub-Gaussian premise used in used Remark 3.8.

### F.2.1 THEORETICAL ANALYSIS OF MODEL DISTRIBUTION TIME

We begin by quantifying the minimum time required to distribute models in a single training round under various FL setups. Here, the distribution time refers to the time needed for: (i) clients to transmit their local models to the aggregation parties (either a central server or a set of aggregators), and (ii) all clients to receive the updated global model. For clarity, we assume full client participation in each round; however, the same analysis readily extends to partial participation scenarios by adjusting the number of active clients accordingly.

**Single-server Federated Learning.** In traditional centralized FL, we consider a single server and $K$ clients. Let $u_s$ and $d_s$ denote the server's upload and download rates, and let $u_k$, $d_k$ be the $k$-th client's upload and download rates, respectively. Assume the model has $n$ parameters and each is represented as a 32-bit float, yielding a total model size of $b \approx 32 \cdot n$ bits. The distribution time in a single training round is governed by the following observations:

- The server must collect $K$ local models, each of size $b$ bits, resulting in a total inbound traffic of $K \cdot b$ bits, received at a download rate $d_s$.
- Each client $k$ uploads its local model at an individual rate $u_k$. The server cannot complete the upload phase until the slowest client—i.e., the one with the lowest $u_k$—has finished its transmission.
- Once all local models are received, the server performs aggregation and then broadcasts the aggregated global model back to all $K$ clients. This requires transmitting another $K \cdot b$ bits at the server's upload rate $u_s$.
- Model distribution concludes when every client has received the global model. This process is bounded by the client with the lowest download rate $d_k$, as it determines the last completed transfer.

Putting all these observations together, we derive the minimum distribution time in a single training round for a centralized FL setup without compression such as FedAvg, denoted by $D_{FedAvg}$.

$$D_{FedAvg} \geq \max\left\{ \frac{K \cdot b}{d_s}, \frac{b}{\min\{u_1, \ldots, u_K\}} \right\} + \max\left\{ \frac{K \cdot b}{u_s}, \frac{b}{\min\{d_1, \ldots, d_K\}} \right\} \quad (52)$$

Here, the first term captures the server's time to receive all local model uploads and the slowest client's upload time, while the second term captures the server's model broadcast time and the slowest client's download time.

To reduce distribution time, several FL methods focus on minimizing the volume of transmitted data per round—i.e., decreasing the effective model size $b$. For example, *PriPrune* (Chu et al., 2024) applies structured pruning to eliminate a fraction $p$ of the model's parameters before transmission.

This reduces the transmitted size to $b' \leq 32 \cdot (1-p) \cdot n$ bits. Similarly, *SoteriaFL* (Li et al., 2022d) compresses gradients using a shifting operator controlled by a compression factor $\omega$, leading to a model size bounded by $b' \leq 32 \cdot \frac{1}{\omega+1} \cdot n$.

**Serverless Federated Learning.** We now extend the analysis to ERIS, our proposed serverless FL framework with $K$ clients and $A \leq K$ aggregators. In contrast to centralized schemes, ERIS decentralizes aggregation across $A$ aggregators (a subset of clients) and compresses model updates prior to transmission. We denote the size of the compressed model by $b' \leq 32 \cdot \frac{1}{\omega+1} \cdot n$. To estimate the minimum model distribution time in a single training round under ERIS, we consider the following:

- Each aggregator must collect $K-1$ model shards from the clients (excluding its own), amounting to a total of $(K-1) \cdot \frac{b'}{A}$ bits received per aggregator at a download rate $d_k$. The aggregation process cannot proceed before the slowest aggregator (i.e., the one with the lowest download rate) receives all required shards.
- Each client $k$ uploads one shard of its model to each aggregator, sending a total of $b'$ bits. If the client is not serving as an aggregator (worst case), it must upload the entire set of $A$ shards at an upload rate $u_k$. The aggregation step is gated by the client with the lowest upload rate.
- Once aggregation is complete, each aggregator redistributes its shard of the updated model to all $K$ clients. This amounts to sending $(K-1) \cdot \frac{b'}{A}$ bits at an upload rate $u_k$. Model dissemination is constrained by the aggregator with the slowest upload speed.
- Full model reconstruction occurs only after each client receives one shard from every aggregator. In the worst-case scenario, a non-aggregator client must download the complete model (i.e., $b'$ bits) at a rate $d_k$. The distribution concludes when the client with the lowest download rate completes this transfer.

Putting all these observations together, we derive the minimum distribution time in a single training round for ERIS, denoted by $D_{\text{ERIS}}$:

$$
D_{\text{ERIS}} \geq \max \left\{ \frac{(K-1)b'}{A \cdot \min\{d_1, \ldots, d_A\}}, \frac{b'}{\min\{u_1, \ldots, u_K\}} \right\}
$$
$$
+ \max \left\{ \frac{(K-1)b'}{A \cdot \min\{u_1, \ldots, u_A\}}, \frac{b'}{\min\{d_1, \ldots, d_K\}} \right\} \tag{53}
$$

While ERIS leverages decentralized aggregation to reduce communication overhead, it is not the first framework to exploit distributed training. For instance, *Ako* (Watcharapichat et al., 2016) distributes gradient computations by splitting each model into $v$ disjoint partitions and randomly assigning them to worker nodes. However, this approach differs fundamentally from ERIS: in Ako, not all clients receive the full model in each round, which may hinder convergence to the standard FedAvg solution. Furthermore, in each round, a client uploads and receives $K$ partitions—equivalent to the full model size—resulting in substantial bandwidth usage. We can estimate the minimum distribution time for Ako, denoted by $D_{Ako}$, using a similar worst-case analysis:

$$
D_{Ako} \geq \max \left\{ \frac{b}{\min\{d_1, \ldots, d_K\}}, \frac{b}{\min\{u_1, \ldots, u_K\}} \right\} \tag{54}
$$

*Shatter* (Biswas et al., 2025) is also a privacy preserving distributed learning framework. In shatter, each round consists of three steps. In the first step, each node updates its local model and divides the result in $l$ chunks. Each client (real node) runs $l$ virtual nodes. The virtual nodes form an overlay network over which model parameter updates are multicast with a gossiping protocol. Once received $r$ updates for each of the $l$ virtual nodes running on a real node, the last step consists of the virtual nodes forwarding the received updates to the real node to perform the aggregation. Notice that in this setup, not all clients will receive all model updates, so the communication overhead is reduced at the cost of slower global model convergence. The model distribution occurs in the second step through a multicast among the virtual nodes. The model distribution cannot finish before each real node (via its virtual nodes) has finished uploading its model chunks, and has finished downloading model updates from $r$ other clients. Then, we need to account for the time to upload all the model updates with the total upload capacity being the sum of all individual node upload rates. Therefore, we can estimate the minimum distribution time for shatter, denoted by $D_{\text{shatter}}$:

$$D_{\text{shatter}} \geq \max \left\{ \frac{b}{\min\{u_1, \ldots, u_K\}}, \frac{r \cdot b}{\min\{d_1, \ldots, d_K\}}, \frac{r \cdot b}{\sum_{i=1}^{K} u_i} \right\} \tag{55}$$

### F.2.2 NUMERICAL RESULTS

**Effect of Number of Clients and Model Size on Distribution Time.**  Figure 7 compares the minimum distribution time per training round for ERIS and other FL frameworks under varying numbers of aggregators and model sizes. We assume homogeneous network conditions across all nodes, with upload and download rates fixed at 100 Mbps. For the baselines, we apply a pruning rate of 0.3 for PriPrune, a compression ratio of $1/(\omega + 1) = 0.05$ for SoteriaFL and ERIS, and the overlay topology (i.e., graph degree) for shatter forms a 4-regular graph. Note that $1/(\omega + 1) = 0.05$ (with $\omega = 18$) corresponds to the least aggressive compression used in our experiments, chosen for MNIST to preserve model utility. In other settings, such as IMDB, we adopt much stronger compression (e.g., $1/(\omega + 1) = 0.00012$), further lowering communication overhead without harming performance. Thus, the results in Figure 7 represent a conservative estimate of ERIS's efficiency.

On the left, Figure 7 illustrates how distribution time scales with the number of participating clients: while all methods experience linear growth (except Ako and Shatter, which always exchange with a fixed number of neighbours), ERIS benefits from increased decentralisation—achieving significantly lower distribution times as the number of aggregators $A$ increases. In the worst-case configuration with $A = 2$, ERIS still achieves a $34\times$ speedup over FedAvg and a $2\times$ improvement over SoteriaFL. When $A = 50$, these gains rise dramatically to $1020\times$ and $51\times$, respectively, underscoring the scalability advantages of decentralised aggregation. Maximum efficiency is achieved when the number of aggregators matches the number of clients ($A = K$), maximising parallelism and evenly distributing the communication load. Notably, Ako and Shatter remain constant with respect to the number of clients, as its communication pattern does not involve distributing the full model to all participants—at the expense of model consistency and convergence guarantees. On the right, Figure 7 examines the impact of increasing model size with 50 clients. The results highlight the communication efficiency of decentralised approaches, especially ERIS, which outperforms traditional centralised frameworks as model size grows. This confirms the practical benefits of combining decentralised aggregation with compression.

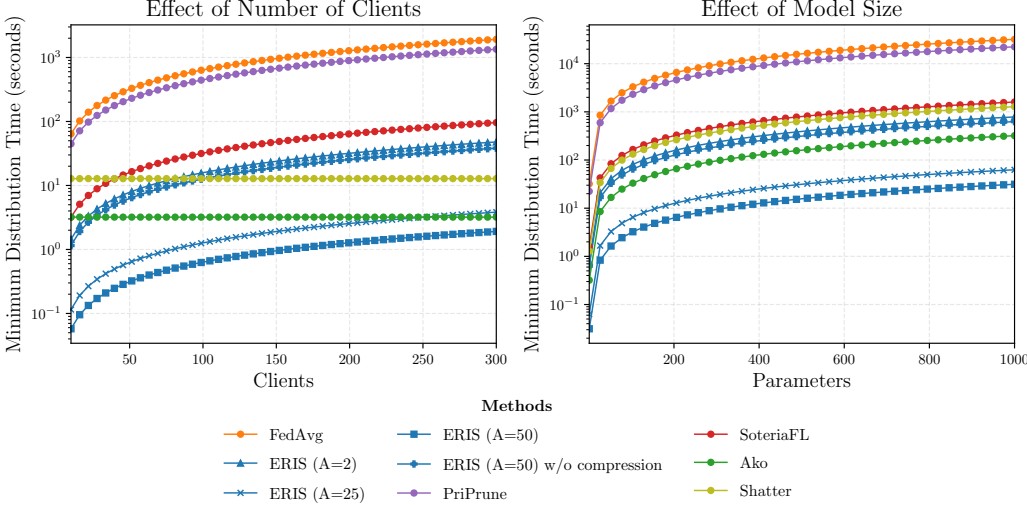

Figure 7: Minimum distribution time for a single training round for FedAvg, PriPrune, SoteriaFL, Ako, Shatter, and ERIS. The figure shows the minimum distribution time for a single round with $M = 320$ Mbit on a logarithmic scale (**left**), and the minimum distribution time for a single round with 50 training clients as the model size increases on a logarithmic scale (**right**).

**Communication efficiency.**  In addition to the main paper, Table 4 reports the full analysis of communication efficiency, extending Table 2 with results on MNIST and IMDB. The table compares

| Method | CNN/DailyMail | | IMDB | | CIFAR-10 | | MNIST | |
|---|---|---|---|---|---|---|---|---|
| | Exchanged | Dist. Time | Exchanged | Dist. Time | Exchanged | Dist. Time | Exchanged | Dist. Time |
| FedAvg (-LDP) | 5.2GB (100%) | 5200s | 268MB (100%) | 670s | 6.6MB (100%) | 33s | 248KB (100%) | 1.24s |
| Shatter | 5.2GB (100%) | 780s | 268MB (100%) | 53.6s | 6.6MB (100%) | 1.32s | 248KB (100%) | 0.05s |
| PriPrune (0.01) | 4.68GB (90%) | 4680s | 241.2MB (90%) | 603s | 6.53MB (99%) | 32.65s | 245.52KB (99%) | 1.23s |
| PriPrune (0.05) | 4.16GB (80%) | 4160s | 214.4MB (80%) | 536s | 6.27MB (95%) | 31.35s | 235.6KB (95%) | 1.18s |
| PriPrune (0.1) | 3.64GB (70%) | 3640s | 187.6MB (70%) | 469s | 5.9MB (90%) | 29.5s | 223.2KB (90%) | 1.12s |
| SoteriaFL | 0.26GB (5%) | 260s | 13.4MB (5%) | 33.5s | 0.33MB (5%) | 1.65s | 12.4KB (5%) | 0.06s |
| ERIS | 46.8MB (1%) | 4.68s | 30.87KB (0.012%) | 0.003s | 0.04MB (0.6%) | 0.0039s | 8.02KB (3.3%) | 0.0008s |

Table 4: Communication efficiency across datasets, showing per-client upload size and minimum distribution time per round.

per-client upload size and minimum distribution time per round (20MB/s bandwidth), under the same experimental settings as in Table 1. Across all datasets, ERIS consistently achieves the lowest communication overhead, reducing upload size to below 1% of FedAvg on CNN/DailyMail and CIFAR-10, and to just 0.012% on IMDB. Distribution time improvements are equally striking: for large models like CNN/DailyMail, ERIS cuts round time from over 5000s to under 5s, while for smaller models (e.g., MNIST) it completes a round in less than a millisecond. These results confirm that the combination of shifted compression and decentralized aggregation enables ERIS to deliver communication savings of several orders of magnitude while maintaining full convergence.

### F.3 EFFECT OF SHIFTED COMPRESSION ON MODEL UTILITY

This section provides additional results complementing the analysis in Paragraph 4.2. Figure 8 illustrates the impact of increasing the compression constant $\omega$ on test accuracy for CIFAR-10, under varying numbers of local training samples per client. We observe that up to $\omega = 340$—which corresponds to a compression rate of approximately 0.29%—test accuracy remains statistically unchanged. This indicates that the communication cost can be substantially reduced, sharing only 0.29% of gradients per client, without degrading performance. However, beyond this threshold, the aggressive compression starts discarding critical information, leading to compromised model convergence. As expected, further increasing $\omega$ results in progressively lower accuracy, highlighting a clear trade-off between compression strength and model utility.

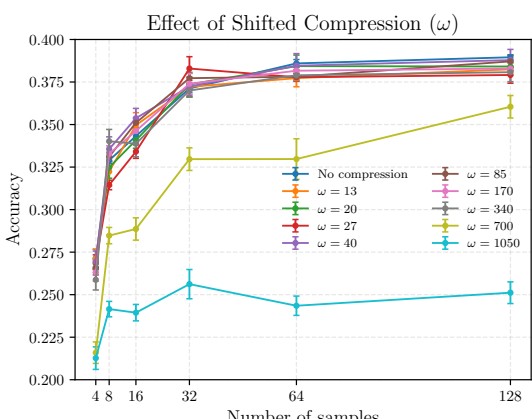

Figure 8: Effect of shifted compression on CIFAR-10 test accuracy, varying $\omega$ across different local training sample sizes.

### F.4 ROBUSTNESS TO AGGREGATOR DROPOUT

To evaluate the robustness of ERIS to aggregator unavailability during training, we conduct a controlled dropout (or failure) experiment in which, at every training round, a fixed proportion of aggregators is randomly deactivated. When an aggregator drops out, its corresponding model shard is not included in the global update, effectively reducing the step magnitude for that round.

Figure 9 reports the effect of increasing dropout rates on both (i) test accuracy and (ii) the best validation round at which the model reaches its peak performance (i.e., the minimum validation loss). The results show that ERIS maintains *nearly constant test accuracy* up to a dropout rate of 70%. This robustness stems from the fact that each aggregator is responsible for only a disjoint shard of the model; losing a subset of them does not invalidate training but simply scales down the effective update. The right plot of Figure 9 provides the key explanation. As the dropout rate increases, the number of rounds required to reach the validation optimum grows steadily. This indicates a progressive *slowdown in convergence*. When dropout remains below 70%, the model still converges

fully within the 200-round training budget. However, once the slowdown becomes large enough (i.e., beyond 70% dropout), the model no longer converges to the same optimum before the round limit is reached, causing the test accuracy to drop.

These results confirm that ERIS inherits strong resilience to aggregator failures: the system continues to improve as long as some fraction of shards is reliably aggregated, and degradation only appears when convergence is limited by a strict round cap rather than by algorithmic instability.

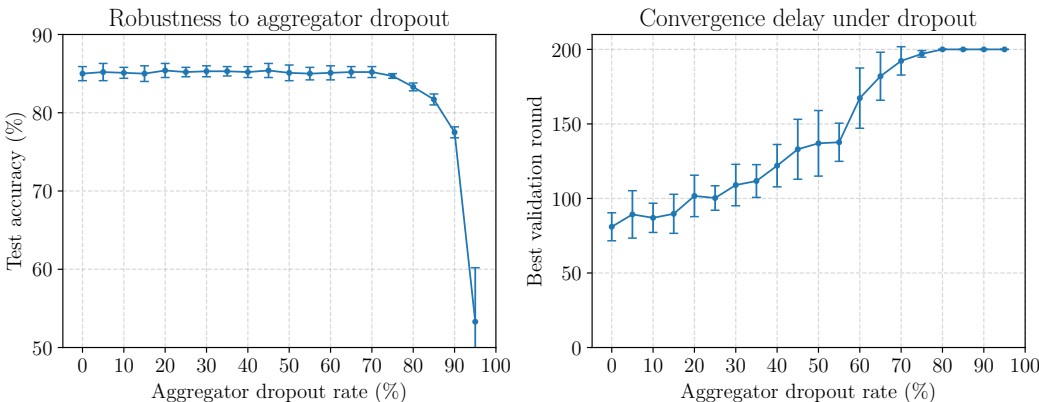

Figure 9: Robustness of ERIS to aggregator dropout. Left: test accuracy remains nearly constant up to a 70% dropout rate. Right: convergence slows as dropout increases, eventually hitting the 200-round cap, which explains the accuracy drop beyond 70%.

### F.5 DATA RECONSTRUCTION ATTACKS

To further assess the privacy guarantees of ERIS, we evaluate its resilience to Data Reconstruction Attacks (DRAs), which represent one of the most severe privacy threats in FL. To favour the attacker and stress-test our approach, we consider the uncommon but worst-case scenario where each client performs gradient descent with a mini-batch of size 1 and transmits the resulting gradient—which can be intercepted by an eavesdropper or a compromised aggregator/server. Therefore, we assume the adversary has white-box access to the client gradient.

Given this gradient, reconstruction methods such as DLG (Zhu et al., 2019), iDLG (Zhao et al., 2020), and ROG (Yue et al., 2023) aim to recover the original training sample by optimising inputs to match the leaked gradient. Unlike earlier gradient-matching attacks, ROG projects the unknown image into a low-dimensional latent space (e.g., via bicubic downsampling or an autoencoder) and optimises that compact representation so that the decoded image's gradients align with the leaked gradient, before applying a learnt enhancement module to obtain perceptually faithful reconstructions. In our experiments, a dedicated enhancement decoder was trained for each dataset using a hold-out set.

Figure 10 reports the reconstruction quality, measured via the LPIPS score (Zhang et al., 2018), as a function of the percentage of model parameters available to the attacker. The x-axis is plotted on a non-linear scale to improve readability in the low-percentage regime. The results are averaged over 200 reconstructed samples and tested across three datasets: MNIST, CIFAR-10, and LFW. The findings show that in the full-gradient setting (e.g., FedAvg), all DRAs can almost perfectly reconstruct the original image. However, as the proportion of accessible gradients decreases, the reconstruction quality of DLG and iDLG degrades significantly, with LPIPS scores approaching the baseline of random images when only 90% of the parameters are visible. Remarkably, even in the least favourable configuration of ERIS—with only two aggregators—the system already provides sufficient obfuscation to render reconstruction attacks ineffective, as highlighted by the shaded regions in the figure. A different pattern emerges for ROG, which tends to maintain higher reconstruction quality. Closer inspection of MNIST and LFW examples, however, reveals that this apparent advantage stems primarily from the trained enhancement decoder. This module effectively biases reconstructions toward the training distribution, thereby inflating similarity scores. In fact, even when random gradients are passed through the decoder (purple dashed line), the outputs achieve LPIPS values lower than random images, underscoring that the improvement reflects postprocessing artefacts. However,

once 25% or fewer gradients are accessible, the reconstructed outputs are severely distorted and no longer capture any semantically meaningful features of the original data.

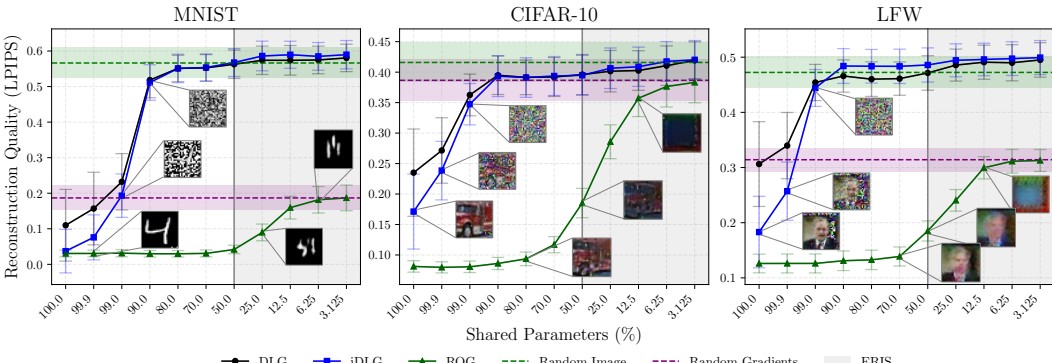

Figure 10: Reconstruction quality under DLG, iDLG, and ROG attacks as a function of the percentage of model parameters available to the attacker. The LPIPS score (higher is better) is averaged over 200 samples. The x-axis is plotted on a non-linear scale for improved clarity of the low-percentage regime. Shaded regions highlight the obfuscation achieved by ERIS, which renders reconstruction attacks ineffective even in its weakest configuration (two aggregators and no compression).

To further characterize the privacy guarantees, we evaluate the performance of the more advanced ROG attack—which better reveals residual privacy leakage even in obfuscated settings—across all implemented baselines, as well as several standalone compression methods (Table 5). The results confirm that compression alone is insufficient: both QSGD and uniform quantisation degrade reconstruction quality only at aggressive rates (e.g., $s = 4$), while Top-$k$ sparsification becomes effective only at extreme sparsity levels (0.98–0.99), where utility is severely compromised. Similarly, differentially private training via DP-SGD shows a clear trade-off between privacy and utility: with mild clipping (clip $= 10$) and low noise ($\sigma = 10^{-4}$), reconstructions remain close to FedAvg, whereas stronger noise or tighter clipping substantially degrades image quality but at the expense of model performance. PriPrune exhibits a comparable pattern, with higher pruning probabilities providing stronger obfuscation but leading to distorted reconstructions and reduced SSIM. Finally, ERIS achieves robust protection even in its least favourable setting ($A = 2$ aggregators), and the privacy guarantees strengthen as either the number of aggregators increases or the compression rate $\omega$ increases. Notably, at $A = 4$ or $\omega \geq 4$, the reconstruction quality approaches that of random gradients, indicating that ERIS effectively obfuscates client updates while preserving utility.

### F.6 BALANCING UTILITY AND PRIVACY - IID SETTING

This section provides detailed numerical results supporting the analysis in Paragraph 4.2. Specifically, we report test accuracy and Membership Inference Attack (MIA) accuracy for all evaluated methods across multiple datasets and varying local training sizes in IID setting. Tables 6–12 report results for CNN/DailyMail, IMDB, CIFAR-10, and MNIST. IMDB, CIFAR-10, and MNIST are evaluated across 4–128 client training samples, while CNN/DailyMail is limited to 16–128 samples, as overfitting saturates already at 16. Part of these values serve as the coordinates for Figure 3, which visualizes the utility–privacy trade-off achieved by ERIS and baselines. Importantly, the tables extend beyond the conditions illustrated in the figure by covering a wider set of hyperparameter configurations—namely, additional pruning rates ($p$) and privacy budgets ($\epsilon$) for LDP-based methods.

Notably, these results (both in Figure 3 and Tables 6–12) show two clear trends. First, a smaller amount of local data leads all methods to lower task accuracy and higher MIA accuracy, reflecting the stronger overfitting in this regime. Second, especially under low-data conditions, ERIS (with and without compression $\omega = 0$) consistently delivers markedly better privacy preservation while retaining competitive accuracy. For instance, on CNN/DailyMail with 16 samples, for the same ROUGE-1 score as FedAvg, ERIS reduces MIA accuracy from 100% to 77.7%; on IMDB with 4 samples, it lowers MIA accuracy from 82.9% to 65.2%, closely approaching the unattainable upper-bound of 64.4%. These findings highlight ERIS's robustness across data modalities and model capacities.

| Method | LPIPS ($\downarrow$) | SNR (dB) ($\downarrow$) | SSIM ($\downarrow$) |
|---|---|---|---|
| FedAvg | $0.193 \pm 0.059$ | $21.860 \pm 2.049$ | $0.871 \pm 0.054$ |
| QSGD (s=16) | $0.209 \pm 0.060$ | $21.503 \pm 2.082$ | $0.859 \pm 0.054$ |
| QSGD (s=8) | $0.250 \pm 0.065$ | $20.556 \pm 1.982$ | $0.829 \pm 0.056$ |
| QSGD (s=4) | $0.343 \pm 0.074$ | $18.402 \pm 1.794$ | $0.739 \pm 0.075$ |
| Uniform Quantization (s=16) | $0.243 \pm 0.066$ | $20.621 \pm 1.948$ | $0.833 \pm 0.059$ |
| Uniform Quantization (s=8) | $0.302 \pm 0.071$ | $18.816 \pm 1.924$ | $0.772 \pm 0.070$ |
| Uniform Quantization (s=4) | $0.403 \pm 0.079$ | $16.065 \pm 1.936$ | $0.634 \pm 0.091$ |
| Top-$k$ Sparsification (sparsity=0.90) | $0.228 \pm 0.063$ | $20.535 \pm 2.057$ | $0.841 \pm 0.058$ |
| Top-$k$ Sparsification (sparsity=0.98) | $0.392 \pm 0.083$ | $15.350 \pm 2.370$ | $0.644 \pm 0.094$ |
| Top-$k$ Sparsification (sparsity=0.99) | $0.456 \pm 0.094$ | $12.380 \pm 2.490$ | $0.472 \pm 0.105$ |
| DP-SGD (clip=10, $\sigma$=1e-4) | $0.200 \pm 0.062$ | $21.673 \pm 2.079$ | $0.867 \pm 0.051$ |
| DP-SGD (clip=10, $\sigma$=1e-3) | $0.340 \pm 0.071$ | $18.481 \pm 1.378$ | $0.741 \pm 0.081$ |
| DP-SGD (clip=10, $\sigma$=1e-2) | $0.498 \pm 0.091$ | $11.159 \pm 0.913$ | $0.252 \pm 0.109$ |
| DP-SGD (clip=1, $\sigma$=1e-4) | $0.432 \pm 0.094$ | $11.781 \pm 3.420$ | $0.446 \pm 0.127$ |
| DP-SGD (clip=1, $\sigma$=1e-3) | $0.436 \pm 0.095$ | $11.766 \pm 3.429$ | $0.445 \pm 0.128$ |
| DP-SGD (clip=1, $\sigma$=1e-2) | $0.472 \pm 0.090$ | $11.082 \pm 3.035$ | $0.354 \pm 0.098$ |
| PriPrune ($p = 1 \times 10^{-5}$) | $0.305 \pm 0.075$ | $19.072 \pm 1.894$ | $0.768 \pm 0.078$ |
| PriPrune ($p = 10^{-3}$) | $0.506 \pm 0.087$ | $9.614 \pm 2.250$ | $0.148 \pm 0.053$ |
| PriPrune ($p = 0.1$) | $0.569 \pm 0.067$ | $8.477 \pm 2.637$ | $0.094 \pm 0.043$ |
| ERIS ($\omega = 1, A = 2$) | $0.453 \pm 0.081$ | $11.678 \pm 2.259$ | $0.407 \pm 0.071$ |
| ERIS ($\omega = 4, A = 2$) | $0.524 \pm 0.078$ | $8.906 \pm 2.494$ | $0.134 \pm 0.047$ |
| ERIS ($\omega = 9, A = 2$) | $0.547 \pm 0.073$ | $8.556 \pm 2.609$ | $0.095 \pm 0.045$ |
| ERIS ($\omega = 0, A = 2$) | $0.458 \pm 0.081$ | $11.658 \pm 2.268$ | $0.404 \pm 0.070$ |
| ERIS ($\omega = 0, A = 4$) | $0.514 \pm 0.078$ | $9.216 \pm 2.418$ | $0.170 \pm 0.046$ |
| ERIS ($\omega = 0, A = 8$) | $0.546 \pm 0.075$ | $8.620 \pm 2.591$ | $0.105 \pm 0.043$ |
| Random Gradients | $0.572 \pm 0.065$ | $8.482 \pm 2.639$ | $0.094 \pm 0.045$ |

Table 5: Reconstruction quality under ROG attacks across privacy-preserving mechanisms and compression techniques on CIFAR-10. Lower metric values indicate stronger defenses.

| Local Training Size | 16 **samples** | | 32 **samples** | | 64 **samples** | | 128 **samples** | |
|---|---|---|---|---|---|---|---|---|
| Method | R-1 ($\uparrow$) | MIA Acc. ($\downarrow$) | R-1 ($\uparrow$) | MIA Acc. ($\downarrow$) | R-1 ($\uparrow$) | MIA Acc. ($\downarrow$) | R-1 ($\uparrow$) | MIA Acc. ($\downarrow$) |
| FedAvg | $30.37 \pm 1.25$ | $100.00 \pm 0.00$ | $32.21 \pm 1.46$ | $98.75 \pm 1.25$ | $34.27 \pm 0.65$ | $96.46 \pm 0.36$ | $36.04 \pm 0.57$ | $96.57 \pm 0.90$ |
| FedAvg $(10, \delta)$-LDP | $25.66 \pm 0.86$ | $54.17 \pm 5.20$ | $26.26 \pm 0.07$ | $50.00 \pm 3.31$ | $26.30 \pm 0.10$ | $49.33 \pm 2.53$ | $25.78 \pm 0.10$ | $54.41 \pm 1.47$ |
| FedAvg $(100, \delta)$-LDP | $26.33 \pm 0.07$ | $54.17 \pm 5.20$ | $26.36 \pm 0.10$ | $50.42 \pm 2.89$ | $24.90 \pm 0.13$ | $48.75 \pm 2.25$ | $26.32 \pm 0.10$ | $54.31 \pm 1.19$ |
| SoteriaFL ($\epsilon = 100$) | $25.89 \pm 0.12$ | $54.13 \pm 1.15$ | $26.09 \pm 0.07$ | $54.27 \pm 5.20$ | $26.01 \pm 0.11$ | $50.35 \pm 2.89$ | $24.02 \pm 0.15$ | $49.15 \pm 2.15$ |
| SoteriaFL ($\epsilon = 10$) | $25.78 \pm 0.91$ | $54.02 \pm 1.48$ | $25.90 \pm 0.36$ | $54.17 \pm 5.20$ | $25.90 \pm 0.75$ | $50.83 \pm 2.60$ | $24.75 \pm 0.79$ | $49.54 \pm 2.60$ |
| PriPrune ($p = 0.1$) | $32.21 \pm 0.44$ | $95.83 \pm 2.89$ | $26.01 \pm 0.70$ | $82.92 \pm 1.91$ | $30.97 \pm 0.55$ | $90.00 \pm 1.08$ | $33.90 \pm 1.45$ | $88.73 \pm 0.90$ |
| PriPrune ($p = 0.2$) | $29.96 \pm 0.94$ | $88.33 \pm 2.89$ | $26.61 \pm 0.72$ | $76.67 \pm 3.61$ | $29.00 \pm 0.41$ | $79.38 \pm 3.80$ | $31.49 \pm 1.89$ | $79.02 \pm 0.61$ |
| PriPrune ($p = 0.3$) | $18.41 \pm 15.11$ | $74.17 \pm 3.82$ | $21.80 \pm 0.67$ | $70.00 \pm 3.31$ | $29.00 \pm 1.10$ | $70.83 \pm 3.44$ | $29.49 \pm 1.67$ | $70.39 \pm 0.74$ |
| Shatter | $30.05 \pm 1.22$ | $78.50 \pm 7.50$ | $30.38 \pm 0.59$ | $69.12 \pm 2.60$ | $33.04 \pm 0.65$ | $66.18 \pm 5.12$ | $34.35 \pm 0.38$ | $68.16 \pm 0.90$ |
| ERIS ($\omega \approx 100$) | $30.04 \pm 0.95$ | $77.73 \pm 6.29$ | $31.60 \pm 0.95$ | $68.27 \pm 3.61$ | $34.14 \pm 0.76$ | $64.38 \pm 5.12$ | $35.62 \pm 0.48$ | $67.84 \pm 0.74$ |
| ERIS ($\omega \approx \omega_{\text{SoteriaFL}}$) | $30.05 \pm 0.86$ | $78.33 \pm 6.29$ | $32.15 \pm 1.75$ | $68.75 \pm 3.31$ | $34.12 \pm 0.38$ | $63.75 \pm 4.38$ | $35.06 \pm 1.27$ | $67.65 \pm 0.78$ |
| ERIS ($\omega = 0$) | $30.37 \pm 1.25$ | $78.50 \pm 7.50$ | $32.41 \pm 1.46$ | $69.12 \pm 2.60$ | $34.04 \pm 0.76$ | $66.18 \pm 5.12$ | $36.04 \pm 0.57$ | $68.16 \pm 0.90$ |
| Min. Leakage | $30.37 \pm 1.25$ | $67.83 \pm 10.10$ | $32.41 \pm 1.46$ | $60.58 \pm 1.91$ | $34.27 \pm 0.65$ | $54.08 \pm 4.77$ | $36.04 \pm 0.57$ | $59.61 \pm 2.54$ |

Table 6: Comparison of ERIS with and without compression ($\omega$) against SOTA baselines in terms of ROUGE-1 and MIA accuracy on CNN/DailyMail with 16, 32, 64, and 128 local training samples.

## F.7 BALANCING UTILITY AND PRIVACY - NON-IID SETTING

We further evaluate the utility–privacy trade-off under non-IID client data, using a Dirichlet partition with $\alpha$=0.5 for IMDB and $\alpha$=0.2 for CIFAR-10 and MNIST. Figure 11 illustrates the utility–privacy trade-off across methods and datasets, where the ideal region corresponds to the top-right corner (high accuracy and low privacy leakage). Non-IID distributions generally make convergence more challenging, lowering overall accuracy and increasing variability across clients. Nevertheless, ERIS remains stable and consistently reduces privacy leakage. For example, on IMDB with 4 local samples,

| Local Training Size | 4 samples | | 8 samples | | 16 samples | |
|---|---|---|---|---|---|---|
| Method | Accuracy ($\uparrow$) | MIA Acc. ($\downarrow$) | Accuracy ($\uparrow$) | MIA Acc. ($\downarrow$) | Accuracy ($\uparrow$) | MIA Acc. ($\downarrow$) |
| FedAvg | 71.73 ± 4.93 | 82.93 ± 6.39 | 79.56 ± 0.61 | 78.40 ± 3.95 | 80.52 ± 0.30 | 66.91 ± 1.81 |
| FedAvg ($\epsilon = 100, \delta$)-LDP | 53.79 ± 0.08 | 55.56 ± 4.91 | 53.82 ± 0.06 | 53.33 ± 1.00 | 53.92 ± 0.11 | 50.06 ± 2.81 |
| FedAvg ($\epsilon = 10, \delta$)-LDP | 53.80 ± 0.03 | 52.80 ± 5.82 | 53.81 ± 0.02 | 50.40 ± 3.12 | 53.83 ± 0.06 | 49.89 ± 1.74 |
| SoteriaFL ($\epsilon = 100, \delta$) | 53.46 ± 0.15 | 55.56 ± 6.00 | 54.73 ± 0.15 | 54.40 ± 1.96 | 53.74 ± 0.16 | 50.30 ± 2.69 |
| SoteriaFL ($\epsilon = 10, \delta$) | 53.36 ± 0.29 | 55.20 ± 5.63 | 54.01 ± 0.24 | 51.36 ± 3.48 | 53.67 ± 0.29 | 50.11 ± 1.58 |
| PriPrune ($p = 0.1$) | 54.40 ± 5.46 | 80.53 ± 4.59 | 71.70 ± 2.09 | 74.72 ± 3.63 | 77.64 ± 1.44 | 65.31 ± 2.18 |
| PriPrune ($p = 0.2$) | 52.78 ± 2.39 | 74.67 ± 4.99 | 58.55 ± 5.30 | 71.36 ± 2.23 | 65.33 ± 8.65 | 62.84 ± 2.41 |
| PriPrune ($p = 0.3$) | 53.52 ± 2.79 | 70.40 ± 3.20 | 55.92 ± 3.49 | 65.76 ± 3.63 | 60.32 ± 5.71 | 59.82 ± 2.85 |
| Shatter | 68.52 ± 4.66 | 67.52 ± 2.80 | 74.84 ± 1.88 | 62.56 ± 3.77 | 77.91 ± 0.55 | 54.75 ± 1.97 |
| ERIS ($\omega \approx 8000$) | 71.28 ± 4.74 | 65.22 ± 2.95 | 79.28 ± 0.71 | 60.51 ± 2.50 | 80.11 ± 0.46 | 54.12 ± 1.78 |
| ERIS ($\omega \approx \omega_{\text{SoteriaFL}}$) | 71.74 ± 4.94 | 65.87 ± 3.22 | 79.55 ± 0.61 | 60.52 ± 3.00 | 80.51 ± 0.31 | 54.56 ± 1.57 |
| ERIS ($\omega = 0$) | 70.15 ± 4.24 | 67.67 ± 2.89 | 79.39 ± 0.57 | 62.45 ± 3.79 | 80.49 ± 0.33 | 54.72 ± 1.90 |
| Min. Leakage | 72.39 ± 1.99 | 64.44 ± 2.27 | 79.33 ± 0.75 | 58.67 ± 2.10 | 80.68 ± 0.09 | 53.21 ± 2.53 |

Table 7: Comparison of ERIS with and without compression ($\omega$) against SOTA baselines in terms of test and MIA accuracy on IMDB with 4, 8, and 16 local training samples.

| Local Training Size | 32 samples | | 64 samples | | 128 samples | |
|---|---|---|---|---|---|---|
| Method | Accuracy ($\uparrow$) | MIA Acc. ($\downarrow$) | Accuracy ($\uparrow$) | MIA Acc. ($\downarrow$) | Accuracy ($\uparrow$) | MIA Acc. ($\downarrow$) |
| FedAvg | 81.62 ± 0.11 | 63.58 ± 1.47 | 81.70 ± 0.05 | 60.54 ± 2.11 | 82.45 ± 0.18 | 56.89 ± 0.81 |
| FedAvg ($\epsilon = 100, \delta$)-LDP | 54.11 ± 0.15 | 51.56 ± 1.71 | 54.50 ± 0.19 | 50.67 ± 1.60 | 55.09 ± 0.32 | 51.26 ± 0.49 |
| FedAvg ($\epsilon = 10, \delta$)-LDP | 53.97 ± 0.10 | 50.02 ± 1.52 | 54.12 ± 0.12 | 49.84 ± 0.86 | 54.30 ± 0.19 | 50.37 ± 0.92 |
| SoteriaFL ($\epsilon = 100, \delta$) | 54.87 ± 0.72 | 51.94 ± 1.94 | 55.81 ± 0.62 | 50.98 ± 1.87 | 56.08 ± 1.19 | 51.17 ± 0.52 |
| SoteriaFL ($\epsilon = 10, \delta$) | 54.35 ± 0.39 | 50.10 ± 1.83 | 54.79 ± 0.35 | 50.12 ± 0.93 | 55.28 ± 0.57 | 50.64 ± 0.92 |
| PriPrune ($p = 0.1$) | 79.93 ± 0.23 | 61.87 ± 1.25 | 80.34 ± 0.10 | 59.61 ± 2.11 | 80.87 ± 0.11 | 56.11 ± 0.93 |
| PriPrune ($p = 0.2$) | 71.48 ± 7.20 | 59.54 ± 1.71 | 72.12 ± 1.91 | 58.21 ± 2.11 | 77.52 ± 0.55 | 55.02 ± 1.09 |
| PriPrune ($p = 0.3$) | 61.20 ± 7.02 | 56.43 ± 1.55 | 62.01 ± 5.91 | 56.63 ± 1.76 | 68.95 ± 1.92 | 54.20 ± 1.18 |
| Shatter | 79.34 ± 0.64 | 53.79 ± 1.56 | 80.19 ± 0.48 | 53.68 ± 0.95 | 80.84 ± 0.19 | 52.02 ± 1.01 |
| ERIS ($\omega \approx 8000$) | 80.99 ± 0.22 | 53.44 ± 1.26 | 81.16 ± 0.19 | 52.77 ± 1.62 | 81.59 ± 0.21 | 51.80 ± 1.07 |
| ERIS ($\omega \approx \omega_{\text{SoteriaFL}}$) | 81.62 ± 0.12 | 53.81 ± 1.47 | 81.71 ± 0.05 | 52.78 ± 1.70 | 82.45 ± 0.16 | 51.54 ± 1.27 |
| ERIS ($\omega = 0$) | 81.63 ± 0.12 | 53.98 ± 1.48 | 81.70 ± 0.06 | 53.54 ± 0.93 | 82.38 ± 0.13 | 52.08 ± 1.07 |
| Min. Leakage | 81.58 ± 0.11 | 52.57 ± 1.27 | 81.67 ± 0.04 | 53.05 ± 1.32 | 82.44 ± 0.09 | 51.53 ± 1.12 |

Table 8: Comparison of ERIS with and without compression ($\omega$) against SOTA baselines in terms of test and MIA accuracy on IMDB with 32, 64, and 128 local training samples.

| Local Training Size | 4 samples | | 8 samples | | 16 samples | |
|---|---|---|---|---|---|---|
| Method | Accuracy ($\uparrow$) | MIA Acc. ($\downarrow$) | Accuracy ($\uparrow$) | MIA Acc. ($\downarrow$) | Accuracy ($\uparrow$) | MIA Acc. ($\downarrow$) |
| FedAvg | 27.12 ± 1.20 | 84.80 ± 4.59 | 32.98 ± 0.61 | 75.84 ± 2.85 | 34.43 ± 1.04 | 70.15 ± 1.41 |
| FedAvg ($\epsilon = 10, \delta$)-LDP | 10.33 ± 0.53 | 81.20 ± 3.90 | 14.93 ± 2.01 | 72.40 ± 2.60 | 18.92 ± 1.31 | 62.55 ± 1.19 |
| FedAvg ($\epsilon = 1, \delta$)-LDP | 10.34 ± 0.22 | 66.50 ± 3.78 | 10.00 ± 0.00 | 63.60 ± 2.42 | 10.00 ± 0.00 | 56.05 ± 0.49 |
| SoteriaFL ($\epsilon = 10, \delta$) | 10.00 ± 0.00 | 69.87 ± 1.86 | 10.06 ± 0.12 | 64.16 ± 1.25 | 10.85 ± 1.06 | 58.25 ± 2.10 |
| SoteriaFL ($\epsilon = 1, \delta$) | 9.99 ± 0.00 | 65.67 ± 1.11 | 10.00 ± 0.00 | 62.10 ± 1.56 | 10.00 ± 0.00 | 53.86 ± 0.67 |
| PriPrune ($p = 0.01$) | 13.74 ± 2.05 | 74.80 ± 2.87 | 28.42 ± 0.39 | 75.36 ± 3.60 | 29.57 ± 0.70 | 69.82 ± 1.59 |
| PriPrune ($p = 0.05$) | 10.09 ± 0.36 | 67.33 ± 2.63 | 12.77 ± 2.52 | 61.68 ± 3.04 | 11.55 ± 1.80 | 54.44 ± 1.50 |
| PriPrune ($p = 0.1$) | 10.00 ± 0.00 | 64.27 ± 2.62 | 10.03 ± 0.04 | 58.80 ± 2.83 | 10.00 ± 0.00 | 52.62 ± 1.37 |
| Shatter | 11.47 ± 1.75 | 77.95 ± 5.63 | 11.57 ± 1.96 | 70.49 ± 2.74 | 12.42 ± 1.65 | 64.22 ± 1.85 |
| ERIS ($\omega \approx 170$) | 26.31 ± 1.16 | 71.63 ± 4.28 | 33.28 ± 1.06 | 68.48 ± 2.30 | 34.62 ± 1.42 | 59.58 ± 2.26 |
| ERIS ($\omega \approx \omega_{\text{SoteriaFL}}$) | 26.84 ± 0.68 | 73.86 ± 5.27 | 32.48 ± 1.43 | 67.95 ± 1.45 | 34.08 ± 1.05 | 59.23 ± 1.43 |
| ERIS ($\omega = 0$) | 27.13 ± 1.19 | 77.90 ± 5.55 | 32.90 ± 0.40 | 70.75 ± 2.70 | 34.32 ± 0.91 | 64.21 ± 1.95 |
| Min. Leakage | 27.11 ± 1.17 | 70.27 ± 4.69 | 33.11 ± 0.62 | 65.44 ± 2.43 | 34.62 ± 0.91 | 56.87 ± 1.34 |

Table 9: Comparison of ERIS with and without compression ($\omega$) against SOTA baselines in terms of test and MIA accuracy on CIFAR-10 with 4, 8, and 16 local training samples.

| Local Training Size | 32 samples | | 64 samples | | 128 samples | |
|---|---|---|---|---|---|---|
| Method | Accuracy (↑) | MIA Acc. (↓) | Accuracy (↑) | MIA Acc. (↓) | Accuracy (↑) | MIA Acc. (↓) |
| FedAvg | 37.24 ± 0.41 | 64.57 ± 0.72 | 38.50 ± 0.44 | 59.29 ± 0.79 | 38.88 ± 0.32 | 56.11 ± 0.75 |
| FedAvg ($\epsilon = 10, \delta$)-LDP | 22.31 ± 1.12 | 57.14 ± 1.39 | 23.36 ± 0.85 | 53.99 ± 0.83 | 24.13 ± 0.32 | 52.81 ± 0.49 |
| FedAvg ($\epsilon = 1, \delta$)-LDP | 10.00 ± 0.00 | 57.57 ± 0.26 | 13.96 ± 1.14 | 54.42 ± 0.40 | 19.29 ± 0.37 | 53.12 ± 0.16 |
| SoteriaFL ($\epsilon = 10, \delta$) | 19.68 ± 0.78 | 55.68 ± 1.09 | 26.04 ± 0.52 | 52.94 ± 0.74 | 26.46 ± 0.25 | 52.07 ± 0.55 |
| SoteriaFL ($\epsilon = 1, \delta$) | 10.00 ± 0.00 | 54.57 ± 0.50 | 10.00 ± 0.00 | 53.28 ± 0.61 | 12.20 ± 1.25 | 52.58 ± 0.53 |
| PriPrune ($p = 0.01$) | 29.39 ± 0.50 | 63.73 ± 0.96 | 28.70 ± 0.51 | 57.09 ± 0.67 | 27.99 ± 0.32 | 53.22 ± 0.70 |
| PriPrune ($p = 0.05$) | 11.80 ± 2.44 | 52.80 ± 1.80 | 11.05 ± 1.60 | 52.01 ± 0.35 | 10.21 ± 0.28 | 51.01 ± 0.20 |
| PriPrune ($p = 0.1$) | 10.00 ± 0.00 | 51.94 ± 1.86 | 10.00 ± 0.01 | 51.06 ± 0.64 | 10.00 ± 0.00 | 50.48 ± 0.81 |
| Shatter | 12.32 ± 2.03 | 58.58 ± 0.95 | 12.96 ± 2.16 | 54.65 ± 0.54 | 13.64 ± 1.55 | 52.03 ± 0.49 |
| ERIS ($\omega \approx 170$) | 37.40 ± 1.36 | 57.49 ± 0.85 | 38.16 ± 1.01 | 53.98 ± 0.41 | 38.30 ± 0.88 | 51.70 ± 0.53 |
| ERIS ($\omega \approx \omega_{\text{SoteriaFL}}$) | 37.36 ± 1.59 | 56.54 ± 0.94 | 38.43 ± 1.45 | 53.56 ± 0.57 | 38.41 ± 0.51 | 51.66 ± 0.52 |
| ERIS ($\omega = 0$) | 37.12 ± 0.55 | 58.63 ± 0.98 | 38.59 ± 0.50 | 54.60 ± 0.49 | 38.95 ± 0.32 | 52.04 ± 0.41 |
| Min. Leakage | 37.25 ± 0.38 | 55.81 ± 0.81 | 38.57 ± 0.37 | 53.06 ± 0.47 | 38.88 ± 0.36 | 51.67 ± 0.48 |

Table 10: Comparison of ERIS with and without compression ($\omega$) against SOTA baselines in terms of test and MIA accuracy on CIFAR-10 with 32, 64, and 128 local training samples.

| Local Training Size | 4 samples | | 8 samples | | 16 samples | |
|---|---|---|---|---|---|---|
| Method | Accuracy (↑) | MIA Acc. (↓) | Accuracy (↑) | MIA Acc. (↓) | Accuracy (↑) | MIA Acc. (↓) |
| FedAvg | 80.69 ± 1.71 | 82.13 ± 1.65 | 86.42 ± 0.88 | 72.00 ± 3.01 | 89.23 ± 0.74 | 65.78 ± 2.24 |
| FedAvg ($\epsilon = 10, \delta$)-LDP | 39.65 ± 3.14 | 69.07 ± 1.67 | 50.84 ± 4.75 | 59.44 ± 2.00 | 64.40 ± 1.53 | 57.67 ± 1.61 |
| FedAvg ($\epsilon = 1, \delta$)-LDP | 9.73 ± 0.46 | 69.50 ± 1.72 | 10.70 ± 1.27 | 58.50 ± 0.71 | 19.80 ± 1.38 | 57.55 ± 1.66 |
| SoteriaFL ($\epsilon = 10, \delta$) | 8.83 ± 2.65 | 71.47 ± 1.81 | 32.15 ± 2.00 | 57.68 ± 1.83 | 67.01 ± 1.31 | 56.87 ± 1.69 |
| SoteriaFL ($\epsilon = 1, \delta$) | 10.31 ± 0.38 | 67.50 ± 2.33 | 10.84 ± 0.79 | 57.90 ± 2.37 | 12.72 ± 1.33 | 57.27 ± 1.44 |
| PriPrune ($p = 0.01$) | 47.89 ± 8.33 | 77.20 ± 3.33 | 70.60 ± 3.67 | 68.32 ± 4.28 | 84.81 ± 0.31 | 63.56 ± 2.09 |
| PriPrune ($p = 0.05$) | 17.01 ± 4.22 | 58.00 ± 3.45 | 18.97 ± 3.12 | 50.16 ± 2.60 | 26.47 ± 2.59 | 54.51 ± 1.62 |
| PriPrune ($p = 0.1$) | 11.99 ± 1.87 | 56.67 ± 3.18 | 13.33 ± 2.21 | 49.44 ± 2.54 | 19.46 ± 1.35 | 53.42 ± 1.54 |
| Shatter | 11.96 ± 2.33 | 70.42 ± 2.21 | 12.32 ± 2.92 | 56.51 ± 2.86 | 14.55 ± 4.24 | 55.61 ± 1.33 |
| ERIS ($\omega \approx 30$) | 78.72 ± 1.19 | 68.48 ± 3.11 | 84.84 ± 0.58 | 55.14 ± 2.73 | 90.26 ± 0.11 | 56.11 ± 1.58 |
| ERIS ($\omega \approx \omega_{\text{SoteriaFL}}$) | 78.31 ± 1.45 | 68.70 ± 3.08 | 85.59 ± 0.67 | 54.43 ± 2.14 | 90.04 ± 0.43 | 55.58 ± 1.77 |
| ERIS ($\omega = 0$) | 80.47 ± 1.75 | 70.39 ± 2.19 | 86.28 ± 1.00 | 56.35 ± 2.89 | 89.27 ± 0.73 | 55.60 ± 1.27 |
| Min. Leakage | 80.68 ± 1.95 | 66.67 ± 2.67 | 86.30 ± 1.06 | 54.32 ± 1.67 | 89.26 ± 0.74 | 55.38 ± 1.56 |

Table 11: Comparison of ERIS with and without compression ($\omega$) against SOTA baselines in terms of test and MIA accuracy on MNIST with 4, 8, and 16 local training samples.

| Local Training Size | 32 samples | | 64 samples | | 128 samples | |
|---|---|---|---|---|---|---|
| Method | Accuracy (↑) | MIA Acc. (↓) | Accuracy (↑) | MIA Acc. (↓) | Accuracy (↑) | MIA Acc. (↓) |
| FedAvg | 91.48 ± 0.37 | 59.94 ± 1.11 | 92.55 ± 0.06 | 56.68 ± 1.92 | 93.11 ± 0.16 | 54.14 ± 0.65 |
| FedAvg ($\epsilon = 10, \delta$)-LDP | 70.35 ± 1.31 | 53.05 ± 1.09 | 70.43 ± 1.27 | 53.00 ± 1.00 | 70.48 ± 0.39 | 51.20 ± 0.91 |
| FedAvg ($\epsilon = 1, \delta$)-LDP | 57.75 ± 0.87 | 54.71 ± 0.85 | 70.96 ± 0.71 | 52.72 ± 0.74 | 70.56 ± 0.34 | 51.40 ± 0.66 |
| SoteriaFL ($\epsilon = 10, \delta$) | 77.95 ± 2.38 | 53.28 ± 0.60 | 78.15 ± 2.66 | 52.16 ± 0.84 | 79.50 ± 1.45 | 51.29 ± 0.77 |
| SoteriaFL ($\epsilon = 1, \delta$) | 8.49 ± 1.87 | 54.14 ± 0.53 | 40.37 ± 1.44 | 52.83 ± 0.24 | 68.75 ± 0.38 | 51.89 ± 0.53 |
| PriPrune ($p = 0.01$) | 87.77 ± 0.15 | 56.99 ± 1.14 | 87.01 ± 0.17 | 54.54 ± 1.03 | 86.38 ± 0.19 | 52.65 ± 0.69 |
| PriPrune ($p = 0.05$) | 33.01 ± 0.73 | 51.41 ± 1.34 | 34.81 ± 1.98 | 51.39 ± 1.00 | 33.87 ± 0.63 | 50.69 ± 0.93 |
| PriPrune ($p = 0.1$) | 21.08 ± 1.39 | 50.74 ± 1.58 | 20.68 ± 0.90 | 51.14 ± 0.98 | 20.46 ± 0.37 | 50.65 ± 0.87 |
| Shatter | 16.51 ± 6.16 | 52.02 ± 1.09 | 18.50 ± 6.06 | 51.46 ± 0.79 | 21.29 ± 7.22 | 51.61 ± 0.76 |
| ERIS ($\omega \approx 30$) | 92.56 ± 0.30 | 52.71 ± 0.86 | 93.58 ± 0.23 | 51.74 ± 0.66 | 94.02 ± 0.19 | 51.65 ± 0.64 |
| ERIS ($\omega \approx \omega_{\text{SoteriaFL}}$) | 92.27 ± 0.21 | 52.66 ± 0.73 | 93.23 ± 0.10 | 51.57 ± 0.74 | 93.78 ± 0.10 | 51.71 ± 0.67 |
| ERIS ($\omega = 0$) | 91.52 ± 0.32 | 52.08 ± 1.04 | 92.54 ± 0.04 | 51.45 ± 0.85 | 93.12 ± 0.16 | 51.53 ± 0.79 |
| Min. Leakage | 91.47 ± 0.39 | 52.06 ± 1.04 | 92.56 ± 0.05 | 51.41 ± 0.84 | 93.14 ± 0.17 | 51.50 ± 0.85 |

Table 12: Comparison of ERIS with and without compression ($\omega$) against SOTA baselines in terms of test and MIA accuracy on MNIST with 32, 64, and 128 local training samples.

ERIS matches FedAvg in accuracy while reducing MIA accuracy from 91.7% to 80.7%, approaching the ideal upper-bound of not sharing local gradients. On CIFAR-10 and MNIST, ERIS even matches or slightly surpass non-private FedAvg in terms of accuracy, while still offering strong privacy protection. By contrast, privacy-enhancing baselines such as Shatter and FedAvg-LDP struggle to maintain utility, often remaining close to random-guess performance, particularly when models are trained from scratch. Table 13 reports detailed mean accuracy and MIA accuracy values, averaged over varying local sample sizes. Together, these results confirm that the advantages of ERIS extend robustly to heterogeneous data distributions.

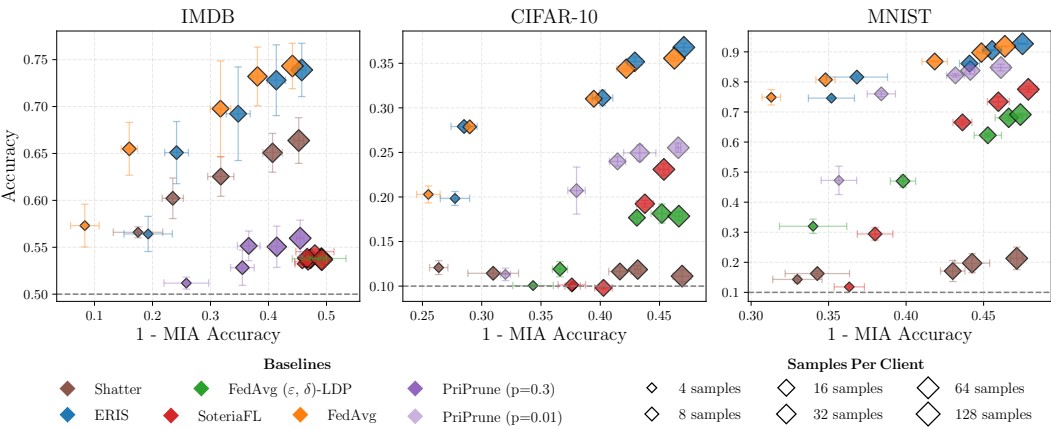

Figure 11: Comparison of test accuracy and MIA accuracy across varying model sizes and client-side overfitting levels, controlled via the number of client training samples under non-IID setting.

| Method | IMDB – DistilBERT | | CIFAR-10 – ResNet9 | | MNIST – LeNet5 | |
|---|---|---|---|---|---|---|
| | Acc. ($\uparrow$) | MIA Acc. ($\downarrow$) | Acc. ($\uparrow$) | MIA Acc. ($\downarrow$) | Acc. ($\uparrow$) | MIA Acc. ($\downarrow$) |
| FedAvg | $68.02 \pm 7.03$ | $72.34 \pm 3.33$ | $29.83 \pm 0.85$ | $63.52 \pm 1.28$ | $84.80 \pm 1.76$ | $60.17 \pm 1.29$ |
| FedAvg $(\epsilon, \delta)$-LDP | $53.79 \pm 0.30$ | $52.00 \pm 3.97$ | $15.13 \pm 1.37$ | $58.83 \pm 1.73$ | $55.69 \pm 2.83$ | $57.39 \pm 2.31$ |
| SoteriaFL $(\epsilon, \delta)$ | $53.75 \pm 0.81$ | $52.70 \pm 3.86$ | $14.46 \pm 0.55$ | $59.07 \pm 1.64$ | $51.75 \pm 2.77$ | $57.64 \pm 1.80$ |
| PriPrune $(p_1)$ | $57.01 \pm 7.37$ | $70.48 \pm 3.42$ | $21.30 \pm 2.36$ | $59.73 \pm 1.52$ | $74.80 \pm 3.78$ | $58.50 \pm 1.56$ |
| PriPrune $(p_2)$ | $53.64 \pm 3.40$ | $67.52 \pm 3.52$ | $11.51 \pm 1.41$ | $57.71 \pm 3.03$ | $24.94 \pm 5.34$ | $53.95 \pm 1.89$ |
| PriPrune $(p_3)$ | $54.03 \pm 3.68$ | $63.04 \pm 4.23$ | $10.98 \pm 0.91$ | $55.53 \pm 2.24$ | $15.54 \pm 1.47$ | $52.64 \pm 2.45$ |
| Shatter | $62.16 \pm 4.19$ | $68.26 \pm 4.85$ | $11.63 \pm 1.31$ | $62.19 \pm 2.18$ | $17.73 \pm 5.56$ | $59.67 \pm 2.67$ |
| ERIS | $67.49 \pm 7.51$ | $66.95 \pm 4.72$ | $30.16 \pm 1.35$ | $62.72 \pm 2.04$ | $85.10 \pm 0.84$ | $58.17 \pm 2.24$ |
| Min. Leakage | $68.88 \pm 6.75$ | $68.85 \pm 3.00$ | $29.80 \pm 0.86$ | $61.92 \pm 3.09$ | $84.95 \pm 1.73$ | $56.08 \pm 1.67$ |

Table 13: Mean test accuracy and MIA accuracy, averaged over varying local sample sizes. For DP-based methods, $\epsilon=10$; for PriPrune, pruning rates are $p \in \{0.1, 0.2, 0.3\}$ on IMDB and $p \in \{0.01, 0.05, 0.1\}$ on CIFAR-10/MNIST.

## F.8 BALANCING UTILITY AND PRIVACY - BIASED GRADIENT ESTIMATOR

In this section, we extend our analysis of the utility–privacy trade-off to the biased setting (already adopted for CNN/DailyMail dataset), where each client performs multiple local updates per communication round, thereby introducing bias into the gradient estimator. The training hyperparameters are detailed in Section E.1.

Figure 12 summarizes the performance of ERIS and several SOTS privacy-preserving baselines in terms of test accuracy and MIA accuracy across different model sizes and local training regimes. As in the main paper (Figure 3), we evaluate datasets with distinct memorization characteristics—ranging from lightweight models such as LeNet-5 on MNIST to large-scale architectures like GPT-Neo 1.3B on CNN/DailyMail—and vary client-side overfitting by controlling the number of training samples per client. The observed trends mirror those under unbiased conditions: ERIS consistently achieves the best overall trade-off, retaining accuracy comparable to non-private FedAvg while substantially reducing privacy leakage toward the idealized *Min. Leakage* baseline. For instance, on IMDB with 4 local samples per client and identical training conditions (e.g., same communication rounds), ERIS achieves an accuracy of $67.8 \pm 4.9$, comparable to FedAvg's $66.9 \pm 5.5$, while significantly reducing MIA accuracy from 92.3% to 68.2%—approaching the unattainable upper bound of 66.9% obtained

by not sharing local gradients. The only methods that surpass ERIS in privacy protection are DP-based approaches, which, however, degrade test accuracy to nearly random-guess levels, namely SoteriaFL ($53.1 \pm 0.8$) and FedAvg-LDP ($53.4 \pm 0.5$). Indeed, DP-based methods reduce leakage only at the cost of severe utility degradation, particularly for larger models, while decentralized methods with partial gradient exchange, such as Shatter, often fail to converge within the predefined number of communication rounds—especially when models are trained from scratch.

Table 14 reports mean test and MIA accuracy under the biased setting, complementing trends in Figure 12. Consistent with the figure, ERIS delivers the strongest utility–privacy balance across datasets, maintaining accuracy close to FedAvg while reducing leakage toward the *Min. Leakage* baseline. DP-based methods achieve lower leakage but at a steep accuracy cost, PriPrune trades off utility and privacy depending on the pruning rate, and Shatter struggles to converge reliably. These results further confirm that ERIS offers the most favorable trade-off, even in biased local training regimes.

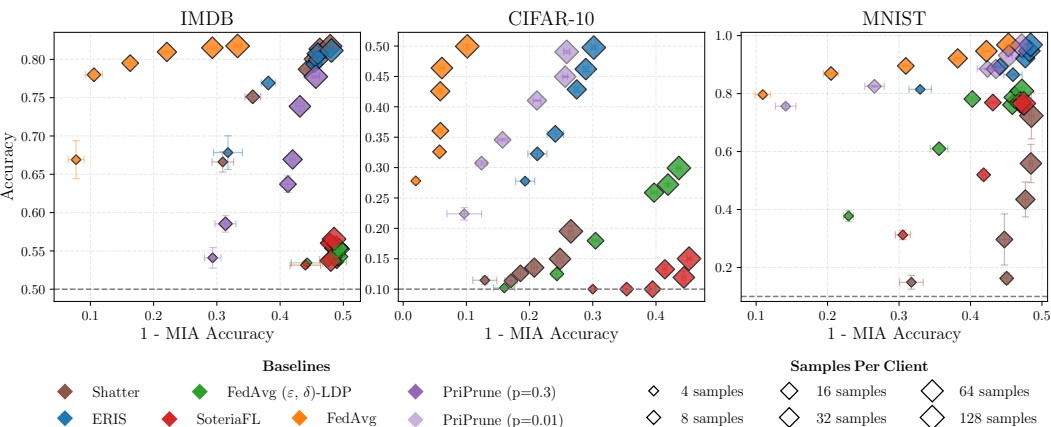

Figure 12: Comparison of test accuracy and MIA accuracy across varying model capacities (one per dataset) and client-side overfitting levels, controlled via the number of training samples per client using a biased gradient estimator.

| Method | IMDB – DistilBERT | | CIFAR-10 – ResNet9 | | MNIST – LeNet5 | |
| | Acc. ($\uparrow$) | MIA Acc. ($\downarrow$) | Acc. ($\uparrow$) | MIA Acc. ($\downarrow$) | Acc. ($\uparrow$) | MIA Acc. ($\downarrow$) |
|---|---|---|---|---|---|---|
| FedAvg | $78.11 \pm 1.42$ | $80.13 \pm 2.11$ | $39.23 \pm 0.76$ | $94.03 \pm 0.69$ | $89.95 \pm 0.77$ | $68.65 \pm 1.63$ |
| FedAvg ($\epsilon_1, \delta$)-LDP | $54.39 \pm 0.57$ | $51.75 \pm 2.69$ | $10.78 \pm 0.36$ | $59.17 \pm 1.21$ | $29.02 \pm 1.20$ | $58.81 \pm 0.95$ |
| FedAvg ($\epsilon_2, \delta$)-LDP | $55.00 \pm 1.10$ | $52.69 \pm 2.74$ | $20.62 \pm 0.57$ | $67.34 \pm 1.46$ | $68.77 \pm 2.09$ | $60.33 \pm 1.42$ |
| SoteriaFL ($\epsilon_1, \delta$) | $54.90 \pm 0.74$ | $52.66 \pm 2.73$ | $10.03 \pm 0.01$ | $56.30 \pm 0.84$ | $14.23 \pm 1.39$ | $56.25 \pm 0.95$ |
| SoteriaFL ($\epsilon_2, \delta$) | $55.40 \pm 1.88$ | $53.44 \pm 2.62$ | $11.70 \pm 0.75$ | $60.69 \pm 0.89$ | $65.11 \pm 1.88$ | $57.16 \pm 1.16$ |
| PriPrune ($p_1$) | $76.52 \pm 1.08$ | $73.00 \pm 2.30$ | $37.12 \pm 0.75$ | $81.59 \pm 2.17$ | $91.11 \pm 0.23$ | $57.83 \pm 1.63$ |
| PriPrune ($p_2$) | $71.62 \pm 1.30$ | $66.66 \pm 2.24$ | $25.32 \pm 1.13$ | $64.16 \pm 1.81$ | $61.29 \pm 1.16$ | $55.14 \pm 1.39$ |
| PriPrune ($p_3$) | $65.82 \pm 1.90$ | $61.22 \pm 2.10$ | $13.91 \pm 0.83$ | $56.35 \pm 1.67$ | $53.99 \pm 1.46$ | $51.72 \pm 1.42$ |
| Shatter | $77.33 \pm 0.87$ | $58.26 \pm 2.03$ | $13.91 \pm 1.77$ | $79.90 \pm 1.74$ | $32.03 \pm 11.54$ | $56.47 \pm 1.68$ |
| ERIS | $77.59 \pm 1.38$ | $57.44 \pm 2.24$ | $39.06 \pm 1.01$ | $74.81 \pm 1.99$ | $90.16 \pm 0.68$ | $55.45 \pm 1.68$ |
| Min. Leakage | $78.11 \pm 1.42$ | $57.02 \pm 2.10$ | $39.23 \pm 0.87$ | $76.22 \pm 1.91$ | $90.06 \pm 0.71$ | $55.20 \pm 1.50$ |

Table 14: Mean test accuracy and MIA accuracy, averaged over varying local sample sizes using a biased gradient estimator. For DP-based methods, $\epsilon \in \{10, 100\}$ on IMDB and $\epsilon \in \{1, 10\}$ on others; for PriPrune, pruning rates are $p \in \{0.1, 0.2, 0.3\}$ on IMDB and $p \in \{0.01, 0.05, 0.1\}$ on others.

### F.9 PARETO ANALYSIS UNDER VARYING PRIVACY CONSTRAINTS

This section complements the analysis in Paragraph 4.2 of the main text with additional details and numerical results. We study how the utility–privacy trade-off evolves under different strengths of privacy-preserving mechanisms and varying numbers of local training samples. Shatter is excluded, as it already fails to converge reliably with 16 samples per client (Figure 4). For DP-based approaches (FedAvg+LDP and SoteriaFL), we vary the privacy budget $\epsilon$ together with the clipping norm $C$ to simulate different protection levels. Following the same configuration, we also evaluate ERIS+LDP, where LDP is applied on top of its native masking mechanism. For pruning-based methods (PriPrune), we vary the pruning rate $p$ to control information flow through gradient sparsification. The exact configurations of these hyperparameters are summarized in Table 15.

Figure 13 shows the resulting utility–privacy trade-off across different numbers of local training samples. As expected, the Pareto frontier shifts toward higher accuracy and lower privacy leakage as clients are assigned more local data. Across all regimes, ERIS dominates the Pareto frontier, contributing the large majority of favorable points, while alternative baselines are mostly dominated.

Table 15 reports the underlying quantitative results, including additional configurations not visualized in the figure. These include finer granularity in both $p$ and $\epsilon$ values, enabling a more exhaustive comparison. The results confirm the trends observed in the main text: ERIS consistently occupies favorable positions in the utility–privacy space, contributing most points along the Pareto frontier. Moreover, when augmented with LDP, ERIS demonstrates further privacy gains with only minor utility losses—outperforming other baselines that suffer substantial degradation as privacy constraints tighten. Overall, this detailed breakdown reinforces that ERIS achieves strong privacy guarantees and high utility, even under stringent privacy budgets and aggressive compression strategies.

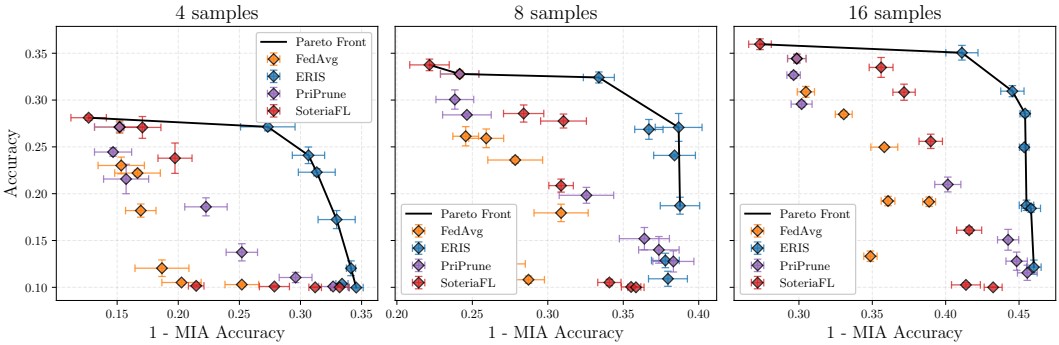

Figure 13: Utility–privacy trade-off on CIFAR-10 under varying strengths of the privacy-preserving mechanisms. Each subplot shows test accuracy vs. MIA accuracy for methods with different client training samples. The Pareto front represents a set of optimal trade-off points.

## G LARGE LANGUAGE MODEL USAGE DISCLOSURE

We disclose the use of Large Language Models (LLMs) in the preparation of this manuscript. Specifically, we used Claude (Anthropic) and GPT-4o solely for writing assistance and polishing. LLMs were used exclusively for:

1. grammar correction and sentence structure improvement,
2. clarity enhancement and readability optimization,
3. consistency in technical terminology and notation, and
4. general writing style refinement.

No content, ideas, analyses, or experimental results were generated by LLMs. All suggestions were carefully reviewed, edited, and approved by the authors before incorporation. The authors retain full responsibility for the entire content, including any errors or inaccuracies.

| Method | 4 samples | | 8 samples | | 16 samples | |
| --- | --- | --- | --- | --- | --- | --- |
| | Accuracy (↑) | MIA Acc. (↓) | Accuracy (↑) | MIA Acc. (↓) | Accuracy (↑) | MIA Acc. (↓) |
| **FedAvg + LDP** | | | | | | |
| No LDP | 27.12% ± 1.20% | 84.80% ± 4.59% | 32.98% ± 0.61% | 75.84% ± 2.85% | 34.43% ± 1.04% | 70.15% ± 1.41% |
| LDP ($\epsilon$=0.001, C=10) | 23.01% ± 2.00% | 84.67% ± 4.24% | 26.13% ± 2.30% | 75.44% ± 1.90% | 30.88% ± 1.30% | 69.53% ± 1.31% |
| LDP ($\epsilon$=0.01, C=5) | 22.20% ± 0.71% | 83.33% ± 4.17% | 25.91% ± 2.23% | 74.08% ± 2.58% | 28.48% ± 0.72% | 66.95% ± 1.29% |
| LDP ($\epsilon$=0.1, C=2) | 18.20% ± 1.58% | 83.07% ± 2.78% | 23.60% ± 0.76% | 72.16% ± 4.06% | 24.97% ± 0.48% | 64.18% ± 2.05% |
| LDP ($\epsilon$=0.3, C=1) | 12.04% ± 1.98% | 81.33% ± 4.94% | 17.95% ± 2.08% | 69.12% ± 4.01% | 19.14% ± 1.12% | 61.13% ± 0.99% |
| LDP ($\epsilon$=0.6, C=1) | 10.52% ± 0.55% | 79.73% ± 3.59% | 12.50% ± 1.42% | 72.88% ± 3.10% | 19.22% ± 1.12% | 63.93% ± 1.05% |
| LDP ($\epsilon$=1.0, C=1) | 10.29% ± 0.40% | 74.80% ± 3.08% | 10.81% ± 0.88% | 71.28% ± 2.37% | 13.32% ± 1.24% | 65.13% ± 1.01% |
| **ERIS + LDP** | | | | | | |
| No LDP | 27.14% ± 0.95% | 72.67% ± 4.99% | 32.21% ± 1.32% | 66.62% ± 2.30% | 35.05% ± 1.75% | 58.89% ± 2.45% |
| LDP ($\epsilon$=0.001, C=10) | 24.10% ± 1.98% | 69.35% ± 2.95% | 26.87% ± 2.37% | 63.30% ± 2.10% | 30.95% ± 1.32% | 55.45% ± 1.77% |
| LDP ($\epsilon$=0.01, C=5) | 22.29% ± 0.86% | 68.67% ± 3.37% | 27.08% ± 3.33% | 59.32% ± 3.47% | 28.55% ± 0.71% | 54.59% ± 0.77% |
| LDP ($\epsilon$=0.1, C=2) | 17.24% ± 2.15% | 67.04% ± 3.40% | 24.09% ± 0.33% | 61.60% ± 3.09% | 24.96% ± 0.63% | 54.64% ± 0.73% |
| LDP ($\epsilon$=0.3, C=1) | 12.03% ± 1.77% | 65.87% ± 0.84% | 18.72% ± 2.03% | 61.24% ± 2.92% | 18.76% ± 1.23% | 54.49% ± 1.10% |
| LDP ($\epsilon$=0.6, C=1) | 10.38% ± 0.16% | 66.61% ± 1.12% | 12.87% ± 1.73% | 62.21% ± 2.06% | 18.43% ± 0.93% | 54.18% ± 1.45% |
| LDP ($\epsilon$=1.0, C=1) | 9.97% ± 0.04% | 65.44% ± 1.27% | 10.91% ± 1.80% | 62.94% ± 2.88% | 12.15% ± 1.69% | 53.99% ± 1.11% |
| **SoteriaFL** | | | | | | |
| No LDP | 28.11% ± 0.61% | 87.33% ± 3.24% | 33.76% ± 1.38% | 77.84% ± 2.92% | 35.96% ± 1.28% | 72.65% ± 1.72% |
| LDP ($\epsilon$=0.001, C=10) | 27.08% ± 2.60% | 82.93% ± 3.39% | 28.56% ± 2.05% | 71.60% ± 2.98% | 33.48% ± 2.37% | 64.40% ± 1.82% |
| LDP ($\epsilon$=0.01, C=5) | 23.79% ± 3.62% | 80.27% ± 3.12% | 27.76% ± 1.67% | 68.96% ± 3.39% | 30.84% ± 1.93% | 62.84% ± 1.70% |
| LDP ($\epsilon$=0.1, C=2) | 10.16% ± 0.26% | 78.53% ± 1.42% | 20.86% ± 1.56% | 69.12% ± 1.81% | 25.59% ± 1.71% | 61.02% ± 1.78% |
| LDP ($\epsilon$=0.3, C=1) | 10.09% ± 0.18% | 72.13% ± 2.75% | 10.52% ± 0.76% | 65.92% ± 1.72% | 16.10% ± 0.78% | 58.40% ± 1.91% |
| LDP ($\epsilon$=0.6, C=1) | 10.00% ± 0.00% | 68.80% ± 1.15% | 10.04% ± 0.08% | 64.48% ± 1.44% | 10.26% ± 0.52% | 58.62% ± 2.20% |
| LDP ($\epsilon$=1.0, C=1) | 10.00% ± 0.00% | 66.80% ± 1.65% | 10.00% ± 0.00% | 64.16% ± 1.20% | 10.00% ± 0.00% | 56.76% ± 1.37% |
| **PriPrune** | | | | | | |
| No Pruning | 27.12% ± 1.20% | 84.80% ± 4.59% | 32.98% ± 0.61% | 75.84% ± 2.85% | 34.43% ± 1.04% | 70.15% ± 1.41% |
| Pruning ($p$=0.0005) | 24.45% ± 1.00% | 85.33% ± 3.40% | 30.06% ± 2.30% | 76.16% ± 2.80% | 32.67% ± 0.79% | 70.36% ± 1.02% |
| Pruning ($p$=0.001) | 21.57% ± 3.51% | 84.27% ± 4.12% | 28.42% ± 0.39% | 75.36% ± 3.60% | 29.57% ± 0.70% | 69.82% ± 1.59% |
| Pruning ($p$=0.005) | 18.60% ± 2.15% | 77.73% ± 3.88% | 19.83% ± 1.89% | 67.44% ± 4.04% | 20.98% ± 1.77% | 59.85% ± 1.99% |
| Pruning ($p$=0.01) | 13.74% ± 2.05% | 74.80% ± 2.87% | 15.20% ± 2.67% | 63.60% ± 3.71% | 15.08% ± 2.49% | 55.75% ± 1.61% |
| Pruning ($p$=0.03) | 11.05% ± 1.22% | 70.40% ± 3.00% | 14.00% ± 3.18% | 62.64% ± 3.00% | 12.79% ± 2.17% | 55.16% ± 1.63% |
| Pruning ($p$=0.05) | 10.09% ± 0.36% | 67.33% ± 2.63% | 12.77% ± 2.52% | 61.68% ± 3.04% | 11.55% ± 1.80% | 54.44% ± 1.50% |

Table 15: Mean test accuracy and privacy leakage (with standard deviation) for various privacy-preserving mechanisms across different local sample sizes. DP-based methods use epsilon $\epsilon$ and clipping norm $C$; PriPrune uses rate $p$.

