# OpenReview forum: "ERIS: Enhancing Privacy and Communication Efficiency in Serverless Federated Learning"
_ICLR.cc/2026/Conference — Submitted to ICLR 2026_

### Official Review · Reviewer_RH9A · 2025-10-30

**Soundness:** 2
**Presentation:** 3
**Contribution:** 2
**Rating:** 4
**Confidence:** 3

**Summary:**

This paper proposes ERIS, a serverless federated learning framework that combines gradient partitioning across multiple client-side aggregators with shifted compression. The authors claim simultaneous improvements in communication efficiency, privacy, and convergence while maintaining equivalence to FedAvg. The work presents theoretical convergence guarantees and information-theoretic privacy bounds, validated through experiments on image and text datasets ranging from small networks to LLMs.

**Strengths:**

S1- The paper provides formal convergence guarantees (Theorem 3.6) and information-theoretic privacy bounds (Theorem 3.7), with detailed proofs showing the approach maintains FedAvg equivalence.
S2-  Evaluation spans multiple datasets (MNIST, CIFAR-10, IMDB, CNN/DailyMail), model scales (62K to 1.3B parameters), and compares against six baselines under different privacy attack scenarios (MIA, DRA).
S3- The paper provides thorough ablation studies on the impact of compression and partitioning separately.
S4- The framework successfully scales to billion-parameter models where many privacy-preserving baselines fail to maintain utility.

**Weaknesses:**

W1- The paper conflates two orthogonal benefits throughout. Each client in ERIS uploads and downloads the same total amount of data (b' bits) as any compression-only method like SoteriaFL. The claimed "communication efficiency" actually refers to reduced distribution time through parallelization, not reduced per-client communication volume. Table 2 is particularly misleading—ERIS shows 1% vs SoteriaFL's 5% primarily due to more aggressive compression (different \omega values), not the partitioning scheme.
W2- The core techniques are from prior work: (a) distributed aggregation exists in Ako (2016), Shatter (2025), C-DFL (2022); (b) shifted compression is directly from Li et al. (2022d). The main contribution is combining these with a straightforward proof that partitioning with disjoint/complete masks preserves FedAvg convergence (Theorem B.1), which is relatively obvious since it merely reorders aggregation operations.
W3- The privacy guarantees only hold against honest-but-curious aggregators who do not observe network traffic beyond their assigned shard. A realistic adversary monitoring network communications can reconstruct the full gradient by observing all client transmissions, reducing privacy to compression-only protection. Under collusion (Corollary D.2), privacy degrades linearly—if adversaries observe all A channels, privacy advantages vanish. The paper needs to discusses these limitations.
W4- Despite aggregators being selected from clients (who "may vary in computational resources and connection stability," Section 5.2), the paper does not provide analysis of aggregator dropout/failure during training rounds,  evaluation of model sensitivity to aggregator unavailability or discussion of aggregator selection strategies.
W5- Figure 6 compares ERIS (A=50, with compression) against FedAvg (no compression), exaggerating gains. Fair comparisons should use identical compression ratios.
W6- Theorem 3.6 shows convergence rate depends on ω (compression), matching SoteriaFL. Partitioning contributes nothing to convergence improvement, it only redistributes computation. This should be stated explicitly.

**Questions:**

Q1- Can you provide a detailed breakdown in Table 2 showing: (a) per-client upload bytes, (b) per-client download bytes, (c) compression ratio, and (d) distribution time, clearly separating gains from compression vs. parallelization?
Q2- What happens when aggregators drop mid-round? Does training continue with (A-1) aggregators, restart the round, or fail? Please provide analysis of robustness to aggregator failures.
Q3- Can you provide experiments where ERIS and SoteriaFL use identical compression ratios (same compression ratio)? This would isolate the benefit of distributed aggregation from compression.
Q4- How do privacy guarantees degrade when adversaries can observe all network traffic (not just content at aggregators)? Can you quantify information leakage in this realistic threat model?

---

> ### Author Response · Authors · 2025-11-20
>
> ## W1&Q1: Breakdown of per-client communication and separation of compression vs. decentralization gains.
>
> We thank the reviewer for the insightful observation and for the suggestion to provide an extended breakdown table. We agree that this clarification is valuable, as it cleanly separates the contributions of compression and decentralized aggregation (via model partitioning) to ERIS’s overall communication efficiency. **We will include the table below reporting: (a) compression rate, (b) per-client upload/download bytes, (c) per-client total bytes communicated in one round, and (d) distribution time.**
>
> ### CNN/Daily Mail — GPT-Neo (1.3B)
> |Method|Comp. Ratio|Upload/Download per-client|Tot.Comm. per-client |Dist. Time|
> |-|-|-|-|-|
> |FedAvg|100%|5.2 GB|10.4 GB|5200 s|
> |Shatter|100%|5.2 GB|36.4 GB|780 s|
> |Ako $(p=5)$|100%|9.36 GB|18.72 GB|936.0 s|
> |Q-DPSGD-1 $(K_n=0.4)$|18.8%|3.51 GB|7.02 GB|351.0 s|
> |PriPrune(0.01)|90%|4.68 GB|9.36 GB|4680 s|
> |PriPrune(0.05)|80%|4.16 GB|8.32 GB|4160 s|
> |PriPrune(0.1)|70%|3.64 GB|7.28 GB|3640 s|
> |SoteriaFL|5%|260 MB|520 MB|260 s|
> |ERIS$(ω_{SoteriaFL})$|5%|234 MB|468 MB|23.4 s|
> |ERIS|1%|46.8 MB|93.6 MB|4.68 s|
>
> ### IMDB  — DistilBERT (69M)
> |Method|Comp. Ratio|Upload/Download per-client|Tot.Comm. per-client|Dist. Time|
> |-|-|-|-|-|
> |FedAvg|100%|268 MB|536 MB|670 s|
> |Shatter|100%|268 MB|2.41 GB|53.6 s|
> |Ako $(p=5)$|100%|1.29 GB|2.57 GB|128.64 s|
> |Q-DPSGD-1 $(K_n=0.4)$|18.8%|482.4 MB|964.8 MB|48.24 s|
> |PriPrune(0.01)|90%|241.2 MB|482.4 MB|603 s|
> |PriPrune(0.05)|80%|214.4 MB|428.8 MB|536 s|
> |PriPrune(0.1)|70%|187.6 MB|375.2 MB|469 s|
> |SoteriaFL|5%|13.4 MB|26.8 MB|33.5 s|
> |ERIS $(ω_{SoteriaFL})$|5%|12.86 MB|25.73 MB|1.29 s|
> |ERIS|0.012%|30.87 KB|61.75 KB|0.003 s|
>
> ### CIFAR-10  — ResNet9 (1.65M)
> |Method|Comp. Ratio|Upload/Download per-client|Tot.Comm. per-client|Dist. Time|
> |-|-|-|-|-|
> |FedAvg|100%|6.6 MB|13.2 MB|33 s|
> |Shatter|100%|6.6 MB|59.4 MB|1.32 s|
> |Ako $(p=5)$|100%|64.68 MB|129.36 MB|6.47 s|
> |Q-DPSGD-1 $(K_n=0.4)$|18.8%|24.25 MB|48.51 MB|2.43 s|
> |PriPrune(0.01)|99%|6.53 MB|13.07 MB|32.67 s|
> |PriPrune(0.05)|95%|6.27 MB|12.54 MB|31.35 s|
> |PriPrune(0.1)|90%|5.94 MB|11.88 MB|29.7 s|
> |SoteriaFL|5%|330 KB|660 KB|1.65 s|
> |ERIS $(ω_{SoteriaFL})$|5%|323.4 KB|646.8 KB|0.03 s|
> |ERIS|0.6%|38.81 KB|77.62 KB|0.004 s|
>
> ### MNIST  — LeNet5 (62KM)
> |Method|Comp. Ratio|Upload/Download per-client|Tot.Comm. per-client|Dist. Time|
> |-|-|-|-|-|
> |FedAvg|100%|248 KB|496 KB|1.24 s|
> |Shatter|100%|248 KB|2.23 MB|0.05 s|
> |Ako $(p=5)$|100%|2.43 MB|4.86 MB|0.24 s|
> |Q-DPSGD-1 $(K_n=0.4)$|18.8%|911.4 KB|1.82 MB|0.09 s|
> |PriPrune(0.01)|99%|245.52 KB|491.04 KB|1.23 s|
> |PriPrune(0.05)|95%|235.6 KB|471.2 KB|1.18 s|
> |PriPrune(0.1)|90%|223.2 KB|446.4 KB|1.12 s|
> |SoteriaFL|5%|12.4 KB|24.8 KB|0.06 s|
> |ERIS $(ω_{SoteriaFL})$|5%|12.15 KB|24.3 KB|0.001 s|
> |ERIS|3.3%|8.02 KB|16.04 KB|0.001 s|
>
> As correctly noted, compression directly reduces the number of transmitted parameters and therefore the per-client communication volume. This effect is independent of ERIS’s aggregation and would similarly benefit any compression-only baseline such as SoteriaFL. However, unlike many privacy-preserving methods—including SoteriaFL—**ERIS’s convergence does not rely on additional perturbations** (e.g., noise, stochastic masking), **allowing it to adopt stricter compression levels for the same convergence behavior**. As a result, ERIS can use more aggressive compression levels while preserving accuracy, which contributes to the gains reported in Table 2 (or above).
>
> Beyond compression, **ERIS’s decentralized aggregation provides a complementary benefit: it removes the single-server bottleneck and distributes the upload/download load across $A$ aggregators**, each handling only a shard of the model. Since bandwidth is a physical constraint of each device, spreading communication over multiple channels yields a substantial reduction in distribution time, especially for large models. **ERIS’s design also leads to a slight reduction in per-client communication volume**. Because a client also acts as the aggregator for exactly one shard, it does not upload or download that shard. For example, in the GPT-Neo experiments with $A=K=10$, each client exchanges only 90% of the model per round (before compression). This gain is visible in the table above when comparing the per-client upload of SoteriaFL with ERIS under the same compression ratio, where the communication drops from 260 MB to 234 MB. Although modest, this reduction is a consistent, measurable effect of partitioning.
>
> Taken together, **compression and decentralized aggregation play complementary roles. This combination improves efficiency and also has a secondary privacy benefit**: while compression alone is not a privacy mechanism, using it jointly with model partitioning reduces per-aggregator visibility of update information, aligning with the privacy goals of ERIS. We will incorporate these clarifications and include the requested extended table in the revised manuscript.

---

> > ### Author Response · Authors · 2025-11-20
> >
> > ## W2: Clarification of ERIS’s contribution, and distinction from prior partition-based methods.
> >
> > We thank the reviewer for this comment, which allows us to extend and clarify our comparison with existing baselines in the Related Work. We agree that centralized shifted compression has previously been adopted in SoteriaFL. However, **the main contribution of ERIS lies in introducing a new decentralized aggregation scheme based on model partitioning that fully preserves the collaborative power (and therefore the utility) of standard centralized FL**. While we adapt shifted compression to our setting, the reference vector and shift are recomputed independently for each shard and updated across multiple aggregators, which is different from centralized compression schemes.
> >
> > Importantly, **although prior works also partition models into shards, their aggregation mechanisms differ substantially and do not preserve full-round collaboration**. For example, Ako splits the model into $p$ parts but exchanges only one part per round, delaying the aggregation of the remaining shards to subsequent rounds. Similarly, Shatter first routes each shard to a small subset of virtual nodes, which then forward those shards to only a limited subset of other virtual nodes (e.g., 4) before aggregation and return to the client. As a result, in both methods each shard is aggregated using updates from only a small fraction of clients rather than the full population, and thus they do not produce a unique global model per round. As confirmed by our results in Table 1, these restricted-collaboration designs lead to significantly weaker convergence.
> >
> > In contrast, ERIS ensures that every shard is aggregated using updates from all participating clients in every round, producing a single global model equivalent to full FedAvg collaboration while distributing the computation and communication load. We will add this clarification in Line 441 of the revised manuscript
> >
> > ## W3&Q4: Discuss ERIS’s limitations under collusion and full network-visibility adversaries.
> >
> > We thank the reviewer for raising this important discussion. We will clarify this aspect in Lines 086, 468, and 334. A passive adversary with full network visibility could indeed access all transmitted updates—and thus reconstruct full gradients—if communication were unencrypted. However, FL protocols, including ERIS, follow the standard assumption of encrypted communication (e.g., TLS), which prevents eavesdroppers from reading raw client updates and is widely adopted as a baseline security guarantee in production FL systems.
> >
> > Regarding collusion and global visibility, we agree that the effective privacy guarantees degrade as more channels become observable. In ERIS, each client update is partitioned across $A$ aggregators, and a non-colluding adversary only sees a fraction of each gradient. As shown in Corollary D.2, **the privacy benefit decreases linearly with the number of colluding or compromised aggregators**: if an adversary observes $c$ out of $A$ shards, it gains visibility proportional to $c/A$. This provides a quantitative measure of privacy degradation under realistic threat models, including network-level observation and partial collusion.
> >
> > **To validate this behaviour empirically, we added a new subsection in Appendix D.2** (together with a figure visualizing the table below, Line ~1620) reporting the effect of honest-but-curious client collusion on MIA accuracy. The results show that **leakage increases smoothly with the collusion group size but remains consistently below the FedAvg baseline, even when 50% of clients collude** (which we find be a highly improbable scenario). Notably, for small collusion groups (e.g., 2%), the leakage remains very close to the minimum-leakage regime.
> >
> > |Collusion (%)|MIA Accuracy|
> > |-|-|
> > |Min. Leakage|54.3 ± 1.7|
> > |2|56.4 ± 1.8|
> > |4|58.9 ± 1.1|
> > |10|63.2 ± 1.2|
> > |20|65.6 ± 1.3|
> > |30|66.8 ± 1.8|
> > |40|68.8 ± 1.2|
> > |50|69.7 ± 1.2|
> > |FedAvg|72.0 ± 3.0|
> >
> > **These findings confirm that the shard-based decomposition in ERIS meaningfully limits the adversary’s advantage, even under strong collusion scenarios**. We will explicitly highlight these limitations and the empirical validation in the revised manuscript.

---

> > > ### Author Response · Authors · 2025-11-20
> > >
> > > ## W4&Q2: Analysis of aggregator dropout, failure handling, and aggregator selection strategies in ERIS.
> > > We thank the reviewer for highlighting this point. We agree that a more detailed description of aggregator selection strategies, along with an explicit analysis of aggregator dropout or failure, would further strengthen the presentation of ERIS. We will add a dedicated subsection in the revised manuscript that formally covers these aspects, and the considerations discussed below.
> > >
> > > *In the current implementation, each aggregator confirms its availability before being assigned a model shard*. However, an aggregator may still fail during the aggregation phase. *A failure-detection mechanism could be incorporated to improve robustness by reassigning the missing shard to another available aggregator, thereby avoiding the loss of that portion of the update*. While such an extension is not included in the present version of ERIS, we appreciate the reviewer’s suggestion and discuss it as a natural enhancement.
> > >
> > > In addition, to assess the impact of aggregator dropout, **we conducted a new experiment in which, at each training round, we randomly drop a specified percentage of aggregators** and thus lose the corresponding update shards. We fixed the number of communication rounds to 200, following the experimental setup used in Table 1. Notably, **the results show that final accuracy remains stable up to a dropout rate of 70%, after which performance begins to degrade**, while the convergence speed (Best Round Loss) slows progressively as dropout increases. These findings indicate that **ERIS is inherently robust to aggregator failures: because each aggregator handles only a disjoint shard, losing a subset of them does not invalidate the entire update** but only reduces the effective magnitude of the global step, allowing learning to continue. In the revised paper (now available in the updated version, Figure 9), we include a dedicated subsection and a figure providing a more fine-grained analysis of this experiment (dropout rates in 5% increments).
> > >
> > > |Drop Rate (%) |Accuracy (Mean ± Std)|Best Round Loss (Mean ± Std)|
> > > |-|-|-|
> > > |0|85.0±0.9|81.0±9.4|
> > > |10|85.1±0.7|87.0±9.8|
> > > |20|85.4±0.9|101.7±13.9|
> > > |30|85.3±0.7|109.0±13.9|
> > > |40|85.2±0.7|122.0±14.2|
> > > |50|85.1±1.0|137.0±22.0|
> > > |60|85.1±0.9|167.3±20.2|
> > > |70|85.2±0.7|192.3±9.5|
> > > |80|83.3±0.5|200.0±0.0|
> > > |90|77.5±0.7|200.0±0.0|
> > >
> > > ## W5: Fair comparison in Fig. 6 by matching compression ratios between ERIS and FedAvg.
> > > We sincerely appreciate this suggestion, as aligning compression settings helps accurately isolate the contribution of ERIS’s decentralized aggregation. In the revised manuscript, **we have updated Figure 6** (visible in the updated version of the paper as Figure 7) **to also include ERIS with $A = 50$ and no compression**, enabling a direct comparison with the uncompressed FedAvg baseline. This update also allows us **to clearly disentangle the effects of compression from the effects of ERIS’s decentralized aggregation on distribution time**.
> > >
> > > The new results show that **even without compression, ERIS achieves substantially lower distribution time than FedAvg**, with the advantage becoming more pronounced for larger models. We also note that ERIS remains more communication-efficient than SoteriaFL, even though SoteriaFL applies a 5% compression ratio, further highlighting the benefits of ERIS’s decentralized aggregation. We will include this figure and its corresponding discussion in the final version.
> > >
> > > ## W6. Clarification on the role of partitioning in convergence
> > > We thank the reviewer for this observation. This is correct: as shown in Section B (“Convergence of ERIS-Base”), **model partitioning does not affect convergence**. ERIS’s aggregation step is algebraically identical to centralized FedAvg, and therefore introduces no approximation error. Consequently, the convergence rate in Theorem 3.6 depends only on the compression variance parameter, the learning rate, and the variance of the gradient estimator, and—unlike SoteriaFL—does not include any term that grows with $T$.
> > >
> > > Partitioning is instead crucial for the privacy and communication properties of ERIS. As formalized in Theorem 3.7, sharding reduces the visibility of each client update to any potentially malicious participant. Moreover, partitioning enables the decentralized aggregation scheme that balances communication load across aggregators. We will make this distinction explicit in the revised manuscript, in Line 214, Section 3.3.

---

> > > > ### Author Response · Authors · 2025-11-20
> > > >
> > > > ## Q3: Comparison between ERIS and SoteriaFL under identical compression rate
> > > > We thank the reviewer for this suggestion. We agree that evaluating ERIS and SoteriaFL under identical compression ratios is a valuable way to isolate the benefit of decentralized aggregation from the effect of compression. Following this recommendation, **we reran all experiments from Table 1 (model utility) and Table 2 (communication efficiency) using a fixed 5% compression rate for ERIS, matching SoteriaFL**. These results are now integrated into the revised manuscript and are also reported below. Tables 6–12 have been updated accordingly to reflect the new single-sample-per-client experiments.
> > > >
> > > > ### Table: Model Utility and Privacy Metrics
> > > > |Method|CNN/DailyMail–GPT-Neo ROUGE|CNN/DailyMail–GPT-Neo MIA Acc.|IMDB–DistilBERT Accuracy|IMDB–DistilBERT MIA Acc.|CIFAR-10–ResNet9 Accuracy|CIFAR-10–ResNet9 MIA Acc.|MNIST–LeNet5 Accuracy|MNIST–LeNet5 MIA Acc.|
> > > > |-|-|-|-|-|-|-|-|-|
> > > > |**ERIS**|32.84 ± 1.07|69.62 ± 3.69|79.60 ± 1.03|56.51 ± 2.04|34.60 ± 1.12|60.47 ± 1.69|88.87 ± 0.49|55.77 ± 1.52|
> > > > |**SoteriaFL**|25.40 ± 0.70|52.14 ± 2.97|54.24 ± 0.15|51.25 ± 1.19|17.18 ± 0.24|58.83 ± 0.56|57.27 ± 0.88|57.13 ± 0.56|
> > > >
> > > > ### Table: Communication Efficiency
> > > >  *While tables for Communication Efficiency are reported in the first answer (W1 & Q1).*
> > > >
> > > > From the model-utility results, **we observe that ERIS preserves similarly strong model utility even with fixed compression rate (5%), significantly outperforming SoteriaFL across all datasets** and approaching the idealized lower-bound privacy leakage reported in Table 1. Regarding communication efficiency, the updated results confirm that **ERIS’s improvements are not solely due to compression. A substantial fraction of the gains arises from its distributed aggregation**, which removes the single-server bottleneck and redistributes upload/download load across multiple aggregators.
> > > >
> > > > Finally, we refer the reviewer to our response to W5, where we also report the results for ERIS without compression and baselines with increasing number of clients or model parameters (Fig. 7), further isolating and demonstrating the benefit of decentralized aggregation.

---

### Official Review · Reviewer_5ugv · 2025-10-31

**Soundness:** 2
**Presentation:** 3
**Contribution:** 2
**Rating:** 4
**Confidence:** 3

**Summary:**

This paper proposes ERIS, a serverless federated learning (FL) framework designed to simultaneously improve communication efficiency, scalability, and privacy. The system replaces the central server with multiple decentralized aggregators, and combines this with a shifted gradient compression mechanism and gradient partitioning.

**Strengths:**

1. The paper provides clear utility and privacy analysis.

2. It is a new idea to have gradient partitioning scheme across multiple aggregators.

**Weaknesses:**

1. The paper claimed a few "the first": "ERIS is the first FL framework to simultaneously achieve decentralized aggregation, strong communication efficiency, and provable information-theoretic privacy guarantees without sacrificing model utility. ERIS is also the first to
extend privacy-enhancing federated training to modern LLMs, demonstrating feasibility at scale where prior methods fail to preserve utility and efficiency." I think these claims are ambiguous because no proof or metrics to show them. For example, I suppose there are lots of works for privacy-enhancing federated training even to LLMs.

2. The need for each client and aggregator to track the shifting reference vectors introduces extra memory and synchronization complexity.

3. I have concerns about theorem 3.6 as it is not tight. For example, if we use SGD as the inner optimizer, Eq(6) shows that it converges to an error that is independent to learning rate. This is not a classic rate, different from existing decentralized optimization and federated optimization. Furthermore, a table of convergence rate comparison with existing baseline algorithms in decentralized optimization and federated optimization would be appreciated.

4. The paper attempts to deliver multiple contributions simultaneously, which makes the overall narrative somewhat overwhelming. For example, the claimed communication efficiency is mainly derived from shifted compression, a technique that is not new. As a result, the contribution in this dimension may appear incremental unless more concrete empirical or theoretical advantages are demonstrated.

**Questions:**

The contributions are about communication and privacy. I wonder if the authors can provide direct comparisons in each dimension with previous works. I suppose there are a lot baselines in decentralized optimizaiton and federated optimization.

---

> ### Author Response · Authors · 2025-11-20
>
> ## Q1: Strong ‘first’ claims that are insufficiently motivated with respect to prior work in these areas (Part 1)
> We thank the reviewer for pointing this out. We understand the concern regarding the scope of our ‘first’ claims, and we welcome the opportunity to clarify their intended meaning. Our intention was not to be dismissive of existing work—particularly on privacy enhancing FL for LLMs—and we agree that the current version would benefit from a clearer formulation and comparison. We will clarify it in the revised manuscript.
>
> **For the first claim**, our intention was not to suggest that ERIS is the first privacy-enhancing FL framework overall, but rather to highlight a more specific combination of properties within the design space we consider. As we now discuss in detail in our response Q1 to Reviewer u1h3, prior work typically satisfies at most two of the three axes we emphasize:
> 1. *Cryptographic-based mechanisms*, including secure multiparty computation (SMC) and homomorphic encryption (HE). These provide strong guarantees but typically incur additional communication or computation overhead, because **SMC requires transmitting multiple masked shards of each update, while HE forces aggregation to be performed over costly encrypted arithmetic**. An example is LiPFed [1], which needs to increases communication to provide privacy: each client must upload $l$ model shares instead of a single model, and communicate with multiple edge nodes.
> 2. *Gradient-perturbation mechanisms*, such as DP or sparsification/pruning, which **inject noise into the updates and thus degrade inevitably utility**. A promising example is SoteriaFL, which combines strong communication compression with formal DP guarantees. However, it remains centralized and relies on noise injection, which harms utility relative to the non-private baseline.
>
> In our work we also evaluate a niche of work for similarity to ERIS, which partition the model, such as Ako and Shatter. However, they also differ substantially from ERIS in both privacy, and collaboration (i.e., aggregation). Ako splits the model into $p$ parts but exchanges only one part per round, delaying the aggregation of the remaining shards to subsequent rounds. Shatter first routes each shard to a small subset of virtual nodes, which then forward those shards to only a limited subset of other virtual nodes (e.g., 4) before aggregation and return to the client. The **communication cost increases** (proportionally to the virtual-node degree) compared to standard decentralized learning (see Tab. 2). As a result, in both methods each shard is aggregated using updates from only a small fraction of clients rather than the full population, and thus they do not produce a unique global model per round. In our experiments (Tab. 1), these **restricted-collaboration designs converge more slowly and reach lower accuracy than ERIS and FedAvg**.
>
> By contrast, ERIS **(i) is fully serverless and decentralized balancing network load; (ii) matches or exceeds the communication efficiency of compression-only baselines** (since it does not rely on additional noise-based mechanism); and **(iii) provides an explicit information-theoretic user-level leakage bound stemming from its sharded architecture, while preserving the exact FedAvg iterate sequence** (no utility loss from the privacy mechanism itself). In the revised version, we will revise the introduction to explicitly contrast ERIS with previous SOTA methods along the three axes above, making our contribution and the scope of our claim fully transparent.
>
> **For the second claim** (*first to extend privacy-enhancing federated training to modern LLMs*), we fully agree that there already exists a growing body of work on privacy-enhancing federated and DP training for LLMs. However, **the majority of recent methods in this space adopts parameter-efficient fine-tuning**, especially LoRA, which is more manageable in FL, and typically combines FL with DP or cryptographic mechanisms on the LoRA adapters [2—6]. These methods are complementary but not directly comparable to our setting: ERIS targets full-model training of an LLM, rather than LoRA-style adapter fine-tuning.
>
> —_The response continues in the next comment_—
>
> ---
> References:
> 1. S. Meng, et al. Secure decentralized aggregation to prevent membership privacy leakage in edge-based federated learning, IEEE TNSE (2024)
> 2. Y. Sun et al. Improving LoRA in Privacy-Preserving Federated Learning, ICLR (2024)
> 3. R. Singhal et al. Fed-SB: A Silver Bullet for Extreme Communication Efficiency and Performance in (Private) Federated LoRA Fine-Tuning, ICML Workshop (2025).
> 4. X. Liu, et al. Differentially Private Low-Rank Adaptation of Large Language Model Using Federated Learning, ACM TMIS (2025)
> 5. J. Zhu, et al. Promoting Data and Model Privacy in Federated Learning through Quantized LoRA, EMNLP (2024)
> 6. H. Xu et al. DP-FedLoRA: Privacy-Enhanced Federated Fine-Tuning for On-Device Large Language Models (2025)

---

> > ### Author Response · Authors · 2025-11-20
> >
> > ## Q1: Strong ‘first’ claims that are insufficiently motivated with respect to prior work in these areas (Part 2)
> > **To the best of our knowledge, we have not found prior work that performs a privacy analysis across multiple privacy-enhancing methods in the regime of full-model federated LLM training, comparing empirical evaluation using privacy attacks at scale**. The closest work we are aware of is the user-level DP fine-tuning of LLMs by Charles et al. [7], which trains large models with rigorous DP guarantees, but focuses on the DP accounting and utility trade-offs, without benchmarking empirical leakage against alternative privacy-enhancing mechanisms (e.g., against MIA, DRA, inference-time leakage attacks). Similarly, PriFFT [8] proposes a secret-sharing–based framework for privacy-preserving federated fine-tuning of LLMs but does not provide such an empirical leakage comparison and is evaluated on smaller architectures rather than billion-parameter models. In parallel, there are complementary works that analyze privacy attacks in federated or collaborative fine-tuning of LLMs [9,10,11], without proposing or benchmarking alternative privacy-enhancing training schemes. In any case, we would be very grateful for any additional studies the reviewer recommends that we should consider in our analysis.
> >
> > In our original wording we implicitly referred to this more specific regime of full-parameter federated training for LLMs with explicit privacy analysis, but we acknowledge that this was ambiguous. In the revised version, we will therefore rephrase the sentence as: “*We extend the analysis of privacy-enhancing federated full-parameter training to modern LLMs, demonstrating ERIS’s feasibility at scale where prior methods fail to preserve utility and efficiency.*”
> > Our main intent has always been to highlight that **ERIS’s lightweight, serverless design and communication efficiency scale to full-model federated fine-tuning of a 1.3B-parameter GPT-style model** while (i) preserving FedAvg-level utility, (ii) achieving strong communication savings, and (iii) providing an explicit information-theoretic user-level leakage bound.
> >
> > ## Q2: Memory and synchronization overhead of tracking reference vectors
> > We agree that decentralizing FL inevitably shifts some memory and computational load from the server to the participating clients and aggregators. However, in **ERIS these costs are kept minimal: unlike other decentralized FL frameworks, ERIS introduces no redundant operations**. The work typically performed by a central server is simply partitioned across the $A$ aggregators, rather than replicated.
> > *For example, with $A = 50$ and a compression rate $p = 3\%$, each aggregator receives on average only $p/A \approx 0.06\%$ of the full model from each client per roun. Likewise, each aggregator stores only $1/A$ of the reference vector, corresponding to its shard*. Compared to decentralized FL baselines where each client may need to aggregate or exchange full model parameters from others, ERIS is significantly more lightweight.
> > Regarding synchronization, decentralization naturally increases the need for coordination. However, this comes with the benefit of improved system robustness, as the architecture no longer relies on a single server and avoids a single point of failure. We will clarify these points in the revised manuscript in Line 465.
> >
> > ---
> > References:
> >
> > 7. Z. Charles et al. Fine-Tuning Large Language Models with User-Level Differential Privacy (2024)
> > 8. Z. You, et al. PriFFT: Privacy-preserving Federated Fine-tuning of Large Language Models via Hybrid Secret Sharing (2025)
> > 9. M. N. Vu et al. Analysis of Privacy Leakage in Federated Large Language Models, AISTATS (2024)
> > 10. M. R. U. Rashid et al. FLTrojan: Privacy Leakage Attacks against Federated Language Models through Selective Weight Tampering (2025)
> > 11. F. Wang et al. Data Reconstruction and Protection in Federated Learning for Fine-Tuning Large Language Models," IEEE TBD (2024)

---

> > > ### Author Response · Authors · 2025-11-20
> > >
> > > ## Q3.1: Tightness of Theorem 3.6 and its dependence on the learning rate
> > >
> > > We thank the reviewer for raising this point. **To clarify these concepts, in the revised manuscript we have (i) added Corollary C.6 in Appendix C.1 which specializes Theorem 3.6 to ERIS with SGD** (now available in the revised manuscript), and **(ii) will add an explicit sentence around Line 252 clarifying the role of the learning rate**.
> > > Theorem 3.6 provides an upper bound on the average stationarity error under Assumptions 3.1–3.2. As is common with potential-function analyses for compressed SGD, this bound is not tight with respect to the learning rate. For a constant step size $\lambda_t \equiv \lambda$, Eq. (6) yields an asymptotic term of order $O(C_2/(\lambda L))$. Thus, **the bound depend inversely on the learning rate $\lambda$**. When we instantiate Theorem 3.6 with vanilla SGD in Corollary C.6, we now obtain the bound
> > > $$
> > >     \frac{1}{T}\sum_{t=0}^{T-1} \mathbb{E}\bigl\|\nabla f(\mathbf{x}^t)\bigr\|^2
> > >     \le
> > >     \frac{2\Phi_0}{\lambda T}
> > >     +
> > >     \frac{3\beta (m-b) G^2}{(1+\omega)L \lambda m b},
> > > $$
> > > which explicitly shows the dependence on the learning rate $\lambda$. In addition, Corollary C.6 further fixes $\lambda$ to the maximal admissible value $\lambda_1$ from Eq. (25) to obtain a simpler closed-form expression in terms of problem parameters.
> > > We agree that our generic bound is conservative and not learning-rate-optimal—this is a limitation of the analysis rather than of ERIS itself. Our goal with Theorem 3.6 is not to optimize the dependence on the stepsize, but to show that ERIS maintains the same order of convergence rate as standard shifted-compression methods (e.g., SoteriaFL) while strictly relaxing their admissible convergence conditions.
> > >
> > > ## Q3.2: A table of convergence rate comparison would be appreciated
> > > We genuinely appreciate the reviewer’s valuable suggestion, as a convergence-rate comparison table is clearly useful to contextualize ERIS alongside existing decentralized and federated optimization methods. In the revised manuscript, we have added a dedicated section in Appendix C.5 (Utility Comparison) and introduced Corollary C.6, which formalizes the non-DP utility bound used in the table. Below, we provide a concise version of this section. **The table below summarizes the asymptotic utility/accuracy guarantees** (average squared gradient norm after $T$ rounds) of representative differentially-private FL/DSGD methods, compared against the non-DP utility bound achieved by ERIS-SGD. Existing baselines typically combine compression with DP noise (either central DP or local DP), which worsens convergence as a function of $(\varepsilon,\delta)$, model dimension $n$, and compressor variance $(1+\omega)$. In contrast, **ERIS-SGD does not need to inject DP noise by default and therefore achieves a dimension-free, fully sharded, decentralized convergence guarantees of**
> > > $$\tilde{\mathcal{O}}\left(
> > >     \dfrac{\sqrt{1+\omega}}{\sqrt{K}m} \right)$$
> > >
> > > matching the dependence on $K$ and $(1+\omega)$ of SoteriaFL-style shifted compression while operating in a serverless architecture. When the same LDP mechanism is added to both methods (Fig. 4 in the paper), ERIS and SoteriaFL converge to the same $(\varepsilon,\delta)$-DP utility regime (where $\tau:=(1+\omega)^{3/2} / \sqrt{K}$); however, ERIS requires less injected noise thanks to its inherent privacy amplification through sharded aggregation.
> > >
> > > ### Table: Asymptotic Utility / Accuracy Bounds
> > >
> > > |Algorithm|Privacy|Utility / Accuracy Bound|
> > > |-|-|-|
> > > |Distributed DP-SRM [12]|$(\varepsilon,\delta)$-DP|$\tilde{\mathcal{O}}\left( \frac{\sqrt{n\log(1/\delta)}}{K m \varepsilon} \right)$|
> > > |SDM-DSGD [13]|$(\varepsilon,\delta)$-LDP|$\tilde{\mathcal{O}}\left( \frac{\sqrt{n\log(1/\delta)}}{\sqrt{K} m \varepsilon} \right)$|
> > > |Q-DPSGD-1 [14] | $(\varepsilon,\delta)$-LDP|$\tilde{\mathcal{O}}\left( \frac{ \left(\frac{\tilde{\nu}^2}{K} + \frac{1}{m}\right)^{2/3} (n\log(1/\delta))^{1/3} }{ m^{2/3}\varepsilon^{2/3} } \right)$ |
> > > |CDP-SGD [15]|$(\varepsilon,\delta)$-LDP|$\tilde{\mathcal{O}}\left( \frac{ \sqrt{(1+\omega)\,n\log(1/\delta)} }{ \sqrt{K} m \varepsilon } \right)$|
> > > |SoteriaFL-SGD|$(\varepsilon,\delta)$-LDP|$\tilde{\mathcal{O}}\left( \frac{ \sqrt{(1+\omega)\,n\log(1/\delta)} }{ \sqrt{K} m \varepsilon } (1+\sqrt{\tau}) \right)$ |
> > > |**ERIS-SGD (no DP)**|—| $\tilde{\mathcal{O}}\left( \frac{ \sqrt{1+\omega} }{ m \sqrt{K} } \right)$|
> > >
> > > ---
> > > References:
> > >
> > > 12. L. Wang, et al. Efficient Privacy-Preserving Stochastic Nonconvex Optimization, UAI (2023)
> > > 13. X. Zhang, et al. Private and Communication-Efficient Edge Learning: A Sparse Differential Gaussian-Masking Distributed SGD Approach, MobiHoc (2020)
> > > 14. J. Ding, et al. Differentially Private and Communication Efficient Collaborative Learning, AAAI (2021)
> > > 15. Z. Li, et al. SoteriaFL: A Unified Framework for Private Federated Learning with Communication Compression, NeurIPS (2022)

---

> > > > ### Author Response · Authors · 2025-11-20
> > > >
> > > > ## Q4: Multiple simultaneous contributions make the narrative feel overwhelming, and the communication-efficiency component appears incremental
> > > > We sincerely thank the reviewer for this comment. We agree on two points: (i) the current narrative foregrounds several contributions at once, and hence it requires clear framing to avoid misunderstandings (ii) shifted compression itself is not new; in our work we extended from SoteriaFL to a decentralized setting. Our intention was never to present this as a novel compressor—indeed, Line 188 in Section 3.2 explicitly states that ‘we extend the shifted compression of SoteriaFL to a distributed setting’. We will make this dependency on SoteriaFL more explicit in the revised version.
> > > >
> > > > In the revision, **we will explicitly reframe the contribution hierarchy so that the main novelty is a serverless, sharded aggregation architecture that exactly preserves FedAvg updates while (i) removing the central bottleneck and (ii) providing information-theoretic user-level privacy amplification without noise injection** or cryptographic overhead. In this framing, **communication efficiency becomes an enabling consequence of this architecture**, rather than a stand-alone “third” contribution. We will revise the introduction and contribution bullets (Lines 57–77) to emphasize the new decentralized aggregation and privacy analysis as the core contribution, and (ii) present communication improvements as a secondary, but practically important, benefit obtained by combining this architecture with a standard shifted compressor.
> > > >
> > > > While we fully acknowledge that shifted compression is not novel, our goal in this dimension is to demonstrate that its combination with ERIS’s sharded architecture yields concrete, non-trivial advantages over both (i) centralized shifted compression (SoteriaFL) and (ii) other partition-based decentralized methods (Ako, Shatter):
> > > > 1. **Decentralized, bottleneck-free aggregation with FedAvg-equivalent updates.**
> > > > ERIS distributes both upload and download across $A$ aggregators, each handling disjoint shards, while still reconstructing the exact FedAvg update. In contrast, Ako and Shatter trade off collaboration for decentralization (each shard aggregates over only a subset of clients), which empirically leads to slower convergence and lower accuracy under the same per-client communication budget (Table 1). *This “no loss of collaboration” property is what allows ERIS to maintain FedAvg-level utility even under aggressive compression*.
> > > > 2. **Empirical gains beyond “just using shifted compression once more”**.
> > > > As detailed in our next answer to Q5 (and W1&Q1 of Reviwer RH9A), we have added communication tables that separate the effects of compression and decentralized aggregation. These results show that *the communication improvements achieved by ERIS go beyond shifted compression alone, since even under matched compression ratios ERIS consistently reduces distribution time* by removing the single-server bottleneck and distributing aggregation across shards. We will explicitly refer to these tables in the main text to clarify that (i) compression is a known building block, but (ii) the combination of sharded aggregation + shifted compression achieves improved utility–distribution-time trade-offs, rather than merely reusing shifted compression.
> > > >
> > > > ## Q5: Can the authors provide direct communication and privacy comparisons with existing FL and DFL baselines? (Part 1)
> > > > We thank the reviewer for this suggestion and agree that explicit comparisons help clarify our contributions along both dimensions.
> > > > ### Communication dimension.
> > > > As also recommended by Reviwer RH9A **we now provide a detailed communication comparison across a broad set of FL and DFL baselines** (FedAvg, Shatter, Ako, Q-DPSGD-1, PriPrune with multiple pruning rates, SoteriaFL, and ERIS variants). Tables below (added to the appendix) report, for each dataset/model pair, the compression ratio, upload/download per client, total per-round communication per client, and distribution time. Among decentralized FL methods, Shatter and Ako both rely on a partition of the model, while Q-DPSGD-1 achieves communication reduction through quantization. We further include PriPrune with 3 different pruning rate and SoteriaFL at 5% compression. Finally, we report results for ERIS both (i) at the same compression ratio as SoteriaFL (to isolate the benefit of our decentralized, sharded aggregation) and (ii) under the more aggressive compression regime adopted in our main experiments.
> > > >
> > > > These tables show that, **for a fixed compression ratio, ERIS achieves substantially lower distribution time than both centralized and decentralized baselines,** and additionally enables much more aggressive compression without loss in accuracy (as reported in the main text).
> > > >
> > > > —_The response continues in the next comment_—

---

> > > > > ### Author Response · Authors · 2025-11-20
> > > > >
> > > > > ## Q5: Can the authors provide direct communication and privacy comparisons with existing FL and DFL baselines? (Part 2)
> > > > > ### CNN/Daily Mail — GPT-Neo (1.3B)
> > > > > |Method|Comp. Ratio|Upload/Download per-client|Tot.Comm. per-client |Dist. Time|
> > > > > |-|-|-|-|-|
> > > > > |FedAvg|100%|5.2 GB|10.4 GB|5200 s|
> > > > > |Shatter|100%|5.2 GB|36.4 GB|780 s|
> > > > > |Ako $(p=5)$|100%|9.36 GB|18.72 GB|936.0 s|
> > > > > |Q-DPSGD-1 $(K_n=0.4)$|18.8%|3.51 GB|7.02 GB|351.0 s|
> > > > > |PriPrune(0.01)|90%|4.68 GB|9.36 GB|4680 s|
> > > > > |PriPrune(0.05)|80%|4.16 GB|8.32 GB|4160 s|
> > > > > |PriPrune(0.1)|70%|3.64 GB|7.28 GB|3640 s|
> > > > > |SoteriaFL|5%|260 MB|520 MB|260 s|
> > > > > |ERIS$(ω_{SoteriaFL})$|5%|234 MB|468 MB|23.4 s|
> > > > > |ERIS|1%|46.8 MB|93.6 MB|4.68 s|
> > > > >
> > > > > ### IMDB  — DistilBERT (69M)
> > > > > |Method|Comp. Ratio|Upload/Download per-client|Tot.Comm. per-client|Dist. Time|
> > > > > |-|-|-|-|-|
> > > > > |FedAvg|100%|268 MB|536 MB|670 s|
> > > > > |Shatter|100%|268 MB|2.41 GB|53.6 s|
> > > > > |Ako $(p=5)$|100%|1.29 GB|2.57 GB|128.64 s|
> > > > > |Q-DPSGD-1 $(K_n=0.4)$|18.8%|482.4 MB|964.8 MB|48.24 s|
> > > > > |PriPrune(0.01)|90%|241.2 MB|482.4 MB|603 s|
> > > > > |PriPrune(0.05)|80%|214.4 MB|428.8 MB|536 s|
> > > > > |PriPrune(0.1)|70%|187.6 MB|375.2 MB|469 s|
> > > > > |SoteriaFL|5%|13.4 MB|26.8 MB|33.5 s|
> > > > > |ERIS $(ω_{SoteriaFL})$|5%|12.86 MB|25.73 MB|1.29 s|
> > > > > |ERIS|0.012%|30.87 KB|61.75 KB|0.003 s|
> > > > >
> > > > > ### CIFAR-10  — ResNet9 (1.65M)
> > > > > |Method|Comp. Ratio|Upload/Download per-client|Tot.Comm. per-client|Dist. Time|
> > > > > |-|-|-|-|-|
> > > > > |FedAvg|100%|6.6 MB|13.2 MB|33 s|
> > > > > |Shatter|100%|6.6 MB|59.4 MB|1.32 s|
> > > > > |Ako $(p=5)$|100%|64.68 MB|129.36 MB|6.47 s|
> > > > > |Q-DPSGD-1 $(K_n=0.4)$|18.8%|24.25 MB|48.51 MB|2.43 s|
> > > > > |PriPrune(0.01)|99%|6.53 MB|13.07 MB|32.67 s|
> > > > > |PriPrune(0.05)|95%|6.27 MB|12.54 MB|31.35 s|
> > > > > |PriPrune(0.1)|90%|5.94 MB|11.88 MB|29.7 s|
> > > > > |SoteriaFL|5%|330 KB|660 KB|1.65 s|
> > > > > |ERIS $(ω_{SoteriaFL})$|5%|323.4 KB|646.8 KB|0.03 s|
> > > > > |ERIS|0.6%|38.81 KB|77.62 KB|0.004 s|
> > > > >
> > > > > ### MNIST  — LeNet5 (62KM)
> > > > > |Method|Comp. Ratio|Upload/Download per-client|Tot.Comm. per-client|Dist. Time|
> > > > > |-|-|-|-|-|
> > > > > |FedAvg|100%|248 KB|496 KB|1.24 s|
> > > > > |Shatter|100%|248 KB|2.23 MB|0.05 s|
> > > > > |Ako $(p=5)$|100%|2.43 MB|4.86 MB|0.24 s|
> > > > > |Q-DPSGD-1 $(K_n=0.4)$|18.8%|911.4 KB|1.82 MB|0.09 s|
> > > > > |PriPrune(0.01)|99%|245.52 KB|491.04 KB|1.23 s|
> > > > > |PriPrune(0.05)|95%|235.6 KB|471.2 KB|1.18 s|
> > > > > |PriPrune(0.1)|90%|223.2 KB|446.4 KB|1.12 s|
> > > > > |SoteriaFL|5%|12.4 KB|24.8 KB|0.06 s|
> > > > > |ERIS $(ω_{SoteriaFL})$|5%|12.15 KB|24.3 KB|0.001 s|
> > > > > |ERIS|3.3%|8.02 KB|16.04 KB|0.001 s|
> > > > >
> > > > > ### Privacy dimension.
> > > > > **Direct privacy comparison is more delicate, since the baselines rely on different mechanisms** (architectural sharding, pruning, quantization, central DP, local DP, etc.), which do not share a single closed-form privacy metric. **For this reason, our main comparison in the paper is empirical, using five representative attacks across two widely studied categories**: Membership Inference Attacks (MIA) and Data Reconstruction Attacks (DRA). Under equal training setups, **ERIS consistently reduces attack success rates compared to non-partitioned baselines at matched or better utility** (see Fig. 3 or Table 1 in the paper).
> > > > > In addition, for DP-based methods we can provide a more direct theoretical comparison under a fixed $(\varepsilon,\delta)$ guarantee. The table in Answer Q3.2 summarizes the asymptotic utility/accuracy bounds (average squared gradient norm after $T$ rounds) for several representative differentially private FL/DSGD methods, and contrasts them with the non-DP utility bound of ERIS-SGD.
> > > > > Adding DP mechanisms degrades utility as a function of $(\varepsilon,\delta)$, model dimension $n$, and compressor variance $(1+\omega)$. In contrast, ERIS-SGD already provides architectural privacy amplification via sharded aggregation and therefore does not require DP noise by default, leading to a dimension-free, fully decentralized convergence bound. When the same LDP mechanism is added to both methods (as in Fig. 4 in the paper), ERIS and SoteriaFL converge to the same $(\varepsilon,\delta)$-DP regime in terms of asymptotic rate, but ERIS can use less noise to reach a given privacy level thanks to its inherent sharding-based amplification.
> > > > >
> > > > > We will clarify these two complementary comparisons in the revised version to make the privacy dimension of our contribution more explicit.

---

### Official Review · Reviewer_u1h3 · 2025-11-02

**Soundness:** 3
**Presentation:** 3
**Contribution:** 3
**Rating:** 6
**Confidence:** 3

**Summary:**

This paper introduces ERIS, a decentralized federated learning framework designed to achieve both communication efficiency and information-theoretic privacy guarantees without relying on a central server. The key idea is to partition model parameters into disjoint shards, each handled by a different client-side aggregator, and to apply a “shifted compression” mechanism that reduces communication cost while limiting information leakage. The authors provide convergence and privacy analyses, showing that ERIS maintains FedAvg-like utility bounds(Thm. 3.6) and scales privacy guarantees with the number of aggregators(Thm. 3.7). Empirical evaluations demonstrate improved privacy–utility–communication trade-offs compared to existing decentralized or privacy-preserving FL baselines.

**Strengths:**

- The paper presents a clear and well-motivated problem setup addressing privacy and communication challenges in decentralized FL.
- The overall framework is interesting: partitioning parameters across aggregators leads to linear scalability and better privacy by design.
- This paper includes solid theoretical analysis, providing convergence results and a clean information-theoretic privacy bound.
- The paper systematically evaluates both MIAs and DRAs, includes a Pareto analysis, and reports per-round communication/time numbers.

**Weaknesses:**

* The paper states ERIS is the first framework to simultaneously provide decentralized aggregation, communication efficiency, and provable information-theoretic privacy without sacrificing utility. However, it seems that the claims is too strong. Baslines in the experiments such as Shatter and SoterialFL as well as the other prior works (e.g., [Shen et al]) should be discussed and compared before the claim.
* - Shen, Meng, et al. "Secure decentralized aggregation to prevent membership privacy leakage in edge-based federated learning." IEEE Transactions on Network Science and Engineering 11.3 (2024): 3105-3119.

- Algorithm 1 and §3.2 say masks can be predefined/shared or dynamically sampled by each client. For an aggregator-specific shard, it’s more natural that the aggregator (or a coordinator) samples/defines $m_{(a)}^t$ and broadcasts them to all clients per round to guarantee consistent slicing across clients. As written, “dynamically sampled by each client” (line 202) invites inconsistent partitions unless there is a synchronization step. Please clarify the intended control flow (who samples? when? how are masks synchronized/broadcast?)
- The main text analyzes an honest-but-curious non-colluding adversary who only sees a shard; Appendix D mentions an extension to colluding adversaries and §5.2 acknowledges that privacy benefits diminish with collusion, scaling with the number of colluding nodes (Corollary D.2). This is important enough to surface earlier: how much privacy remains if a small constant fraction of aggregators collude? What if an aggregator colludes with a subset of clients?
- The experiments show performance degradation as $A$ grows (Figure 2) while Thm. 3.6 is agnostic to $𝐴$. An intuitive explanation is missing. Is this because increasing$A$ shrinks each shard’s dimensionality and, together with compression, increases effective variance and error-feedback lag on each shard’s reference vector? That can slow optimization or accumulate bias in finite rounds even if the asymptotic bound does not expose an explicit $A$ term.

**Questions:**

- Please see the weaknesses.
- If $A \rightarrow n$ (i.e., per-coordinate sharding), it there a stability/variance blow-up without increasing bandwidth? Any guidance on a practical range of $𝐴$ w.r.t model dimension and client count?

---

> ### Author Response · Authors · 2025-11-20
>
> ## Q1: The ‘first framework’ claim is not adequately justified before comparing ERIS with prior works such as Shatter, SoteriaFL, and LiPFed.
> We understand the reviewer’s concern regarding the strength of our ‘first framework’ claim, and we appreciate the opportunity to clarify its intended scope. Accordingly, we provide additional comparison with respect to prior works such as Shatter, SoteriaFL, and LiPFed [Shen et al.]. In high-level terms, user-level privacy in FL has mainly been pursued via three families of techniques:
> 1. *Cryptographic-based mechanisms*, including secure multiparty computation (SMC, e.g., secret sharing) and homomorphic encryption (HE). These provide strong guarantees but typically incur additional communication or computation overhead, because SMC requires transmitting multiple masked shards of each update, while HE forces aggregation to be performed over costly encrypted arithmetic.
> 2. *Hardware-based mechanisms*, such as Trusted Execution Environments (TEEs), which require specialized hardware and introduce different trust and deployment assumptions (reason why we excluded from our study).
> 3. *Gradient-perturbation mechanisms*, such as local differential privacy (LDP) or sparsification/pruning, which inject noise into the updates and thus degrade inevitably utility.
>
> LiPFed is an excellent representative of the first family. It combines additive secret sharing with a consortium blockchain, allowing each client to protect its update using lightweight SMC. Concretely, after local training a client splits its model $w$ into $l$ additive shares, $w = w_1 + w_2 + \dots + w_l,$ and sends each share to a different edge node. Any individual share is information-theoretically useless, since only the sum of all $l$ shares cancels the masking noise introduced by the secret sharing procedure. Because collocating all shares at a single aggregator would reveal the client’s update, LiPFed then performs a two-stage secure aggregation: each edge node aggregates its corresponding share across all users, and only the combination of all aggregated shares can reconstruct the global FedAvg update. However, _this design increases communication cost_: each client must upload $l$ model shares instead of a single model, and communicate with multiple edge nodes. Moreover, if even a single share from a user were lost during aggregation, the masking terms would not fully cancel. These properties explain why, despite its strong privacy guarantees, we do not consider LiPFed to satisfy our “communication-efficient” criterion.
>
> By contrast, SoteriaFL does explicitly combine communication compression with LDP. It therefore simultaneously improves communication and provides formal privacy guarantees, but it remains server-centric, and its guarantees rely on DP noise injection—so some _loss in utility compared to the non-private baseline_ is unavoidable.
> The model-partitioning baselines we evaluate, Ako and Shatter, also differ substantially from ERIS in both privacy, and collaboration (i.e., aggregation mechanism). Ako splits the model into $p$ parts but exchanges only one part per round, delaying the aggregation of the remaining shards to subsequent rounds. Shatter first routes each shard to a small subset of virtual nodes, which then forward those shards to only a limited subset of other virtual nodes (e.g., 4) before aggregation and return to the client. The communication cost increases (by a factor depending on the degree of the virtual-node graph) compared to standard decentralized learning (see Table 2 or W1&Q1 answer to reviewer RH9A for a more detailed table). As a result, in both methods each shard is aggregated using updates from only a small fraction of clients rather than the full population, and thus they do not produce a unique global model per round. In our experiments (Table 1), these _restricted-collaboration designs converge more slowly and reach lower accuracy_ than ERIS and FedAvg under the same total communication budget.
>
> ERIS is designed to simultaneously satisfy all three desiderata in a decentralized FL setting:
> 1. **Information-theoretic protection**. Each aggregator observes only a random partition of every client’s gradient, providing user-level privacy without relying on cryptographic hardness or noise injection.
> 2. **Communication efficiency**. Decentralized model partitioning is combined with structured, shifted compression, which both balances network load and reduces the total number of transmitted parameters per client.
> 3. **Full-round collaboration and utility preservation.** Each global update in ERIS reconstructs exactly the FedAvg update—that is, the average of all client updates—ensuring no loss of utility relative to standard FL.
>
> We will revise the introduction to explicitly contrast ERIS with LiPFed, Shatter, Ako, and SoteriaFL along the three axes above. This additional context will make the contribution and the scope of our claim fully transparent and properly motivated.

---

> > ### Author Response · Authors · 2025-11-20
> >
> > ## Q2: Clarification on how masks are sampled and synchronized across clients
> > We thank the reviewer for pointing out this source of confusion. The sentence in Line 202 (“_dynamically sampled by each client at each round_”) was indeed too concise and may suggest that clients sample masks independently, which would lead to inconsistencies. Our intended meaning was simply that **the mask assignments can change across rounds, not that each client samples them in isolation** (as opposed to Option 1, where masks are pre-defined via a deterministic partition).
> >
> > To ensure that the masks remain disjoint and complete at every round, a coordinator is required to generate or update the masks and distribute them to all clients. This is also how it is implemented in our code: a lightweight coordinator samples (or predefines) the masks and synchronizes them across all participants before each round begins. We will revise the corresponding sentence in the manuscript to make this control flow explicit.
> >
> > ## Q3: How much privacy remains when a small constant fraction of aggregators or clients collude?
> > We thank the reviewer for raising this important point. We agree that the impact of collusion deserves clearer emphasis and empirical validation earlier in the paper. In ERIS, each client update is partitioned across $A$ aggregators, so a non-colluding adversary only sees a fraction of the gradient. However, as formalized in Corollary D.2, **the privacy amplification offered by sharding decreases linearly with the number of colluding entities**: if an adversary observes $c$ out of $A$ shards, its visibility increases proportionally to $c/A$. This applies both when aggregators collude among themselves and when a subset of clients colludes with an aggregator.
> >
> > To strengthen this analysis, **we explicitly conducted a new experiment** (now reported in Appendix D.2, ~Line 1620, accompanied by a figure) **measuring MIA accuracy as increasing fractions of honest-but-curious clients collude**. This setup directly simulates the visibility gained by an adversary that controls a constant fraction of the shards. The results show that leakage increases with the collusion group size, but remains consistently below the FedAvg baseline—even when 50% of clients collude (which we find be a highly improbable scenario)—and is very close to the minimum-leakage baseline for small collusion groups (e.g., 2%):
> >
> > |Collusion (%)|MIA Accuracy|
> > |-|-|
> > |Min. Leakage|54.3 ± 1.7|
> > |2|56.4 ± 1.8|
> > |4|58.9 ± 1.1|
> > |10|63.2 ± 1.2|
> > |20|65.6 ± 1.3|
> > |30|66.8 ± 1.8|
> > |40|68.8 ± 1.2|
> > |50|69.7 ± 1.2|
> > |FedAvg|72.0 ± 3.0|
> >
> > These findings confirm that **ERIS retains meaningful privacy amplification under realistic collusion levels**, especially when only a small constant fraction of participants collude and continues to outperform the FedAvg baseline even under stronger collusion scenarios. We will surface this discussion earlier in the manuscript and explicitly reference the empirical validation.
> >
> > ## Q4: Clarification on why Fig. 2 shows degradation with increasing $A$ while Theorem 3.6 is agnostic to $A$
> > We thank the reviewer for the question. We would like to clarify that **Fig. 2 reports Membership Inference Attack (MIA) accuracy as a function of the number of aggregators $A$** (and compression rate), and therefore reflects privacy leakage, not model utility.
> > By contrast, Theorem 3.6 characterizes utility convergence. As shown in Section B (“_Convergence of ERIS-Base_”), ERIS’s aggregation rule is exactly identical to the centralized FedAvg update, with no approximation error introduced by partitioning. This is why Theorem 3.6 is agnostic to $A$. Therefore, **increasing $A$** reduces the amount of information visible to each aggregator, which **lowers the attack success rate—but does not impact convergence**, consistent with our theory. We will clarify this distinction in the revised manuscript.

---

> > > ### Author Response · Authors · 2025-11-20
> > >
> > > ## Q5: Stability of ERIS as $A\rightarrow n$.
> > > We thank the reviewer for this interesting question. From an optimization point of view, **ERIS does not suffer from a stability or variance blow-up as the number of aggregators $A$ increases**—even in the extreme case of per-coordinate sharding $(A \to n)$. In Appendix B (Theorem B.1) we show that ERIS–Base (without compression, to isolate the effect of aggregation) produces exactly the same iterate sequence as FedAvg for any $A \ge 1$:
> > > $\mathbf{x}^{t+1}=\mathbf{x}^{t} -\lambda_t\sum_{a=1}^A\bar{\mathbf{g}}_{(a)}^t = \mathbf{x}^{t} - \lambda_t \tilde{\mathbf g}^t$. Consequently, the gradient estimator has the same mean and variance as in FedAvg, and the per-round uplink/downlink bandwidth per client is unchanged; only the network load is distributed across $A$ aggregators. In the compressed setting, Theorem 3.6 (and the newly added Corollary C.6 for SGD) similarly shows that the convergence rate is governed by the variance of the (compressed) gradient estimator and does not depend on $A$.
> > >
> > > From a practical standpoint, taking $A$ very large mainly introduces systems-level trade-offs rather than optimization-level ones: **more aggregators yield finer-grained load balancing and stronger per-aggregator privacy** (our mutual-information bound decreases as $1/A$), but also more coordination and a higher chance of aggregator failures. In our experiments we use $A$ up to 50, which already provides strong privacy and load balancing; larger values are possible when enough stable devices are available, but offer diminishing returns. In the revised version we will add a short remark to clarify these engineering trade-offs and explicitly state the natural constraint $1 \le A \le n$.

---

### Official Review · Reviewer_Ge1m · 2025-11-21

**Soundness:** 3
**Presentation:** 3
**Contribution:** 3
**Rating:** 6
**Confidence:** 3

**Summary:**

This paper studies serverless federated learning (FL) that balances the trade-offs among multiple objectives: communication efficiency, network load distribution, model accuracy, and privacy guarantees. Unlike many solutions in the literature, the proposed ERIS FL framework can achieve good balances among all these objectives. The fundamental technique that ERIS leverages is a combination of a model partitioning strategy and a distributed shifted gradient compression mechanism. The authors demonstrate the superior performance both theoretically and through extensive experiments on image and text datasets.

**Strengths:**

1. I like the nice illustration of ERIS in Figure 1. The authors are able to clearly show how ERIS works during the client computation, shifted compression, model partitioning, and distributed training.

2. I also like it that the authors go beyond the small scale experiments on MNIST, FLMNIST many other papers would use and test the proposed methods on larger scale datasets and models such as 1.3B GPT-Neo.

**Weaknesses:**

My major concern lies in the literature review. The authors consider balancing the trade-offs among many objectives: communication efficiency, network load distribution, model accuracy, and privacy guarantees. This clearly falls into the multi-objective /task federated learning domain. The authors also conduct the utility and privacy trade-off analysis and plot the Pareto frontier of the solutions in the experiments such as those in Figure 4. All of these indicate that the authors might have already been aware of the multi-objective nature of the problem. However, I did not see a systematic review of multi-objective federated learning papers. For example,

Yang, Haibo, et al. "Federated multi-objective learning." Advances in neural information processing systems 36 (2023): 39602-39625.

Kang, Yan, et al. "Optimizing privacy, utility and efficiency in constrained multi-objective federated learning." arXiv preprint arXiv:2305.00312 (2023).

Zhang, Xiaojin, et al. "Trading off privacy, utility, and efficiency in federated learning." ACM Transactions on Intelligent Systems and Technology 14.6 (2023): 1-32.

**Questions:**

Like what I have mentioned in the weakness section, I would like to see the authors have more discussion on the connections and differences with those literatures in multi-objective federated learning domains.

---

> ### Author Response · Authors · 2025-11-24
>
> ## Q1. Missing discussion of multi-objective federated learning and its relation to ERIS (Part 1)
> We sincerely thank the reviewer for this constructive comment. We agree that our setting naturally involves several competing design goals—model utility, privacy guarantees, communication efficiency, and network load—and that this connects to the growing literature on multi-objective / multi-task federated learning. To position ERIS more clearly, in the revised manuscript, **we will add a dedicated paragraph *“Multi-objective federated learning”* in the Related Work section**, where we introduce the concept, review representative methods (including [1,2,3]), and clarify similarities and differences with ERIS. We **will also add a short remark in the Future Works** highlighting multi-objective optimization as a natural direction for extending ERIS.
>
> ### Connections to existing multi-objective FL works.
> Multi-objective FL refers to FL schemes that explicitly treat more than one goal as an optimization target, rather than focusing solely on a single utility loss. This can happen either at the training level (e.g., combining several tasks or client losses into a joint objective) or at the system level (e.g., searching over hyperparameters to balance several objectives such as accuracy, latency, or privacy). In practice, these methods have been used to jointly optimize a variety of trade-offs: (i) *multi-task and fairness-aware utility*, where client or group losses are balanced to improve fairness and robustness [1,4,11,12,17,18]; (ii) *model complexity versus accuracy*, where network architecture or sparsification is evolved together with performance, often with communication savings as a side effect [5,6,16]; (iii) *communication/latency and energy versus accuracy*, particularly in wireless and IoT scenarios [7,8,9,10,13,19]; and (iv) *privacy–utility–efficiency*, where privacy leakage, test performance, and training/communication cost are treated as separate objectives [2,3,15]. **These works typically introduce an explicit multi-objective optimization layer—either by modifying local training objectives (e.g., fairness- or multi-task–aware losses) or by running multiple FL trainings over a population of configurations** (e.g., NSGA-II/III, DQN-based schedulers) and evolving them towards a Pareto front in terms of the chosen metrics. In the revised manuscript, we will summarize these families and cite representative methods in a dedicated “Multi-objective federated learning” paragraph in the Related Work section.
>
> ### Clarifying our perspective on “multi-objective” in ERIS.
> **In contrast, ERIS is not formulated as a multi-objective optimization algorithm** in the sense of [1–3]. At the optimization level, ERIS still minimizes a single training loss (e.g., cross-entropy) via standard stochastic gradients; communication efficiency, privacy, and network load are properties of the protocol (model partitioning, decentralized aggregation, shifted compression) rather than additional differentiable objectives. Canonical multi-objective FL studies such as [1,2,4,11,17,18] explicitly incorporate multiple objectives into the training or an outer-loop search (e.g., NSGA-II/III over DP noise, sparsity, participation ratio, architecture, etc.), often at the cost of running additional FL training rounds. By contrast, **ERIS proposes a new architectural design**—serverless sharded aggregation with structured compression—**that, for a fixed set of training hyperparameters, (i) preserves the exact FedAvg update** (no utility loss from the privacy mechanism itself), **(ii) reduces per-client communication and balances network load, and (iii) provides an information-theoretic bound on user-level leakage** for any non-colluding aggregator. Our empirical Pareto plots are therefore meant to visualize the trade-offs induced by this protocol, rather than to perform multi-objective optimization during training.
>
> *—The response continues in the next comment—*

---

> > ### Author Response · Authors · 2025-11-24
> >
> > ## Q1. Missing discussion of multi-objective federated learning and its relation to ERIS (Part 2)
> > ### Relation to privacy–utility–efficiency trade-offs and the No-Free-Lunch theorem.
> > Among the works cited by the reviewer, [2,3], and [15] are particularly closely related, as they explicitly model privacy leakage, utility loss, and efficiency reduction as three competing objectives. **All these methods adopt protection mechanisms that *distort* the original model updates** (e.g., DP noise, quantization, sparsification) **or apply *cryptographic masking*** (e.g., homomorphic encryption). Their analysis is therefore conducted in a setting where a single stream of model information is protected, and **it is natural to assume that stronger protection inevitably incurs additional communication or computation cost and, in practice, some utility loss**. Under these assumptions, Zhang et al. [3] also derive a *No-Free-Lunch theorem* in a unified FL framework, showing that it is unrealistic to expect a privacy-preserving FL algorithm to simultaneously achieve infinitesimal privacy leakage, utility loss, and efficiency reduction in such scenarios.
> >
> > **ERIS follows a different route**. Instead of distorting a single message, it *proposes a new FL architecture* by partitioning each client update across multiple non-colluding aggregators. Any individual aggregator only observes a random shard plus compression, while the global update reconstructs exactly the FedAvg iterate. This sharded, noiseless design yields a non-trivial information-theoretic user-level leakage bound for any single aggregator (and reduces per-client communication compared to FedAvg). As a result, the monotonic *“more protection ⇒ more cost / less utility”* assumptions used in [3] (in particular, that protection necessarily increases cost and harms utility) do not hold in our setting, and therefore their theorem does not directly constrain ERIS. Conceptually, however, our work is aligned with the high-level message of [2,3,15]: ERIS does not claim to eliminate the privacy–utility–efficiency trade-off, but to **occupy a more favorable region in this design space than existing perturbation- or cryptography-based schemes by leveraging its noiseless and communication-efficient architectural partitioning**.
> >
> > ### Why we did not cast ERIS as a multi-objective optimization problem (and future work).
> > **Our privacy-enhancing mechanism is largely architecture-driven and does not introduce a tunable privacy parameter that trades off directly against utility** (as in DP noise, sparsification strength, or encryption depth). The main structural parameter is the number of aggregators $A$, which increases privacy by reducing the fraction of a client’s update visible to each aggregator, without degrading the FedAvg iterate or inflating communication per client. Therefore, **tuning $A$ is decoupled from the model-utility objective; its upper bound is mostly dictated by system constraints** on the feasible number and placement of aggregators.
> >
> > In practice, the remaining trade-off in ERIS is dominated by the *compression rate* in the shifted compression scheme (utility vs. communication). In this work, we deliberately choose compression ratios in the *utility-preserving regime*, i.e., such that ERIS matches the FedAvg utility across all experiments, because even under this conservative choice, ERIS already achieves substantial improvements in communication and distribution time compared to FedAvg and other baselines (e.g., transmitting only ~1% of model parameters and reducing distribution time to ~0.1% of centralized FedAvg in Table 2). However, we see it as a promising and interesting direction for future work to relax this constraint and explicitly explore the utility–communication trade-off: multi-objective optimization techniques from [2,13,15] could be integrated on top of ERIS to tune compression rates and system-level hyperparameters, given deployment-specific constraints and preferences. We will explicitly mention this in the revised Future Work section.
> >
> > ---
> >
> > We hope that this additional discussion, together with the new paragraph in Related Work, will address the reviewer’s concern and clarify how ERIS relates to and complements the existing body of work on multi-objective FL. We would also be grateful for any additional relevant works that the reviewer considers important and that we may have missed in our current survey.
> >
> > *—All references are in the next comment—*

---

> > > ### Author Response · Authors · 2025-11-24
> > >
> > > ---
> > >
> > > **References:**
> > > 1. H. Yang et al. Federated Multi-Objective Learning, NeurIPS (2023)
> > > 2. Y. Kang et al. Optimizing Privacy, Utility, and Efficiency in Constrained Multi-Objective Federated Learning, ACM TIST (2024)
> > > 3. X. Zhang et al. Trading Oﬀ Privacy, Utility and Eﬃciency in Federated Learning, ACM TIST (2023)
> > > 4. Z. Hu et al. Federated Learning Meets Multi-Objective Optimization, IEEE TNSE (2022)
> > > 5. H. Zhu et al. Multi-Objective Evolutionary Federated Learning, IEEE TNNLS (2020)
> > > 6. Z. Chai et al. Communication efficiency optimization in federated learning based on multi‑objective evolutionary algorithm, Evol. Intel. (2023)
> > > 7. A. Lakhan et al. Federated Learning-Aware Multi-Objective Modeling and blockchain-enable system for IIoT applications, Comput. Electr. Eng. (2022)
> > > 8. Y. Zhou et al. Multi-Objective Optimization for Bandwidth-Limited Federated Learning in Wireless Edge Systems, IEEE ComSoc (2023)
> > > 9. T. Xu et al. A DQN-Based Multi-Objective Participant Selection for Efficient Federated Learning, Future Internet (2023)
> > > 10. J. A. Morell et al. A multi-objective approach for communication reduction in federated learning under devices heterogeneity constraints, FGCS (2024).
> > > 11. Y. Shen et al. Multi-objective federated learning: Balancing global performance and individual fairness, FGCS (2025)
> > > 12. C. Wang et al. Fair Federated Learning with Multi-Objective Hyperparameter Optimization, ACM TKDD (2025)
> > > 13. I. Zidi et al. An approach based on NSGA-III algorithm for solving the multi-objective federated learning optimization problem, IJIT (2024)
> > > 14. S. M. Hamidi et al. Over-the-Air Fair Federated Learning via Multi-Objective Optimization, IEEE COMML (2025).
> > > 15. D. Geng et al. Multi-­ Objective Federated Averaging Algorithm, Expert Systems (2024)
> > > 16. M. M. Abushaega et al. Multi-objective sustainability optimization in modern supply chain networks: A hybrid approach with federated learning and graph neural networks, Alex. Eng. J. (2024)
> > > 17. Z. Pan et al. FedMDFG: Federated Learning with Multi-Gradient Descent and Fair Guidance, AAAI (2023)
> > > 18. M. Badar et al. FairTrade: Achieving Pareto-Optimal Trade-Offs between Balanced Accuracy and Fairness in Federated Learning, AAAI (2024)
> > > 19. S. P. Singh et al. Federated Learning-Based Multi-Objective Optimization for IoT-Enabled Distributed Environmental Monitoring in Consumer Electronics, IEEE Trans. Consum. Electron. (2025)

---

### Author Response · Authors · 2025-11-28

Dear Reviewers,

We would like to thank you again for the thoughtful and constructive feedback provided so far. We have carefully worked to address all comments and have uploaded detailed responses in the rebuttal.

If there is anything else you would like us to clarify or expand on, we would be very happy to do so.

Thank you again for your time and engagement with our work.

---

### Author Response · Authors · 2025-11-30
**Author Final Remarks**

We extend our sincere gratitude to the AC and all reviewers for their invaluable and constructive feedback throughout this review process. We are encouraged by their recognition of ERIS’s key strengths: the novelty of our decentralized, shard-based aggregation (_5ugv, u1h3_), which “_unlike many solutions in the literature_” enables ERIS to “_achieve good balances among communication efficiency, network load, model accuracy and privacy guarantees_” (_Ge1m_); the well-motivated problem setup and clear presentation (_u1h3, 5ugv, Ge1m_); the solid convergence and privacy analysis (_u1h3, RH9A_); and the systematic evaluation across datasets, model scales, baselines, and attack scenarios (MIAs, DRAs), including successful scaling to billion-parameter models (_u1h3, RH9A, Ge1m_). To this end, we summarize the key improvements made during rebuttal that helped strengthen our core contributions:
- **Clarified positioning and claims.** By primarily expanding the Related Work to explicitly contrast ERIS with LiPFed, SoteriaFL, Ako, Shatter, and recent multi-objective FL methods, we clarified why ERIS is the first FL framework that simultaneously (i) preserves the exact FedAvg iterate sequence, (ii) achieves communication efficiency comparable to or better than state-of-the-art compressed baselines, and (iii) provides explicit information-theoretic user-level leakage bounds without noise injection—while prior methods typically satisfy at most two of these properties.
- **Strengthened empirical privacy analysis under collusion.** We added new experiments measuring MIA accuracy as increasing fractions of clients collude, showing that ERIS consistently leaks less than FedAvg even under strong collusion (up to 50%), and remains close to the minimum-leakage regime for small colluding groups—empirically confirming the behaviour predicted by Corollary D.2
- **Extended convergence analysis and utility comparison.** We introduced Corollary C.6, specializing Theorem 3.6 to SGD and clarifying the dependence on the learning rate. Following Reviewer 5ugv’s suggestion, we also added a comparison table situating ERIS’s convergence guarantees alongside representative DP and compressed FL methods, highlighting that ERIS achieves a dimension-free, fully sharded convergence bound without noise-related factors, and matches the order of shifted-compression methods while relaxing their convergence constraints.
- **Robustness under aggregator failures/dropout.** We added a new experiment analyzing aggregator dropout, showing that ERIS remains accurate and stable up to high dropout levels (e.g, 60%), and outlined natural extensions for failure detection and shard reassignment.
- **Expanded communication and systems-level evaluation.** Following RH9A suggestions, we extended our Table 2 by adding for each method, per-client upload/download, total communication cost per round in addition to the adopted compression rate, per-client communication cost, and distribution time. To separate the effects of compression and decentralized aggregation, we now also report ERIS (i) without compression, (ii) at the same compression rate as SoteriaFL, and (iii) under the more aggressive compression used in the main experiments.

The review process helped us improve the clarity, theoretical framing, and empirical validation of ERIS, while the core contributions and evaluation remain consistent with the original submission: a serverless, sharded FL framework that preserves FedAvg-level utility, substantially improves communication efficiency and distribution time, and provides meaningful user-level privacy amplification. We hope these revisions address the reviewers’ concerns and convey why we believe ERIS occupies a favorable region in the privacy–utility–efficiency design space for practical and scalable FL deployments.

---

### Meta-Review · Area_Chair_wWJr · 2025-12-19

**Summary:**

The paper presents a communication-efficient distributed learning method that is claimed to provide strong privacy guarantees.

The reviewers note the following weaknesses:
1. Limited literature review on multi-objective FL.
2. Overblown novelty claim.
3. Not defined how masks are coordinated.
4. Limited discussion of privacy guarantees under collusion and stronger adversaries.
5. Unexplained performance degradation as $A$ grows.
6. Coordination required to track shifting reference vectors.
7. Lack of tightness of Theorem 3.6.
8. Lack of justification for claimed novelty in communication efficiency.
9. Claimed communication efficiency does not reduce per-client communication, just distributes it to multiple aggregators.
10. Missing analysis of aggregator dropouts.
11. Unfair comparison in Fig. 6 under differing compression ratios.

Finally, I have one very important concern not raised by any of the reviewers: misleading privacy claims and lack of transparency about the limitations of the method.

The authors promote their method as "enabling strong privacy guarantee" and reducing vulnerability to membership inference "close to the unattainable ~64% limit".

While the proposed method may be close to optimal among algorithms that do not add any noise, this is not true more generally. Algorithms based on variants of differential privacy can clearly provide more privacy and break the "unattainable ~64% limit". The lack of transparency regarding the provided level of privacy is unacceptable because it can confuse less knowledgeable readers.

I would strongly encourage the authors to review the Shatter paper which is very open about this limitation.

**Reviewer Concerns:**

The authors provide extensive responses to the reviews and claim to have made changes to the submission. However, in many cases I was unable to verify that the promised changes were actually made. At least some of the new references that the authors claim to have added are not in the paper. The final version does not include any highlights for changes and I cannot access any previous version for comparison.

Analysis of specific comments:
1. Authors provide extensive reply but no changes found in the paper.
2. Changes promised, not visible in the paper.
3. Solution requires a coordinator, which the authors promise to mention in the paper. What is not mentioned that this directly contradicts the central claim of serverless FL, even if that server can be more lightweight.
4. New analysis and experiment on collusion added in the supplement.
5. Clarified by the authors.
6. Clarified by the authors, although I am not sure if this properly addresses the point: there will be coordination overhead even if it may be small. More careful analysis under realistic a communication model would strengthen the paper.
7. Clarified by the authors to be a limitation of the technique.
8. The authors promise clarifications to the text.
9. The authors provide new results (in the supplement) showing that lower compression ratio implies less communication. This is hardly suprising, and more transparent comparisons would be needed.
10. New results on aggregator added in the supplement.
11. Added results for uncompressed ERIS in the figure.

Overall, while the rebuttal and revision address some concerns, many of the revisions remain promises. These text changes would have been easy to implement, so it is not clear why the authors have not implemented them already.

**Reviewer Scores:**

Given the lack of concrete improvements to the paper to address many of the critical comments, I find it unlikely that at least reviewers u1h3, 5ugv and RH9A would have raised their scores.

Furthermore, I feel that the misleading privacy claims and lack of transparency on limitations of the method are so severe flaws that the paper should be rejected in any case.

---

### Decision · Program_Chairs · 2026-01-26

Reject